# Nitrogen metabolism profiling reveals cell state-specific pyrimidine synthesis pathway choice

Milan R. Savani [1,2], Bingbing Li[1], Bailey C. Smith [1], Wen Gu [1], Yi Xiao [1], Gerard Baquer [3,4], Tracey Shipman[1], Skyler S. Oken [5,6], Namya Manoj[5,6], Lauren G. Zacharias[1], Vinesh T. Puliyappadamba[1], Sylwia A. Stopka[3,4], Michael S. Regan [3,4], Michael M. Levitt [1], Charles K. Edgar[1,2], William H. Hicks [1,5,6], Soummitra Anand[1], Misty S. Martin-Sandoval[1], Rainah Winston[1], João S. Patrício [1], Xandria Johnson[1], Trevor S. Tippetts [1], Diana D. Shi[1,7,8], Andrew Lemoff [9], Timothy E. Richardson[10], Pascal O. Zinn[5,6], Ashley Solmonson [11,12], Thomas P. Mathews [1], Nathalie Y. R. Agar[3,4,13], Ralph J. DeBerardinis [1,14,15,16,17], Kalil G. Abdullah [5,6] ✉ & Samuel K. McBrayer [1,15,17,18] ✉

Stable isotope-tracing assays track few metabolites, yet cells use many nutrients to sustain nitrogen metabolism. Here we create a platform for tracing 30 nitrogen isotope-labelled metabolites in parallel to enable a system-level understanding of cellular nitrogen metabolism. This platform reveals that while primitive cells engage both de novo and salvage pyrimidine synthesis pathways, differentiated cells nearly exclusively salvage uridine. This link between cell state and pyrimidine synthesis pathway preference persists in murine and human tissues. Mechanistically, we find that S1900 phosphorylation of CAD, the first enzyme of the de novo pathway, is induced by uridine deprivation in differentiated cells and constitutively enriched in primitive cells. Mimicking CAD S1900 phosphorylation in differentiated cells constitutively activates de novo pyrimidine synthesis, while blocking this modification impairs the cellular response to uridine starvation. Collectively, we establish a method for nitrogen metabolism profiling and define a mechanism of cell state-specific pyrimidine synthesis pathway choice.

Technical advances in metabolite profiling and stable isotope tracing have broadened understanding of the influence of metabolism on physiology and disease. Steady-state metabolite profiling measures metabolite pool sizes, while stable isotope tracing captures dynamic metabolic activities. In tracing, a stable isotope (typically [13]C or [15]N) is incorporated into a nutrient and its metabolic fate is then tracked throughout biochemical networks before quantification by magnetic resonance spectroscopy or mass spectrometry. Development of high-resolution mass spectrometers has expanded the number of quantifiable metabolites and enhanced discrimination between isotopes of different atoms[1]. The impact of these advances, however, has not manifested evenly across biochemical pathways. In vivo tracing studies using [13]C isotope-labelled nutrients have changed our understanding of central carbon metabolism in organs and tumours[2–5]. In contrast, the mechanisms used to incorporate nitrogen into the metabolome remain comparatively obscure[6]. This difference is largely attributable to the

greater diversity of nutrients cells use to support nitrogen metabolism compared with carbon metabolism[6,7]. Broad coverage of central carbon metabolic pathways can be achieved by tracing a small number of [13]C-labelled nutrients, such as glucose, glutamine, lactate, acetate and free fatty acids[8]. To similarly cover nitrogen metabolic pathways, many [15]N-labelled tracers would be required. This is a key challenge, as current experimental paradigms are not optimized for multiplexed stable isotope tracing.

To address this limitation, we hypothesized that we could generate a system-level portrait of nitrogen metabolism programmes using a library of cell culture medium preparations to trace numerous [15]N-labelled nutrients in parallel. In these media, we sought to replace a single nitrogenous metabolite from each stock with its [15]N-labelled isotopologue. After culturing cells in these media and analysing intracellular metabolites, data from dozens of individual stable isotope-tracing assays could be computationally integrated to provide a comprehensive view of nitrogen metabolism substrate preferences and pathway activities. To maximize the physiological relevance of our findings, we leveraged human plasma-like medium (HPLM), in which nutrient concentrations mirror those in human plasma[9,10]. Our previous research using stable isotope tracing to comparatively assess substrate preferences for nucleotide biosynthesis in glioma cells provided a template for this effort[11]. By developing this nitrogen metabolism profiling platform and applying it to study malignant and non-transformed neural cells, our work revealed a fundamental link between cellular differentiation state and pyrimidine synthesis pathway preference with implications for cancer and developmental biology.

## Results

### Development and validation of a nitrogen metabolism profiling platform

Cells use various nitrogen-containing nutrients to produce metabolites, including nucleotides, amino acids, heme, creatine, amino sugars, nucleotide sugars, glutathione and polyamines. To comprehensively map contributions of individual nutrients to nitrogen metabolic pathways, we first created a library of 30 HPLM stocks that each contained a distinct stable isotope tracer (Fig. 1a). In each, we replaced a single nitrogenous metabolite with its [15]N isotope-labelled counterpart. We included tracers corresponding to the most abundant nitrogenous compounds in human plasma: all 20 proteinogenic amino acids, non-proteinogenic amino acids (taurine), urea cycle intermediates (ornithine and citrulline), nucleotide precursors (hypoxanthine and uridine), ions (nitrate and ammonium) and end products of nitrogen metabolism (urea and uric acid) (Supplementary Table 1). We also generated tracer-free HPLM as a negative control. We then could perform parallelized stable isotope tracing in paired cultures to elucidate the impact of individual variables (such as drug treatments, genetic mutations or cell states) on nitrogen metabolism. After analysing metabolite extracts using high-resolution liquid chromatography–mass spectrometry (LC–MS)[12,13], we built a computational pipeline to quantify and deconvolute the resulting dataset. Quantitative labelling data for each tracer–metabolite pair (30 [15]N-labelled nutrients traced into >100 nitrogen-containing metabolites) generates: (1) Sankey diagrams describing global nitrogen metabolism, (2) a rank-ordered list of differences in tracer metabolism between cultures (reflected by the 'differential labelling score') and (3) steady-state metabolite levels in each condition.

We first confirmed selective labelling of the metabolite pool of interest in each stock (Extended Data Fig. 1a). Isotope enrichment was not observed in tracer-free medium, indicating specificity of these patterns. We did not detect cysteine (likely due to spontaneous oxidation to cystine) nor quantify ammonia, nitrate or urea (owing to their low mass-to-charge ($m/z$)). We found minimal variance across medium stocks in cognate metabolite pool size for each tracer (Extended Data Fig. 1b). We next sought to validate our

platform by attempting to 'rediscover' an established mechanism of nitrogen metabolism dysregulation. Previously, we reported that (R)-2-hydroxyglutarate ((R)-2HG), the product of mutant isocitrate dehydrogenase (IDH) oncoproteins, competitively inhibits branched-chain amino acid transaminase (BCAT)-dependent glutamate synthesis[14] (Fig. 1b). To compensate, IDH-mutant tumour cells activate glutaminase (GLS)-dependent glutamate synthesis. Treatment of these cells with a mutant IDH inhibitor, such as AGI-5198 (ref. 15), causes (R)-2HG levels to fall, derepressing BCAT enzymes and reestablishing a balance between branched-chain amino acids and glutamine as nitrogen donors to glutamate. Because glutamate is incorporated into glutathione (GSH), similar relationships apply to GSH synthesis. Using BT054, IDH1-mutant glioma stem-like cells (GSCs) derived from a patient with grade 3 oligodendroglioma[16], we tested whether nitrogen metabolism profiling could uncover these metabolic interactions in an unbiased manner.

We first confirmed expression of the IDH1-R132H mutant enzyme in BT054, benchmarking against levels in isogenic GSCs from genetically engineered mouse models of IDH-mutant and IDH-wild-type astrocytoma[11,17,18] (Fig. 1c). Next, we treated BT054 with dimethylsulfoxide (DMSO) vehicle or AGI-5198 to create (R)-2HG[high] and (R)-2HG[low] conditions, respectively, before culture in our HPLM library for 18 h. We confirmed successful labelling of intracellular metabolite pools in each tracer-containing medium stock (Extended Data Fig. 1c) and expected effects of AGI-5198 on intracellular metabolite pool sizes (2HG depletion and glutamate upregulation; Fig. 1d and Supplementary Table 2). Finally, we employed our computational pipeline to systematically assess how (R)-2HG accumulation affects nitrogen metabolism (Extended Data Fig. 1d). We constructed colour-coded Sankey diagrams to provide a system-level representation in which tracers and metabolites representing individual biochemical pathways (Supplementary Table 1) at the top and bottom of these plots, respectively, are connected by lines of widths correlating with labelling between them (Fig. 1e,f). Consistent with our previous hypothesis-driven work[14] (Fig. 1b), nitrogen metabolism profiling revealed that (R)-2HG accumulation diminishes branched-chain amino acid (BCAA)-dependent glutamate production and stimulates compensatory glutamine-dependent glutamate synthesis (Fig. 1g,h).

Other platform outputs also captured (R)-2HG-driven changes in glutamate metabolism. Differential labelling scores reflecting metabolism of [15]N-BCAAs to glutamate and glutathione were elevated in AGI-5198-treated cells whereas those for [15]N$_2$-glutamine-dependent labelling of these metabolites were decreased (Fig. 1i and Supplementary Table 2). Differential labelling scores reflected marked differences in BCAA and glutamine metabolism between the two conditions (Fig. 1j,k). Glutamate and glutathione levels increased as (R)-2HG declined (Extended Data Fig. 1e–g), consistent with differences in BCAT activity. We performed low-throughput stable isotope tracing experiments to validate these findings, showing that AGI-5198 treatment increases BCAA labelling of glutamate and glutathione pools (Extended Data Fig. 1h,i). These data validate platform performance and suggest that it may be useful for unbiased discovery of nitrogen metabolism programmes.

### Nitrogen metabolism profiling reveals a link between pyrimidine synthesis pathway preference and cell state

We next sought to apply this tool to a fundamental question: how do nitrogen metabolic programmes differ between malignant and non-malignant cells? We therefore profiled BT054 GSCs and immortalized normal human astrocytes (NHA)[19]. These lines differ in both malignancy and differentiation state: BT054 are primitive and transformed whereas NHA are differentiated and non-tumorigenic. We first optimized plating to avoid medium nutrient depletion (Extended Data Fig. 2a) and confirmed intracellular tracer accumulation (Extended Data Fig. 2b). Differences in intracellular tracer accumulation between lines may

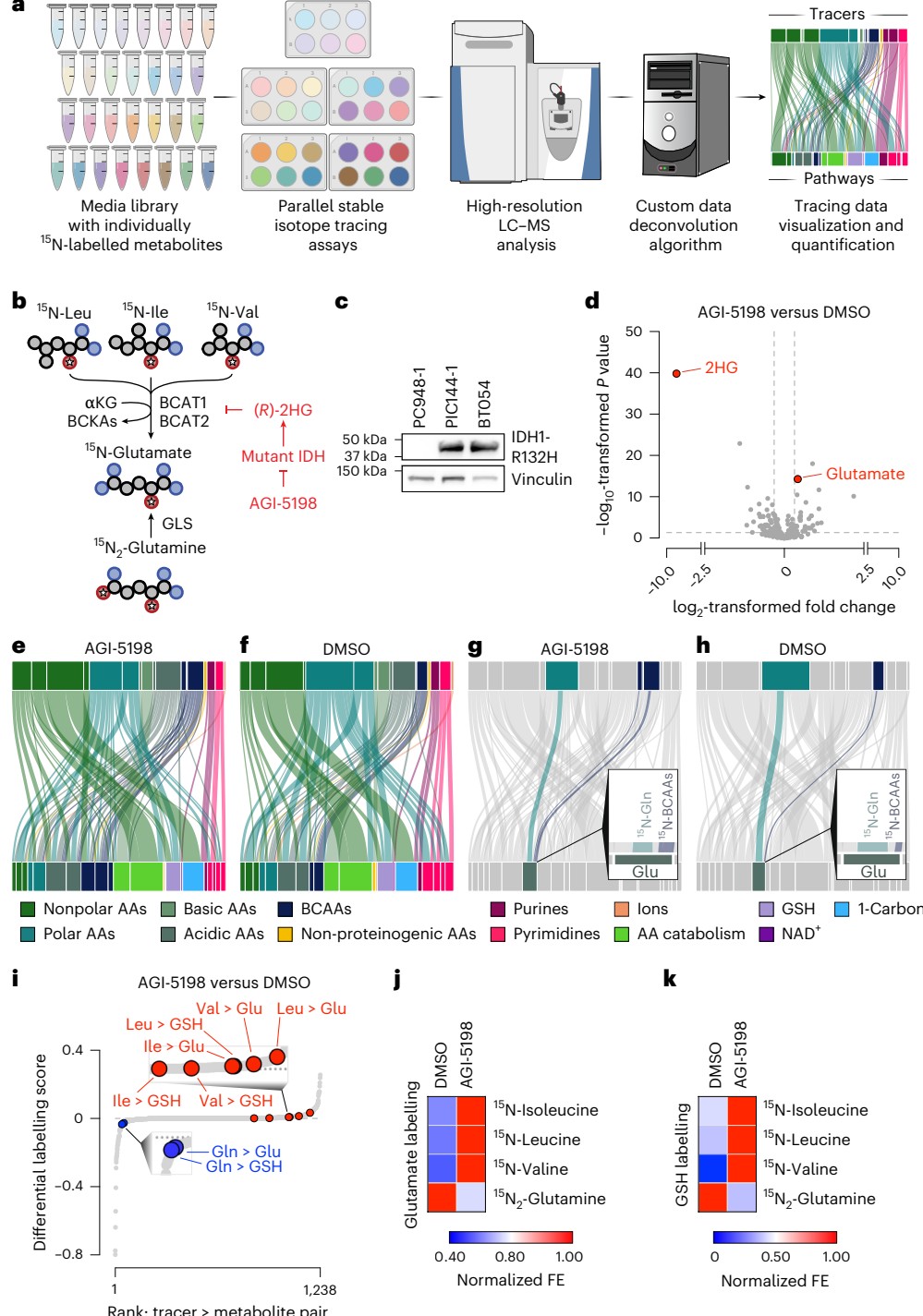

**Fig. 1 | Development and validation of a nitrogen metabolism profiling platform. a**, Schema depicting nitrogen metabolism profiling platform. **b**, Schema of BCAT- or GLS-dependent nitrogen transfer. αKG, α-ketoglutarate; BCKA, branched-chain ketoacids. **c**, Immunoblot of IDH1-R132H expression in BT054 and murine primary glioma cell lines derived from IDH1 WT (PC: *Pik3ca^{mut}*, *Trp53^{mut}* and *Atrx^{mut}*) or IDH1-R132H mutant (PIC, *Idh1^{R132H}*, *Pik3ca^{mut}*, *Trp53^{mut}* and *Atrx^{mut}*) GSC lines (*n* = 1). **d**, Nitrogen metabolism profiling in BT054 treated with 3 µM AGI-5198 or DMSO for 72 h, followed by tracing for 18 h (*n* = 1 per tracer per treatment). Volcano plot of metabolite levels (*n* = 30). Two-tailed *P* values were determined by an unpaired *t*-test. **e–h**, Sankey diagrams depicting labelling from 15N-labelled tracers (top bars) to representative intermediates of metabolic pathways (bottom bars) in AGI-5198 (**e,g**) or DMSO (**f,h**) conditions. Width of connecting lines indicates fractional enrichment of tracer in representative intermediate. Insets in **g,h** highlight BCAA or glutamine contribution to glutamate. AA, amino acid; NAD+, nicotinamide adenine dinucleotide. **i**, Waterfall plot ranking differential labelling scores. Blue, BCAA labelling of Glu and GSH; red, Gln labelling of Glu and GSH. **j,k**, Heatmap of label from indicated tracer (*y* axis) in glutamate (**j**) or GSH (**k**) in each treatment (*x* axis). Fractional enrichments normalized to condition with maximal labelling for each tracer. See also Extended Data Fig. 1. FE, fractional enrichment. Credit: tubes, Helicase 11 under a Creative Commons license CC BY 4.0; plate icons, Bioicons.com under Creative Common license CC0 1.0; mass spectrometer icon, NIAID; computer icon, Freesvg.org.

represent differential uptake, utilization or synthesis. Many differences in nitrogen metabolism existed between these lines (Fig. 2a,b), including substrate preference for pyrimidine nucleotide synthesis (Fig. 2c–e and Supplementary Table 3). NHA nearly exclusively used uridine to sustain uridine monophosphate (UMP), uridine diphosphate (UDP) and uridine triphosphate (UTP) pyrimidine pools, whereas BT054 predominantly used glutamine. Pyrimidine nucleotides can be produced by de novo synthesis (for which glutamine, aspartate and bicarbonate serve as substrates) or uridine-dependent salvage[20] (Fig. 2f). Therefore, NHA cells rely principally on pyrimidine salvage and BT054 cells engage both de novo and salvage pathways. De novo pathway intermediates and pyrimidine nucleotides were nearly all elevated in BT054 relative to NHA, whereas uridine was diminished (Extended Data Fig. 2c,e–k and Supplementary Table 3). Notably, enhanced de novo pyrimidine nucleotide synthesis could not be explained by the demands of rapid proliferation because NHA divided more quickly than BT054[11] (Extended Data Fig. 2d). Differing growth factors/serum conditions used to culture either line also did not account for these differences (Extended Data Fig. 3b,c). While the pyrimidine substrates aspartate and uridine demonstrated differences in cognate intracellular pool labelling (Extended Data Fig. 2b), this may reflect selective expression of the Excitatory Amino Acid Transporter 1 (EAAT1 or SLC1A3) in differentiated glia and de novo pyrimidine synthesis-dependent uridine production in BT054, respectively.

De novo pyrimidine synthesis is aberrantly activated and is being investigated as a tumour-selective vulnerability in a spectrum of human cancers[20]. However, the determinants of pyrimidine synthesis pathway choice in individual cells are not well defined. We profiled BT054 and NHA under identical nutrient conditions, implying that pyrimidine synthesis pathway preference is cell-autonomous. To investigate this idea, we evaluated de novo and salvage pyrimidine synthesis pathways in a panel of cell lines by quantifying UMP labelling from amide-15N-glutamine and 15N2-uridine tracers. We first confirmed specific labelling of cognate intracellular pools of traced metabolites (Extended Data Fig. 3a). GSCs derived from both IDH-wild-type glioblastomas (GBMs) and IDH-mutant gliomas constitutively activated de novo pyrimidine synthesis (Fig. 2g–k). Differentiated, non-transformed cells (immortalized astrocytes derived from two different donors and BJ fibroblasts), however, displayed minimal de novo synthesis but robust salvage pathway flux (Fig. 2l–n). To ask whether these differences were associated with differentiation state or malignancy, we also analysed two non-transformed yet primitive neural stem cell (NSC) lines. These lines displayed comparable de novo pyrimidine synthesis activity to GSCs (Fig. 2o,p and adherent culture conditions in Extended Data Fig. 3d-e). Data in Fig. 2g-p were normalized for differences in intracellular glutamine labelling (Extended Data Fig. 3a); we also include un-normalized data in Supplementary Fig. 1.

To mitigate confounding from genetic differences, we created isogenic primitive and differentiated cell cultures then compared their pyrimidine synthesis programmes. We expanded differentiated human normal fibroblasts (HNFs) or reprogrammed them to induced pluripotent stem cells (iPSCs), then primitive neural progenitor cells (NPCs) or differentiated neurons (Fig. 2q). We confirmed differentiation state by RNA sequencing, examining the fibroblast markers *DCN* and *THY1*, primitive cell markers *OLIG2*, *NES* and *SOX2*, or interneuron markers *DLX1* and *DLX2* (Fig. 2r). NPCs recapitulated the preference for de novo pyrimidine synthesis observed in GSCs and NSCs, whereas HNFs and neurons, like other differentiated cells, predominantly relied on pyrimidine salvage (Fig. 2s-u). Divergent pyrimidine synthesis programmes in GSCs, differentiated cells and NSCs/NPCs were not explained by tracer accumulation (Extended Data Fig. 3a). Taken together, our findings indicate that differentiation state is a cell-intrinsic determinant of pyrimidine synthesis pathway choice.

To test the relevance of de novo pyrimidine synthesis activity to primitive cell identity, we evaluated how de novo pyrimidine synthesis affects NPC self-renewal. We treated NPCs with an inhibitor (orludodstat[21]) of the de novo pyrimidine synthesis enzyme dihydroorotate dehydrogenase (DHODH) under a range of uridine conditions. DHODH inhibition reduced NPC self-renewal in the absence of uridine (Fig. 2v). This effect on self-renewal could not be fully rescued by either physiological (3 μM) and supraphysiological (100 μM) levels of uridine (Fig. 2w-y). Cytotoxicity did not account for decreased self-renewal (Extended Data Fig. 3f). These findings suggest that de novo pyrimidine synthesis is necessary for NPC self-renewal and that pyrimidine salvage cannot fully compensate for this requirement.

We next tested whether uridine was required for NPC differentiation to neurons (Fig. 2z). Uridine deprivation did not impact the expression of primitive cell or interneuron markers (Extended Data Fig. 3g-j). Availability of uridine, however, promoted expression of gene programmes that are vital to neuronal function, including regulation of cell polarity, neuronal axonogenesis and neuronal migration (Fig. 2aa,bb). Collectively, these data suggest that the switch from reliance on de novo pyrimidine synthesis to pyrimidine salvage during neuronal differentiation promotes the establishment of neuronal function.

## Cellular differentiation states are associated with distinct pyrimidine synthesis programmes in primary brain tissue

Gliomas are enriched in primitive cell populations while adult brain parenchyma largely comprises terminally differentiated neural cells[22]. We exploited this distinction to ask if cell state-specific differences in pyrimidine synthesis are observed in vivo. We randomized naive mice or mice bearing MGG152 (ref. 23) orthotopic patient-derived xenografts of IDH-mutant glioma to continuous infusions of amide-15N-glutamine, 15N2-uridine or saline and performed conventional and spatial metabolomics on brain tissues (Fig. 3a). Approximately 40% of plasma glutamine and uridine pools were labelled in tracer-infused mice, leading to ~20–25% and ~15% labelling of glutamine and uridine, respectively, in non-neoplastic brain tissue and glioma xenografts (Fig. 3b-e). Glutamine infusion did not elevate plasma or brain glutamine levels (Extended Data Fig. 4a-b). Although uridine infusion

**Fig. 2 | Nitrogen metabolism profiling links pyrimidine synthesis pathway choice and differentiation state. a–d**, Nitrogen metabolism profiling in BT054 and NHA donor #1 for 18 h (*n* = 1 per tracer per line). Sankey diagrams depicting labelling from 15N tracers (top bars) to representative intermediates of metabolic pathways (bottom bars) in BT054 (**a,c**) or NHA (**b,d**). Insets in **c,d** highlight glutamine and uridine contribution to pyrimidines. **e**, Waterfall plot ranking differential labelling scores. **f**, Schema of pyrimidine synthesis pathways. **g–p**, Amide-15N-glutamine (Gln) and 15N2-uridine tracing to UMP in GSC (**g–k**), differentiated (**l–n**) or NSC (**o,p**) lines (*n* = 3 for all except HK308, for which *n* = 2). **q**, Schema of NPC and neuron derivation from HNFs. **r**, Heatmap depicting relative abundance of indicated transcripts in HNF, NPC and NPC-derived neurons. Data show counts per million mapped reads (CPM) from RNA sequencing, scaled to maximal expression in each row (*n* = 3 per line). **s–u**, Amide-15N-glutamine and 15N2-uridine tracing to UMP in HNF (**s**), NPC (**t**)

and NPC-derived (**u**) neurons (*n* = 3). All amide-15N-glutamine to UMP labelling data are normalized to glutamine M + 1 FE. Un-normalized data are available in Supplementary Fig. 1. **v–y**, Self-renewal capacity of NPCs treated with DMSO or 10 nM orludodstat and cultured in 0 μM (**v**), 3 μM (**w**) or 100 μM (**x**) uridine (Urd), quantified in **y**. Triangles indicate transformed fraction nonresponding of 0. **z**, Schema of neuron differentiation from NPCs. **aa**, Selected overrepresented gene sets in neurons differentiated from NPCs in 3 μM versus 0 μM uridine (*n* = 3 per condition). **bb**, Heatmap depicting relative abundance of central nervous system neuron axonogenesis genes of NPC-derived neurons in 0 μM or 3 μM uridine. Data show CPM from RNA sequencing, scaled to maximal expression in each row (*n* = 3 per condition). *P* value was determined by one-way analysis of variance (ANOVA). For all panels, data show mean ± s.e.m. See also Extended Data Figs. 2 and 3. Credit: icons, NIAID.

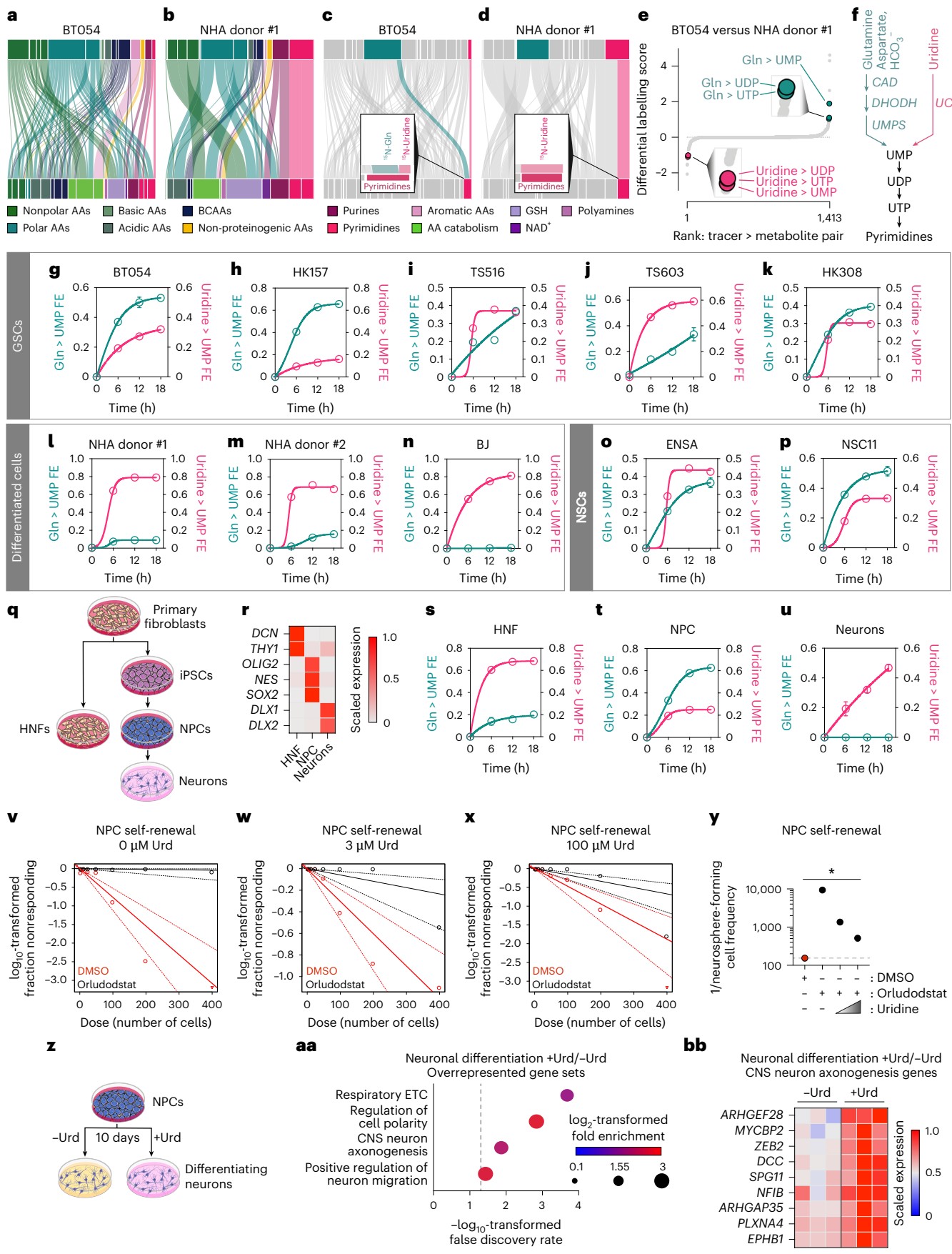

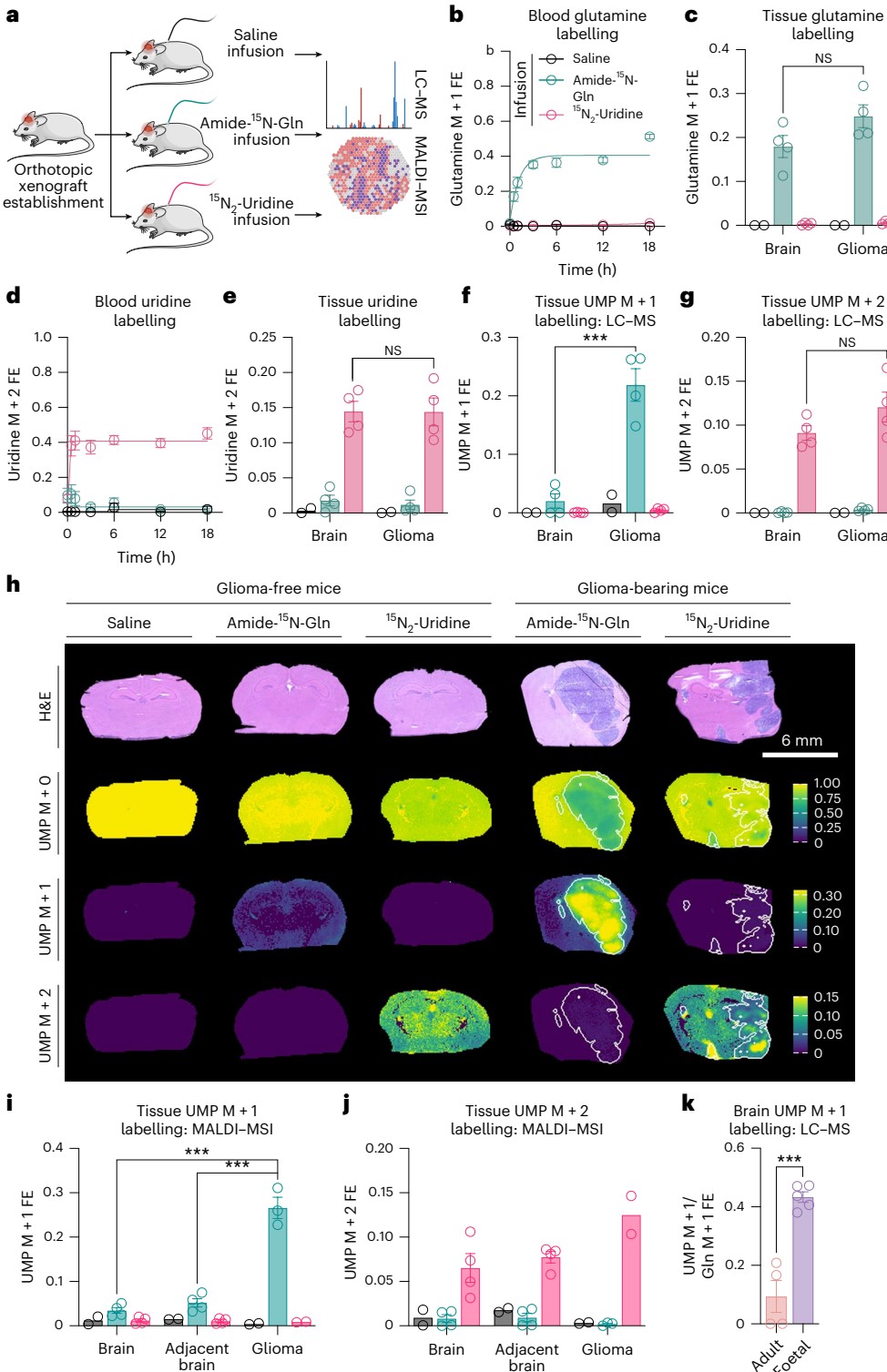

**Fig. 3 | Tissues enriched in primitive cells display constitutive de novo pyrimidine synthesis in vivo. a**, Schema depicting infusion with saline, amide-$^{15}$N-glutamine or $^{15}$N$_2$-uridine for 18 h in mice bearing MGG152 glioma orthotopic xenografts or non-tumour-bearing mice. **b,c**, Glutamine M + 1 FE in (**b**) blood or (**c**) tissue (per group $n$ = 2 for saline, $n$ = 4 for amide-$^{15}$N-glutamine and $^{15}$N$_2$-uridine). **d,e**, Uridine M + 2 FE in blood (**d**) or tissue ($n$ = 2 for saline, $n$ = 4 for amide-$^{15}$N-glutamine and $^{15}$N$_2$-uridine) (**e**). **f,g**, UMP fM + 1 ($P$ = 0.0006) (**f**) or M + 2 (**g**) FE in normal brain or glioma xenograft tissues (per group $n$ = 2 for saline, $n$ = 4 for amide-$^{15}$N-glutamine and $^{15}$N$_2$-uridine). **h**, Representative brain sections from mice bearing MGG152 xenografts. H&E stain (top). UMP M + 0 (second row), M + 1 (third row) or M + 2 (bottom) FE determined by MALDI–MSI. Scale bar, 6 mm.

**i,j**, MALDI-MSI-based quantification of UMP M + 1 ($P$ for brain-glioma = 0.0001, $P$ for adjacent brain-glioma = 0.0002) (**i**) or M + 2 FE (**j**) in indicated tissues (per group $n$ = 2 for tissues from saline-infused mice and for xenograft gliomas from $^{15}$N$_2$-uridine-infused mice; $n$ = 3 for xenograft gliomas from amide-$^{15}$N-glutamine-infused mice; $n$ = 4 for all others). **k**, UMP M + 1 FE relative to glutamine M + 1 FE in adult mouse brains infused for 18 h or foetal mouse brains from dams infused for 5 h ($n$ = 4 for adult and $n$ = 5 for foetal, $P$ = 0.0003). NS, not significant, *$P$ < 0.05, ***$P$ < 0.001. Two-tailed $P$ values were determined by an unpaired $t$-test. For all panels, data show mean ± s.e.m. See also Extended Data Fig. 4.
Credit: mouse, Servier under a Creative Common license CC BY 3.0; tumour, Freepik.com.

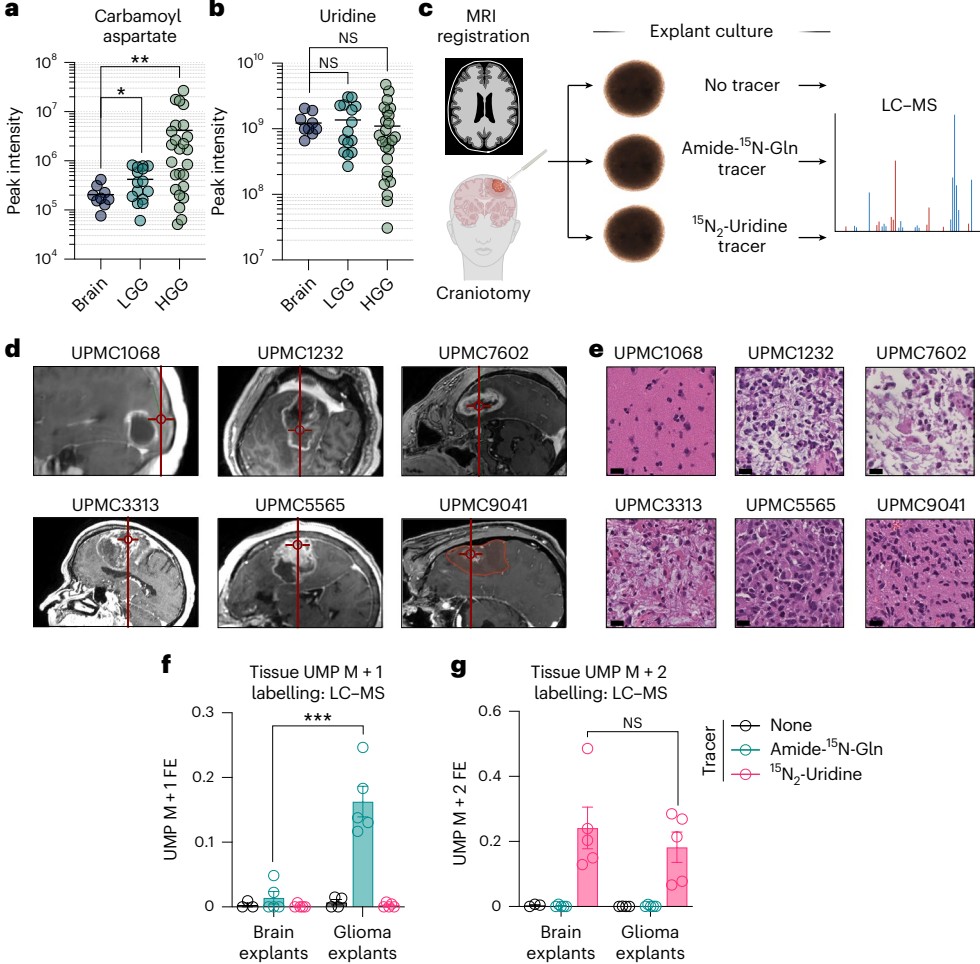

**Fig. 4 | Human gliomas display constitutive de novo pyrimidine synthesis ex vivo. a,b,** Peak intensity of carbamoyl aspartate (**a**) (*P* for brain-LGG = 0.0144, *P* for Brain-HGG = 0.0069) and uridine (**b**) in normal brain (*n* = 9), index resections of LGG (grades 2 and 3, *n* = 15), and index resections of HGG (grade 4, *n* = 26) from human surgical specimens. Data are from reanalysis of Abdullah et al.[24]. **c,** Schema depicting stable isotope tracing in organoid explant cultures. **d,** Representative intraoperative stereotactic neuronavigation of sample collection site from non-neoplastic neocortex (UPMC1068) or GBM (all others). **e,** H&E stains of tissues from **d**. Scale bars, 20 μm (*n* = 1 per sample). **f,g,** 18-h stable isotope tracing of explants with amide-$^{15}$N-glutamine (*n* = 5 per cohort), $^{15}$N$_2$-uridine (*n* = 5 per cohort) or no tracer (*n* = 3 for brain cohort, *n* = 4 for glioma cohort). UMP M + 1 (*p* = 0.0004) (**f**) and M + 2 FE (**g**) values are shown. \**P* < 0.05, \*\**P* < 0.01, \*\*\**P* < 0.001. Two-tailed *P* values were determined by an unpaired *t*-test. For all panels, data show mean ± s.e.m. See also Extended Data Fig. 5. Credit: face, Servier under a Creative Common license CC BY 3.0; tumour, Freepik.com; brain, DBCLS under a Creative Common license CC0 1.0.

increased plasma uridine, levels in brain tissue were not affected (Extended Data Fig. 4c-d). Therefore, our experimental approach did not artificially alter substrate availability.

Although glutamine tracer accumulation was similar in normal brain and glioma, robust glutamine-dependent labelling of UMP was only present in glioma (Fig. 3f), whereas uridine-dependent labelling of UMP was similar in normal brain and glioma (Fig. 3g). Glutamine labelled metabolites other than UMP in normal brain, including CDP-ethanolamine (Extended Data Fig. 4f-g). Labelling of the M + 1 isotopologue of CDP-ethanolamine occurs through two enzymatic activities: (1) synthesis of CTP from glutamine and UTP by CTP synthase, and (2) synthesis of CDP-ethanolamine from CTP and phosphoethanolamine by CTP-phosphoethanolamine cytidylyltransferase (Extended Data Fig. 4e,h,i). Evidence of glutamine metabolism by CTP synthase suggests that low UMP labelling by glutamine in normal brain is specific and cannot be explained by global repression of glutamine utilization.

We also performed spatial metabolomics analysis of brain parenchyma from tracer-infused naive or glioma-bearing mice using matrix-assisted laser desorption ionization mass spectrometry imaging

(MALDI−MSI). Normal brain tissue analysed by MALDI−MSI from naive mice and tumour-adjacent brain tissues from glioma-bearing mice also showed lower levels of de novo pyrimidine synthesis than glioma xenografts (Fig. 3h-i). Uridine-dependent pyrimidine salvage was again constitutively active in all tissues (Fig. 3h,j). Collectively, these data support a tight coupling of differentiation state to de novo pyrimidine synthesis regulation in vivo.

Fetal brain contains more undifferentiated cells (including NSCs, oligodendrocyte precursor cells and neuroblasts) than adult tissue. Consistent with this large population of primitive cells, foetal brain tissue recapitulated the elevated levels of de novo pyrimidine synthesis observed in glioma xenografts (Fig. 3k).

We next asked whether the pyrimidine synthesis pathway preferences we observed extended to primary human brain samples. We first analysed metabolite profiling from 50 human brain specimens[24]. The de novo pyrimidine synthesis pathway intermediate carbamoyl aspartate was increased in both high-grade gliomas (HGGs) and lower-grade gliomas (LGGs) relative to non-malignant brain tissue, consistent with pathway activation (Fig. 4a). In contrast, uridine did not differ between tissues (Fig. 4b). We previously developed an

approach to culture primary brain tissue explants using a formulation of HPLM that preserves viability and key features of resident cell populations[25]. Leveraging this technique, we assessed pyrimidine synthesis in human brain tissues using stable isotope tracing. We generated organoid explants from non-malignant brain and glioma tissues using stereotactic neurosurgical navigation to target specific brain regions. We then immediately cultured explants in HPLM containing amide-$^{15}$N-glutamine, $^{15}N_2$-uridine or no tracer (Fig. 4c). We collected two metastasis-adjacent non-malignant brain specimens and five glioma samples from patients who underwent surgical tumour resection (Fig. 4d and Supplementary Table 4). We also collected non-malignant brain parenchyma from three patients without oncologic disease who underwent ventriculoperitoneal shunt placement. After establishing explants, we stained remaining tissue with haematoxylin and eosin (H&E) and a board-certified neuropathologist (T.E.R.) confirmed the presence or absence of tumour cells (Fig. 4e).

Labelling in explants mirrored xenografts: glioma but not non-malignant brain explants demonstrated constitutive de novo pyrimidine synthesis, whereas all explants displayed robust uridine salvage (Fig. 4f-g). Like cell culture experiments (Extended Data Fig. 3a), glutamine tracer accumulation was higher in non-malignant brain than glioma explants, whereas uridine accumulation was similar (Extended Data Fig. 5a-b). These data suggest our assay may underestimate differences in de novo pathway activity between glioma and non-malignant brain due to larger unlabelled glutamine pools in the former. Together, these results show that de novo pyrimidine synthesis pathway activation is a conserved feature of primitive cell populations in primary brain tissues.

## De novo pyrimidine synthesis is repressed in a uridine-dependent manner in differentiated cells

As we pursued mechanisms by which preference for pyrimidine salvage is established in differentiated cells, we considered a seemingly paradoxical finding. Although NHA cells show a strong preference for pyrimidine salvage (Fig. 2l-m) and rapidly consume uridine relative to GSCs (Fig. 5a), they do not display proliferation defects during routine culture when uridine is depleted. To ask how pyrimidine synthesis is regulated under standard culture conditions, we collected conditioned medium and cells from NHA cultures grown near confluence (to induce substantial uridine depletion) during a 24-h period, replenishing medium every 6 h. We observed a reciprocal, cyclical relationship between intracellular carbamoyl aspartate and medium uridine levels (Fig. 5b). These metabolic shifts did not affect cell proliferation (Fig. 5b) and occurred without cyclical effects on UMP, UTP or other metabolites downstream of the convergence of de novo and salvage pathways (Fig. 5c and Extended Data Fig. 6a-c). We therefore hypothesized that differentiated cells repress de novo pyrimidine synthesis in response to uridine availability.

To test this idea, we evaluated de novo pyrimidine synthesis pathway activity under uridine-replete and depleted conditions. In NHA cells deprived of uridine, de novo pyrimidine synthesis intermediates rose sharply in a time-dependent manner (Fig. 5d and Extended Data Fig. 6d). We next preconditioned cells in HPLM with 3 µM uridine for 24 h before replacing the medium with one of two medium formulations: (1) HPLM with amide-$^{15}$N-glutamine and 3 µM uridine, or (2) HPLM with amide-$^{15}$N-glutamine and without uridine. We then assessed UMP labelling over 18 h (Fig. 5e-n). Immortalized astrocytes and fibroblasts rapidly activated de novo pyrimidine synthesis upon uridine withdrawal (Fig. 5j-l). Conversely, de novo pyrimidine synthesis in most GSCs and NSCs was unaffected by uridine deprivation (Fig. 5e-i,m-n). De novo pyrimidine synthesis did increase in TS516 GSCs and ENSA NSCs upon uridine deprivation, but these responses were more subtle than those of differentiated cell lines. We also evaluated isogenic fibroblasts, iPS cell-derived NPCs and NPC-derived neurons (Fig. 5o). Although all cultures activated de novo pyrimidine synthesis upon uridine withdrawal, differentiated fibroblasts

and neurons responded more robustly than primitive NPCs. Taken together, our data show that uridine availability strongly represses de novo pyrimidine synthesis in differentiated, but not primitive, cells (Extended Data Fig. 6e,f).

We sought to characterize the mechanistic link between uridine availability and de novo pyrimidine synthesis activity. We first considered whether low expression of de novo pyrimidine synthesis enzymes in differentiated cells may restrict pathway activity, as previous work has implicated expression as a driver of de novo pyrimidine synthesis in tumour cells[26]. Indeed, protein levels of the three enzymes that comprise the de novo synthesis pathway (CAD, DHODH and UMP synthase (UMPS)) are higher in human GBMs than non-malignant brain (Extended Data Fig. 6g-i). However, the salvage pathway enzyme UCK2 (one of two kinases that phosphorylate uridine to generate UMP) was also upregulated in GBMs (Extended Data Fig. 6j-l). We then evaluated RNA sequencing data[17,25] from our GSCs and NHA and found that NHA generally did not express de novo pathway enzymes at lower levels nor salvage pathway enzymes at higher levels than GSCs (Extended Data Fig. 6m). Therefore, basal differences in gene expression cannot solely explain de novo pyrimidine synthesis repression in differentiated cells under nutrient-rich conditions.

We then evaluated the uridine deprivation response in NHA cells at earlier time points (Extended Data Fig. 6n). Increases in carbamoyl aspartate labelling from amide-$^{15}$N-glutamine were evident within 3 h. Likewise, uridine replenishment inhibited de novo synthesis in uridine-starved differentiated cells within 30 min (Extended Data Fig. 6o). Given these rapid kinetics, we suspected a post-transcriptional mechanism. After validating specificity of CAD, DHODH and UMPS antibodies (Extended Data Fig. 7a), we found minimal changes in these proteins following short- (Fig. 5p) or long-term (Extended Data Fig. 7b-d) uridine starvation in NHA cells and BJ fibroblasts. Cultured human platelets, which lack nuclei and do not display transcription but do harbour de novo pyrimidine synthesis enzymes[27], activated de novo pyrimidine synthesis after uridine withdrawal (Fig. 5q). These data collectively indicate a post-transcriptional mechanism is responsible for uridine-dependent repression of de novo pyrimidine synthesis.

We considered whether changes in intracellular metabolite levels upon uridine withdrawal stimulate de novo pyrimidine synthesis. The only metabolites consistently altered in NHA cells following uridine withdrawal (Extended Data Fig. 6d) included uridine and de novo pyrimidine synthesis intermediates. De novo pathway substrate levels were not altered upon uridine withdrawal. We also evaluated metabolites involved in allosteric control of pyrimidine synthesis: UTP, which represses CAD[28–30], and uric acid, which inhibits UMPS[9]. At early time points (1 or 6 h, respectively) when uridine deprivation stimulated de novo pyrimidine synthesis (Extended Data Fig. 7e,h), we observed no differences in UTP (Extended Data Fig. 7f,i) or uric acid (Extended Data Fig. 7g,j).

Activating phosphorylation of CAD on S1859 downstream of the insulin–PI3K–mTORC1 signalling axis is an important post-translational mechanism of pyrimidine synthesis regulation[31,32]. To ask if uridine availability changes this modification, we first validated antibodies against S6 phospho-S240, S6K phospho-T389 (both downstream targets of mTORC1 signalling) and CAD phospho-S1859 (Extended Data Fig. 7k). Neither NHA nor BJ fibroblasts increased mTORC1 signalling markers during short- or long-term uridine starvation (Extended Data Fig. 7l-o). In the same conditions, CAD phospho-S1859 levels did not consistently increase (Extended Data Fig. 7p-s), suggesting that uridine governs de novo pyrimidine synthesis activity in differentiated cells independently of mTORC1 activation and CAD S1859 phosphorylation.

## CAD phosphorylation at S1900 activates de novo pyrimidine synthesis in differentiated cells

We next asked whether protein–protein interactions or post-translational modifications (PTMs) involving CAD were affected by

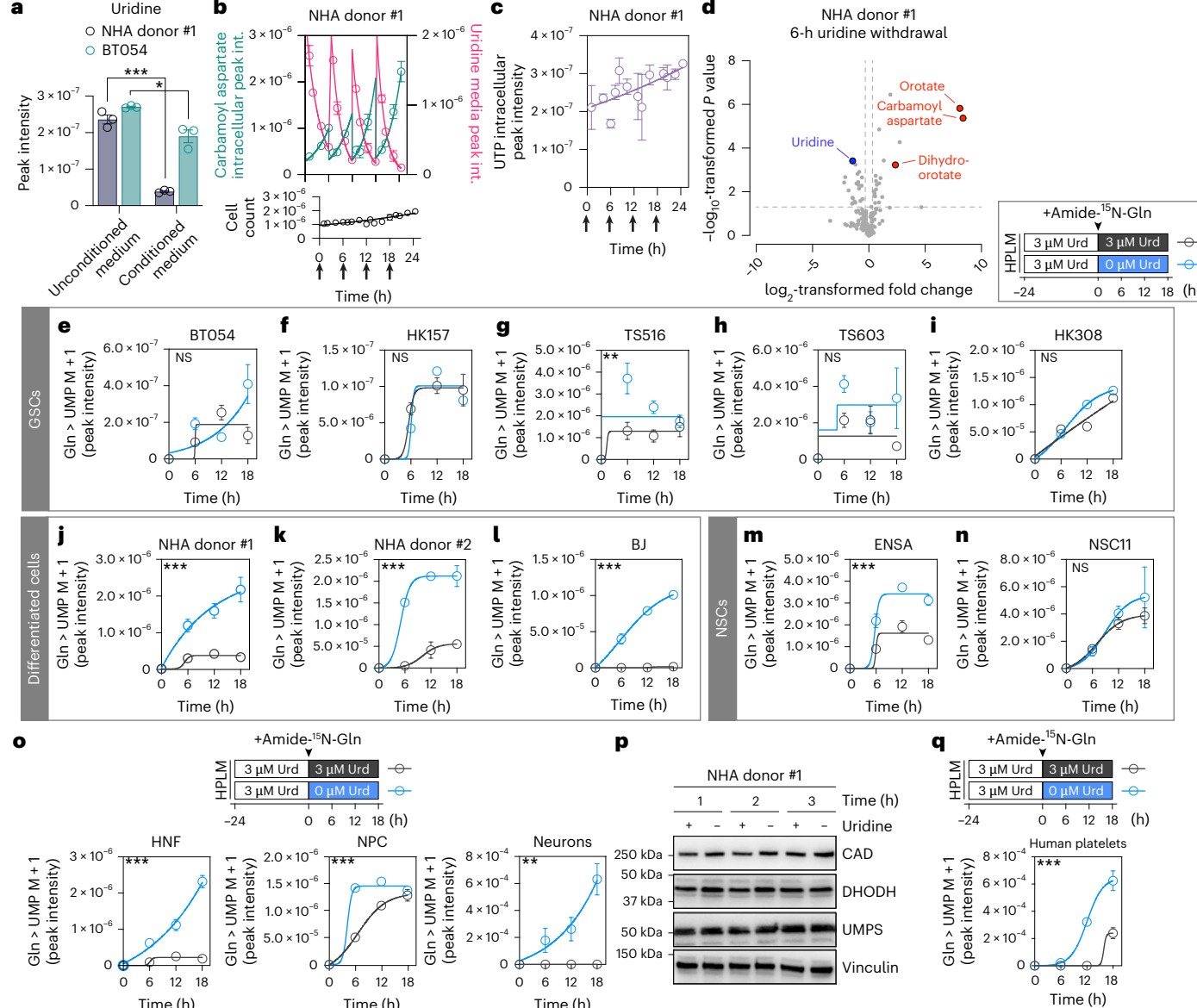

**Fig. 5 | Uridine deprivation acutely activates de novo pyrimidine synthesis in differentiated cells through a post-transcriptional mechanism. a**, Peak intensity of uridine in unconditioned and conditioned medium from 18-h cultures of NHA donor #1 or BT054 cells ($n = 3$ per group, $P$ for NHA < 0.0001, $P$ for BT054 = 0.0429). **b,c**, Quantification of intracellular carbamoyl aspartate and medium uridine (**b**, top), cell count (**b**, bottom) and intracellular UTP (**c**) in NHA donor #1 cells cultured in DMEM with 3 µM uridine. Medium changes are indicated by arrows. Curves were fit by nonlinear regression for a 6-h window (**b**) after each medium change or full 24-h experiment (**c**) ($n = 3$). Int., intensity. **d**, Volcano plot of metabolites in NHA donor #1 cells cultured in 0 µM or 3 µM uridine for 6 h after 24 h of culture with 3 µM uridine ($n = 6$ per group). Red, de novo pyrimidine synthesis intermediates; blue, pyrimidine salvage substrate uridine. **e–n**, Amide-[15]N-glutamine (Gln) stable isotope tracing in GSC (**e–i**), differentiated (**j–l**) or NSC (**m,n**) lines cultured in 0 µM or 3 µM uridine after

24 h of culture with 3 µM uridine ($n = 3$ for all except HK308, for which $n = 2$, $P$ for TS516 = 0.0057, $P$ for NHA donor #1 = 0.0001, $P$ for NHA donor #2 < 0.0001, $P$ for BJ < 0.0001, $P$ for ENSA < 0.0001). **o**, Amide-[15]N-glutamine stable isotope tracing in HNF, NPC or NPC-derived neurons cultured in 0 µM or 3 µM uridine after 24 h of culture with 3 µM uridine ($n = 3$ per group, $P$ for HNF < 0.0001, $P$ for NPC < 0.0001, $P$ for neurons = 0.0033). **p**, Representative immunoblots for CAD, DHODH and UMPS in NHA donor #1 cells cultured in 0 µM or 3 µM uridine after 24 h of culture with 3 µM uridine ($n = 3$). **q**, Amide-[15]N-glutamine stable isotope tracing in human donor platelets cultured in 0 µM or 3 µM uridine after 24 h of culture with 3 µM uridine ($n = 3$, $P < 0.0001$). *$P < 0.05$, **$P < 0.01$, ***$P < 0.001$. Two-tailed $P$ values were determined by an unpaired $t$-test. $t$-tests in **e–o,q** compare area under the curve values for 0 µM uridine and 3 µM uridine conditions. For all panels, data show mean ± s.e.m. See also Extended Data Figs. 6 and 7.

uridine availability. We engineered NHA cells to express Flag-tagged CAD, cultured these cells in the presence or absence of physiological uridine content for 3 h, and performed immunoprecipitation (IP)–mass spectrometry (MS) (Fig. 6a and Extended Data Fig. 8a). Among the PTMs measured on CAD and interacting proteins, only one modification was significantly elevated in uridine-deprived cells: phosphorylation of CAD at S1900 (Fig. 6b,c, Extended Data Fig. 8b,c and Supplementary Table 5). Although CAD S1900 phosphorylation has

been captured in previous proteomic profiling studies[31] and increases following Kaposi's sarcoma-associated herpesvirus (KSHV) infection of human cells[33], the endogenous function of this modification is not clear.

We asked whether CAD S1900 phosphorylation could overcome the inhibitory effect of uridine availability on de novo pyrimidine synthesis in differentiated cells (Fig. 5j-l). We used NHA cells expressing the following constructs: wild-type (WT) CAD, a CAD S1900 phosphomimetic mutant (S1900D), a CAD mutant unable to be phosphorylated

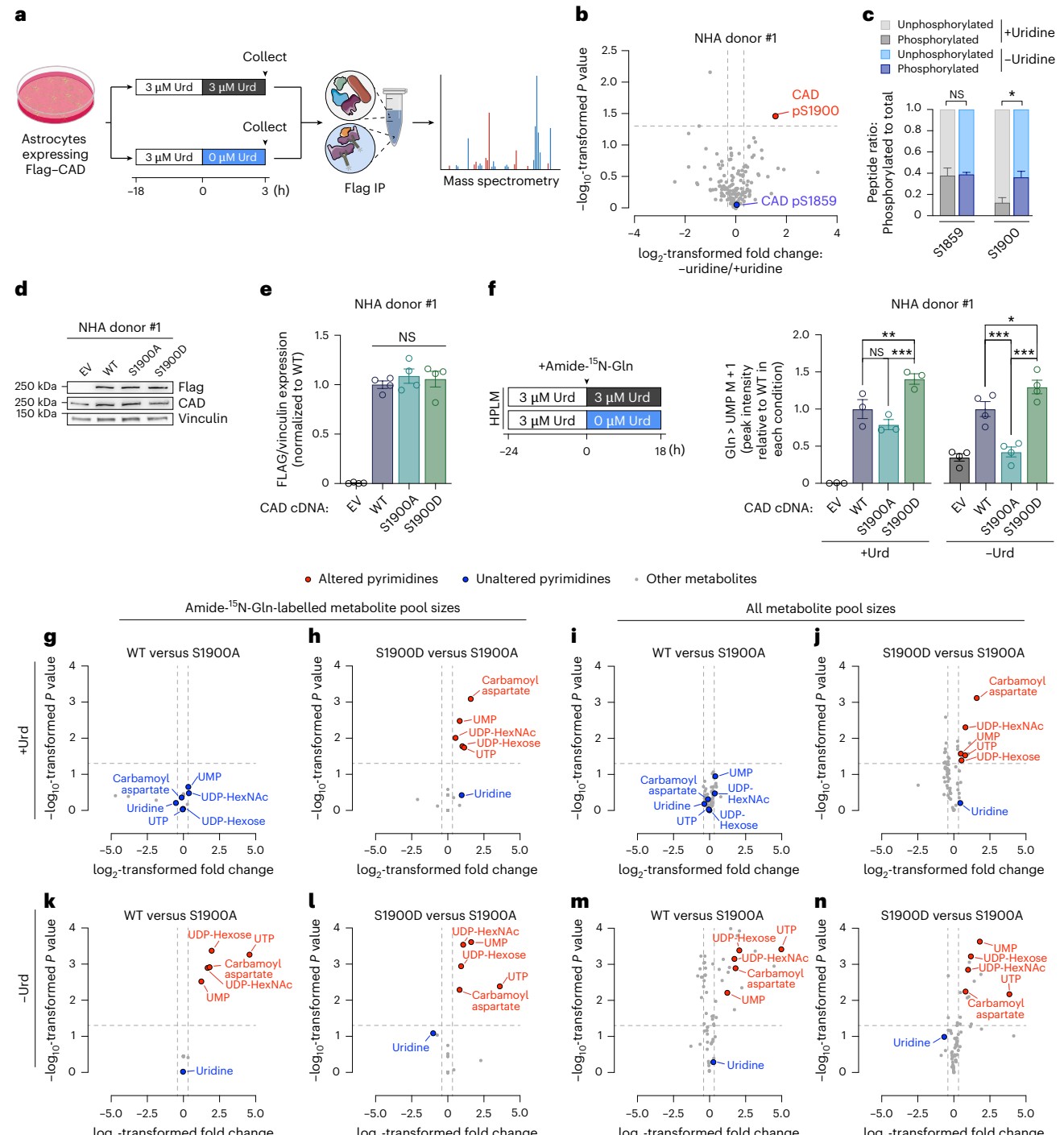

**Fig. 6 | CAD phosphorylation at S1900 activates de novo pyrimidine synthesis in differentiated cells. a**, Schema depicting Flag IP–MS of Flag-tagged CAD and co-immunoprecipitating proteins. **b**, Volcano plot of PTMs on all detected proteins in Flag IP of NHA donor #1 cells expressing Flag-tagged CAD cultured in DMEM with 0 µM or 3 µM uridine for 3 h after 18 h of culture in DMEM with 3 µM uridine (n = 3 per group). Blue, PTM that did not differ between the two conditions; red, enriched PTM. **c**, Ratio of phosphorylated to total peptide abundance for CAD phosphorylation sites in CAD–Flag IP (n = 3 per group, P for S1900 = 0.0348). **d**, Immunoblot for Flag and CAD in NHA donor #1 cells expressing EV, WT CAD, CAD S1900A or CAD S1900D (n = 4 per line). **e**, Densitometry quantifying relative Flag expression in NHA donor #1 cells expressing either EV, WT CAD, CAD S1900A or CAD S1900D (n = 4 per line). Data are normalized to WT CAD. **f**, Amide-15N-glutamine stable isotope tracing for 18 h in 3 µM or 0 µM uridine in NHA donor #1 cells expressing either EV, WT CAD, CAD S1900A or CAD S1900D (n = 3 per group for 3 µM uridine and n = 4 per group for

0 µM uridine, P for WT-S1900D in +Urd = 0.0038, P for S1900A-S1900D in +Urd < 0.0001, P for WT-S1900A in −Urd < 0.0001, P for WT-S1900D in −Urd = 0.0118, P for S1900A-S1900D in −Urd < 0.0001. Data are normalized to WT CAD in each condition. **g–n**, Volcano plot of amide-15N-glutamine (Gln)-labelled (**g,h,k,l**) or total metabolite levels (**i,j,m,n**) in NHA donor #1 cells expressing WT CAD (**g,k,i,m**) or S1900D CAD (**h,j,l,n**) versus S1900A CAD in 3 µM (**g–j**) or 0 µM (**k–n**) uridine (n = 3 per group for 3 µM uridine and n = 4 per group for 0 µM uridine). Blue, pyrimidines that did not differ between the two conditions; red, pyrimidines that differ between the two conditions. UDP-HexNAc, uridine diphosphate-N-acetylhexosamine. *P < 0.05, **P < 0.01, ***P < 0.001. In **b,c,f–n**, two-tailed P values were determined by unpaired t-test; in **e**, P value was determined by one-way ANOVA. For all panels, data show mean ± s.e.m. See also Extended Data Fig. 8. Credit: plates and astrocytes, NIAID; tube, Hellicase 11 under a Creative Commons license CC BY 4.0; proteins and antibody, Servier under a Creative Common license CC BY 3.0.

at S1900 (S1900A) or an empty vector (EV) (Fig. 6d-e). Tracing with amide-[15]N-glutamine under physiological uridine or in uridine-free conditions revealed that, as expected, both WT CAD expression and uridine deprivation independently increased labelling of UMP by amide-[15]N-glutamine (Fig. 6f and Extended Data Fig. 8d). CAD S1900D-expressing cells displayed higher de novo pyrimidine synthesis activity than WT CAD-expressing cells under both uridine-replete and uridine-free conditions. We validated sufficiency of the CAD S1900D phosphomimetic mutant to overcome uridine-dependent repression of de novo pyrimidine synthesis in G9C cells, a line derived from Chinese hamster ovary (CHO) cells that lacks endogenous CAD expression[34] (Extended Data Fig. 8e-g). NHA cells expressing CAD S1900A displayed similar de novo pyrimidine synthesis to WT CAD-expressing cells in uridine-replete conditions, but they failed to activate the de novo pyrimidine synthesis pathway beyond that of EV-expressing cells in response to uridine withdrawal (Fig. 6f).

To complement the analysis of UMP levels, we performed global metabolomics analysis of NHA cells engineered to express WT CAD, CAD S1900D or CAD S1900A (Fig. 6g-n). In the presence of uridine, both amide-[15]N-glutamine-labelled and total pyrimidine levels did not differ between WT CAD- and CAD S1900A-expressing cells (Fig. 6g,i), consistent with low CAD S1900 phosphorylation in NHA cells under these conditions (Fig. 6c). However, CAD S1900A-expressing cells displayed defects in synthesis of both labelled and total pyrimidine pools relative to WT CAD-expressing cells during uridine starvation (Fig. 6k,m). Conversely, CAD S1900D-expressing cells displayed increased labelled and total pyrimidine pool sizes independent of uridine availability (Fig. 6h,j,l,n). Taken together, these data suggest that CAD S1900 phosphorylation is sufficient to overcome differentiated cells' strict preference for pyrimidine salvage and necessary for induction of de novo pyrimidine synthesis in response to uridine withdrawal.

## CAD S1900 phosphorylation is increased in GBM and correlates with primitive cell states

We next asked whether CAD S1900 phosphorylation is associated with basal differences in de novo pyrimidine synthesis activity between tissues with large or small constituent primitive cell populations. We assessed proteomic and phosphoproteomic analyses of human GBM and non-malignant brain tissues[35] to evaluate CAD S1900 phosphorylation in tissues composed of varying content of differentiated cells. First, we used linear regression to evaluate correlations between primitive cell markers (SOX2 and OLIG2) and phosphorylation marks on pyrimidine synthesis enzymes in GBM tissues (Fig. 7a-b). Two modifications positively correlated with primitive cell markers in these tissues: phosphorylation of S61 on UPP1 and S1900 on CAD. To contextualize CAD S1900 phosphorylation, we used AlphaFold[36] to evaluate the S1900 residue (Fig. 7c). CAD S1900 is located near S1859, the established target of activating phosphorylation downstream of mTORC1 signalling[31,32], on a large, unstructured, interdomain loop. We next validated peptides representing established phosphorylation sites on CAD (T456, S1406 and S1859) and S1900 in human GBM and non-malignant brain by MS/MS (Extended Data Fig. 9a–c). GBMs displayed elevated CAD

phosphorylation at S1406, S1859 and S1900 relative to non-malignant brain, whereas T456 phosphorylation was undetectable in both tissues (Fig. 7d-g). These data suggested a link between primitive cell composition (Fig. 7a-b), CAD S1900 phosphorylation (Fig. 7g) and basal rates of de novo pyrimidine synthesis (Fig. 4f) in human brain tissues. Therefore, we evaluated correlations between signalling or primitive neural markers and CAD phosphorylation (Fig. 7h and Extended Data Fig. 9d-u). As expected, CAD S1406 and S1859 phosphopeptide levels correlated with markers of PKA (phosphorylation of GSK3β at S9 (ref. 37); Fig. 7h and Extended Data Fig. 9d,j) and mTORC1 (refs. 31,32) (phosphorylation of ribosomal protein S6 at S235/S236; Fig. 7h and Extended Data Fig. 9e,k) signalling, respectively. Despite the role of MAPK signalling in modulation of CAD T456 (but not S1406 or S1859) phosphorylation[38], both S1406 and S1859 phosphopeptide levels positively correlated with ERK phosphorylation (Extended Data Fig. 9h-i,n-o). These findings suggest either crosstalk or correlation between PKA, mTORC1 and MAPK signalling in GBM. CAD S1900 phosphorylation, however, positively correlated with both SOX2 and OLIG2 (Fig. 7h and Extended Data Fig. 9r-s) but not with markers of PKA, mTORC1 or MAPK signalling (Fig. 7h and Extended Data Fig. 9p-q,t-u). Moreover, CAD phosphorylation at S1406 and S1859 did not positively associate with primitive cell markers (Fig. 7h and Extended Data Fig. 9f-g,l-m). These data suggest two independent regulatory axes converge on CAD in GBM: growth factor signalling-dependent phosphorylation of CAD at S1406 and S1859 and cell state-specific phosphorylation of CAD at S1900.

We then asked whether the cancer relevance of this modification extends beyond GBM. Using a study reporting a proteomic-based stemness index (PROTsi) to quantify primitive cell features in a panel of human tumours[39], we detected a positive correlation between stemness and CAD S1900 phosphopeptide in GBM, pancreatic ductal adenocarcinoma (PDAC) and head and neck squamous cell carcinoma (HNSCC) (Fig. 7i). These data indicate that the association between CAD S1900 phosphorylation and primitive cell states is generalizable across human tumour subtypes.

To assess the relevance of CAD phosphorylation to physiology, we measured Cad phosphopeptides in adult mouse tissues. Phosphorylation of the orthologous site for T456 was not detected in any tissue, whereas phosphorylation of the orthologous sites for S1406 and S1859 in murine Cad was observed in nearly all tissues (Extended Data Fig. 9v). In contrast, the orthologous site of S1900 displayed a tissue-specific pattern of phosphorylation overrepresented for sex organs (including testes, epididymis and ovary), suggesting a link with undifferentiated gametes. Although Cad phosphorylation patterns differed, pyrimidine synthetic enzyme levels were largely similar across tissues (Extended Data Fig. 9w).

To assess whether CAD phosphorylation on S1900 is evolutionarily conserved, we evaluated CAD amino acid sequences across all trifunctional CAD orthologues in UniProt[40]. CAD S1900 and surrounding amino acid residues are strongly conserved across mammalian species, with weaker conservation in non-mammals (Fig. 7j-m). Therefore, CAD S1900 phosphorylation may constitute a molecular link between cell state and pyrimidine synthesis activity that has persisted

**Fig. 7 | The conserved CAD S1900 residue is constitutively phosphorylated in primitive cell populations. a,b**, Volcano plot of Pearson's *r* and *P* values for correlations between phosphorylation modifications of pyrimidine synthesis enzymes and SOX2 (**a**) or OLIG2 (**b**) protein expression in human GBMs (*n* = 99). **c**, AlphaFold predicted structure of CAD. GATase, glutamine amidotransferase; CPSII, carbamoyl phosphate synthetase II; ATCase, aspartate carbamoyltransferase; DHOase, dihydroorotase. **d–g**, Relative abundance of T456 (**d**), S1406 (*P* < 0.0001) (**e**), S1859 (*P* < 0.0001) (**f**) and S1900 (*P* = 0.0002) (**g**) phosphorylation modifications of CAD in human brain and GBM. Phosphorylation site abundance is normalized (norm.) to total CAD protein and expressed relative to brain. ND, not detected. (*n* = 10 brain, *n* = 99 GBM). **h**, Correlation between CAD phosphorylation sites and signalling programme or

stemness markers in human GBM. Phosphorylation site abundance is normalized to total protein. (*n* = 99). Panels **a,b,d–h** are reanalyses of raw data published in ref. 35. **i**, Correlation between PROTsi and CAD S1900 phosphorylation in human GBM, PDAC and HNSCC tissues. Data are from reanalysis of proteomics dataset published in ref. 39. **j–m**, Sequence alignment relative to human CAD S1900. Sequences from selected species (**j**). Sequence logos of all species (**k**), mammals (**l**) or non-mammals (**m**). Amino acids in blue are hydrophilic, green are neutral and black are hydrophobic. **n**, Schema of proposed model of pyrimidine synthesis pathway choice in differentiated and primitive cells. *\*P* < 0.05, \*\**P* < 0.01, \*\*\**P* < 0.001. Two-tailed *P* values were determined by linear regression analysis (**a,b,h,i**) or unpaired *t*-test (**d–g**). For all panels, data show mean ± s.e.m. See also Extended Data Fig. 9.

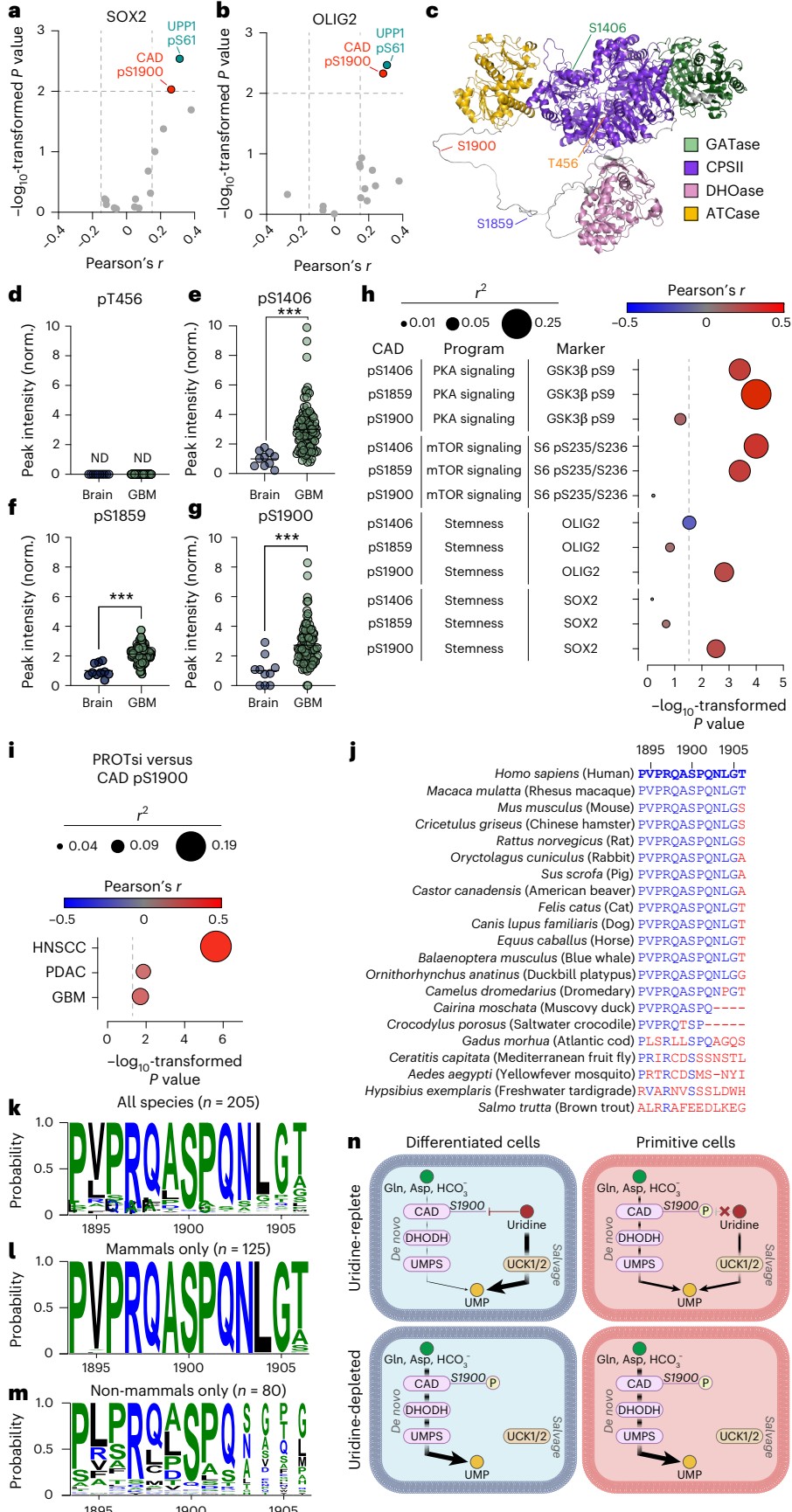

through mammalian evolution. The Kinase Library, a prediction tool for kinase substrate specificity[41], predicted a strong signature for involvement of AMPK family kinases in phosphorylation of this site (Supplementary Table 6). Based on our findings, we propose a model of pyrimidine synthesis control governed by cellular differentiation state (Fig. 7n). In differentiated cells exposed to sufficient nutrients, both CAD S1900 phosphorylation and de novo pyrimidine synthesis are repressed, leading to salvage pathway predominance. During uridine deprivation, however, CAD is rapidly phosphorylated at S1900, stimulating compensatory pyrimidine synthesis via the de novo pathway. In primitive cells, both de novo and salvage pathways for pyrimidine synthesis are constitutively active, with the former uncoupled from regulatory control by uridine. This metabolic trait is associated with basal enrichment of CAD S1900 phosphorylation, which is strongly linked to primitive cell markers rather than uridine availability in this cellular context.

## Discussion

Nitrogen metabolic pathways operate in parallel with carbon metabolism[6]. Amino acids, nucleotide precursors, ions, and other nitrogenous small molecules serve as substrates for enzymes that sustain macromolecule synthesis, energy production, stress resistance and cell growth. System-level understanding of how nitrogen metabolism pathways operate and intersect in human cells, however, has been limited by a lack of appropriate tools. Although systems biology approaches have revealed global nitrogen metabolism programmes in bacteria[42,43], these studies exploit unique bacterial dependence on ammonium for nitrogen assimilation and are difficult to apply to human cells that use a wide range of nitrogenous nutrients. Here we develop, validate and apply an experimental platform that addresses complexities of studying nitrogen metabolism in cells from higher organisms. This platform generates both high-level representations of nitrogen metabolic programmes and quantitative measurements of differences in substrate utilization between paired cultures. To facilitate adoption of this platform, we describe how to prepare the HPLM library and provide access to a computational pipeline for analysis.

We used the platform to ask how nitrogen metabolism differs between differentiated, non-transformed astrocytes and primitive, malignant glioma cells. Pyrimidine synthesis substrate utilization was a key difference; glioma cells preferentially used glutamine to fuel de novo pyrimidine synthesis, whereas immortalized astrocytes nearly exclusively salvaged pyrimidines from uridine. Aberrant activation of de novo pyrimidine synthesis has been reported in many cancer contexts[4,11,21,44–49], but the underlying mechanisms are not fully understood. Previous work has also revealed differences in pyrimidine synthesis pathway preferences among healthy tissues[50–52]. Of note, these reports describe elevated de novo pyrimidine synthesis in the intestine, raising the possibility that the prominent stem cell compartment in this organ may contribute to this metabolic activity. Studies of cancerous and healthy tissues provide evidence of heterogeneity in pyrimidine nucleotide metabolism, thereby establishing a rationale to decipher the molecular and cellular determinants of pyrimidine synthesis pathway choice.

Our data show that pyrimidine synthesis pathway choice is cell-intrinsic, independent of microenvironmental nutrient limitation, and strongly linked to cellular differentiation state. Our study complements previous work highlighting expression as a factor contributing to pyrimidine metabolism programmes in tumours[26]. Indeed, several oncogenes and tumour suppressors directly regulate transcription of genes encoding pyrimidine synthesis enzymes (for example, c-myc-dependent transactivation of *CAD*[53]). Under basal conditions, however, non-malignant, adult brain tissues and cultured astrocytes, neurons and fibroblasts repress de novo pyrimidine synthesis despite constitutive expression of de novo

pathway enzymes. These findings inform a model in which de novo pyrimidine synthesis machinery is poised for activation in differentiated cells but requires molecular cues to initialize. This regulatory framework may provide an advantage to cells during acute nutrient deprivation by allowing them to upregulate de novo pyrimidine synthesis more quickly than engaging transcription and translation would allow.

Our work reveals uridine depletion as an important cue for differentiated cells to activate de novo pyrimidine synthesis. Mechanistically, we found that astrocytes rapidly phosphorylate CAD at S1900 upon uridine starvation. We did not observe other PTMs accumulate on CAD within this time. Blocking CAD S1900 phosphorylation did not affect pyrimidine metabolism under basal, uridine-replete conditions but caused depletion of both de novo synthesis-derived and total pyrimidine nucleotide pools during uridine deprivation. Therefore, we conclude that CAD S1900 phosphorylation is necessary for astrocytes to activate de novo pyrimidine synthesis when uridine is limiting and propose that CAD S1900 phosphorylation is a key component of differentiated cells' response to pyrimidine salvage disruption. Our findings suggest that cells monitor and respond to changes in pyrimidine abundance in a manner that may cooperate with purine sensing mechanisms[54,55] to promote adaptation to nucleotide depletion and metabolic stress.

These insights deepen our understanding of allosteric and post-translational mechanisms that govern de novo pyrimidine synthesis. UTP and uric acid directly bind and inhibit CAD[28–30] and UMPS[9], respectively. Negative feedback on CAD by UTP balances rates of pyrimidine metabolism throughout the cell cycle[30]. Signalling from MAPK[38,56], PKA[37,56] and mTORC1 (refs. [31,32]) pathways converge on CAD, inducing activating phosphorylation events at T456, S1406 and S1859, respectively. Insulin signalling stimulates mTORC1-dependent CAD phosphorylation at S1859 and links systemic nutrient availability with cellular nucleotide metabolism[31,32]. Our data reveal CAD phosphorylation at S1900 as an additional point of endogenous de novo pyrimidine synthesis pathway control. CAD S1900 phosphorylation was previously detected in a study of insulin-dependent CAD S1859 phosphorylation, although it was not regulated by insulin[31], and was recently shown to contribute to de novo pyrimidine synthesis activation in KSHV-infected cells[33], but the relevance of this modification outside the context of viral infection is not clear.

We found that human GBM displayed higher CAD S1900 phosphorylation relative to non-malignant brain, and that these levels positively correlated with markers of primitive cell populations within GBMs. Mimicking CAD S1900 phosphorylation was sufficient to stimulate de novo pyrimidine synthesis in astrocytes in the presence of uridine, thereby phenocopying basal pathway activation in primitive cells. Collectively, our findings suggest that under nutrient-replete conditions, CAD S1900 phosphorylation is a cell state-specific mark that contributes to the de novo pyrimidine synthesis pathway preference of primitive cells. Additional work is required to understand how this modification affects flux through CAD. Previous research indicates that CAD S1859 phosphorylation promotes oligomerization-dependent substrate channelling[32]. Because S1900 resides nearby S1859 on a large, unstructured, interdomain loop within CAD, this mechanism may also be relevant to S1900 phosphorylation.

Selective CAD S1900 phosphorylation and activation of de novo pyrimidine synthesis in discrete cell populations may contribute to therapeutic indices of drugs targeting this pathway, including DHODH inhibitors such as teriflunomide and leflunomide (US Food and Drug Administration (FDA)-approved to treat multiple sclerosis and rheumatoid arthritis, respectively)[57]. Agents targeting DHODH are being investigated to treat various cancers and autoimmune disorders[58–62]. In these studies, CAD S1900 phosphorylation may represent a useful biomarker of basal differences in de novo pyrimidine synthesis activity in healthy and diseased tissues.

## Methods

### Cell lines

NHA donor #1 cells[19] (human astrocytes immortalized with HPV E6 and E7 and hTERT, sex unknown) were obtained from R. Pieper at the University of California, San Francisco. NHA donor #2 cells were generated[11] from commercially obtained primary human astrocytes (Lonza CC-3187). HEK293T (female, ATCC CRL-3216, RRID: CVCL_0063), BJ (male, ATCC CRL-2522, RRID: CVCL_3653), HNF primary fibroblast (Lonza NHDF-Neo) and HeLa (female, Millipore Sigma 93021013, RRID: CVCL_0030) cells were obtained commercially. G9C CHO cells[34] were a gift from R. Possemato at New York University. BT054[16] (female, RRID: CVCL_N707) cells were obtained from S. Weiss at the University of Calgary. HK157 (female) and HK308 (ref. 63) (male) cells were obtained from H. Kornblum at the University of California, Los Angeles. TS516 (sex unknown, RRID: CVCL_A5HY) and TS603 (ref. 15) (sex unknown, RRID: CVCL_A5HW) cells were obtained from I. Mellinghoff at Memorial Sloan-Kettering Cancer Center. ENSA (sex unknown) and NSC11 (ref. 64) (sex unknown) cells were obtained from J. Rich at University of North Carolina-Chapel Hill. MGG152 (ref. 23) (male) cells were obtained from D. Cahill at Massachusetts General Hospital. Murine GSC lines PC948-1 and PIC144-1 were derived from autochthonous astrocytomas that formed in a modified version of a genetically engineered mouse model[11,17]. A portion of murine brain tumour tissue was dissected and dissociated to a single-cell suspension using the Neural Tissue Dissociation kit (Miltenyi Biotec 130-092-628, RRID: SCR_020293) by following the manufacturer's instructions and then cultured for selection of neurosphere-forming cells.

All cell lines were routinely evaluated for mycoplasma contamination with the e-Myco *Mycoplasma* PCR Detection kit (Bulldog Bio 2523348), e-Myco PLUS *Mycoplasma* PCR Detection kit (Bulldog Bio 25233) or MycoAlert *Mycoplasma* Detection Kit (Lonza LT07-318) and confirmed to be negative. Cell line authentication was not performed because reference short-term tandem-repeat profiles have not been established for these lines. Sex and source of each line is stated above and listed as unknown if unreported in the original publication describing its derivation.

### Cell culture

PC948-1 and PIC144-1 cells were cultured in NeuroCult Basal Medium (Mouse and Rat) with Proliferation Supplement (STEMCELL Technologies, 05701), supplemented with EGF (20 ng ml⁻¹, GoldBio, 1150-04), bFGF (20 ng ml⁻¹, GoldBio, 1140-02), heparin (2 μg ml⁻¹, STEMCELL Technologies, 07980), penicillin–streptomycin (100 U ml⁻¹ and 100 μg ml⁻¹, respectively, Thermo Fisher, 15140148), amphotericin B (250 ng ml⁻¹, GeminiBio, 400104) and Plasmocin (250 ng ml⁻¹, InvivoGen, ant-mpp) on ultra-low adherence plates (six-well plates, Corning, 3471, 10-cm dishes, Corning 4615) in 5% CO₂ and 5% oxygen at 37 °C.

BT054, HK157, HK308, TS516 and TS603 cells were cultured in NeuroCult NS-A Basal Medium (Human) with Proliferation Supplement (STEMCELL Technologies, 05751), supplemented with EGF (20 ng ml⁻¹, GoldBio, 1150-04), bFGF (20 ng ml⁻¹, GoldBio, 1140-02), heparin (2 μg ml⁻¹, STEMCELL Technologies, 07980), penicillin–streptomycin (100 U ml⁻¹ and 100 μg ml⁻¹, respectively, Thermo Fisher, 15140148), amphotericin B (250 ng ml⁻¹, GeminiBio, 400104) and Plasmocin (250 ng ml⁻¹, InvivoGen, ant-mpp) on ultra-low adherence plates (six-well plates, Corning, 3471, 10-cm dishes, Corning 4615) in 5% CO₂ and at ambient oxygen at 37 °C.

NHA donor #1, NHA donor #2, HNF and BJ cells were cultured in DMEM (Thermo Fisher, 11995-065) supplemented with fetal bovine serum (FBS; 10%, GeminiBio, 100-106) and penicillin–streptomycin (100 U ml⁻¹ and 100 μg ml⁻¹, respectively, Thermo Fisher, 15140148) in 5% CO₂ and at ambient oxygen at 37 °C.

ENSA, NSC11 and MGG152 cells were cultured in Neurobasal Medium (Thermo Fisher, 21103049) supplemented with glutamine (3 mM, Thermo Fisher, 25030081), B27 supplement (1×, Thermo Fisher, 17504044), N2 supplement (0.25×, Thermo Fisher, 17502048), EGF (20 ng ml⁻¹, GoldBio, 1150-04), bFGF (20 ng ml⁻¹, GoldBio, 1140-02), heparin (2 μg ml⁻¹, STEMCELL Technologies, 07980), penicillin–streptomycin (50 U ml⁻¹ and 50 μg ml⁻¹, respectively, Thermo Fisher, 15-140-148), amphotericin B (125 ng ml⁻¹, GeminiBio, 400104) and Plasmocin (0.25 μg ml⁻¹, InvivoGen, ant-mpp) on ultra-low adherence plates (six-well plates, Corning, 3471, 10-cm dishes, Corning, 4615) in 5% CO₂ and at ambient oxygen at 37 °C.

HNF-derived NPCs were cultured in DMEM/F12 (1:1) (Thermo Fisher, 11320033) supplemented with sodium pyruvate (1 mM, Thermo Fisher, 11360070), B27 supplement minus vitamin A (1×, Thermo Fisher, 12587010), N2 supplement (1×, Thermo Fisher, 17502048), laminin (1 μg ml⁻¹, Thermo Fisher, OB1415-01) and bFGF (20 ng ml⁻¹, GoldBio, 1140-02).

NPC-derived neurons were cultured in DMEM/F12 (1:1) (Thermo Fisher, 11320033) supplemented with sodium pyruvate (1 mM, Thermo Fisher, 11360070), B27 supplement minus vitamin A (1×, Thermo Fisher, 12587010), N2 supplement (1×, Thermo Fisher, 17502048), laminin (1 μg ml⁻¹, Thermo Fisher, OB1415-01) and FBS (10%, GeminiBio, 100-106).

G9C CHO cells were cultured in RPMI 1640 medium (Thermo Fisher, 11875119) supplemented with FBS (10%, GeminiBio, 100-106), penicillin–streptomycin (100 U ml⁻¹ and 100 μg ml⁻¹, respectively, Thermo Fisher, 15140148) and uridine (100 μM, Millipore Sigma, U3003) in 5% CO₂ and at ambient oxygen at 37 °C.

All GSC and NSC lines were maintained as neurospheres and dissociated 1–2 times per week with Accutase (STEMCELL Technologies, 07922). Only low-passage GSC lines were used, and these cells were discarded after 3 months in culture to prevent genetic and/or phenotypic drift.

### Generation of HNF-derived NPCs and neurons

The iPS cells were generated from HNF fibroblasts using the CytoTune-iPS 2.0 Sendai Reprogramming kit (Thermo Fisher Scientific, A16517) per the manufacturer's instructions. HNF cells were cultured in their native medium until 30–60% confluent, then transduced at a multiplicity of infection of 5:5:3 (CytoTune 2.0 KOS, CytoTune 2.0 hc-Myc, CytoTune 2.0 hKlf4) with non-integrating Sendai virus to deliver the Yamanaka factors KLF4, OCT4, SOX2 and C-MYC. After 24 h, medium was refreshed, and cells were cultured for 6 more days before being passaged on to irradiated MEFs. After day 7, cells were cultured in DMEM/F12 (1:1) (Thermo Fisher, 11320033) supplemented with KnockOut Serum Replacement (20%, Thermo Fisher, 10828028), non-essential amino acids (1%, Thermo Fisher, 11140050), penicillin–streptomycin (100 U ml⁻¹ and 100 μg ml⁻¹, respectively, Thermo Fisher, 15140148), bFGF (10 ng ml⁻¹, GoldBio, 1140-02) and 2-mercaptoethanol (0.1 mM, Thermo Fisher, BP176100) with daily medium changes. iPS cell colonies emerged within 12–21 days. Eight clonal iPS cell lines were derived and two were expanded and characterized. Pluripotency was confirmed by alkaline phosphatase staining and immunofluorescence for OCT4, SOX2, NANOG, TRA-1-60 and TRA-1-81. Clearance of Sendai virus was verified by RT–PCR at passage 10. Only virus-free clones were used for subsequent differentiation. To generate NPCs, iPS cells were cultured in mTeSR 1 (STEMCELL Technologies, 85850) on plates coated with Matrigel (Corning 354234) diluted 1:100 in phosphate-buffered saline. Cells were differentiated to NPCs by transition to DMEM/F12 (1:1) (Thermo Fisher, 11320033) supplemented with B27 supplement minus vitamin A (1×, Thermo Fisher, 12587010), N2 supplement (1×, Thermo Fisher, 17502048), LDN193189 (0.1 μM, Stemgent, 04-0074) and SB431542 (10 μM, Stemgent, 04-0010). Neural rosettes appeared by days 7–10 and were manually isolated on day 14. Rosettes were transferred to plates coated with 10 μg ml⁻¹ poly-L-ornithine (Millipore Sigma P3655) in water and 5 μg ml⁻¹ laminin (Thermo Fisher, 23017015) in phosphate-buffered saline and cultured in DMEM/F12 (1:1) (Thermo Fisher, 11320033) supplemented with sodium pyruvate (1 mM, Thermo

Fisher, 11360070), B27 supplement minus vitamin A (1×, Thermo Fisher, 12587010), N2 supplement (1×, Thermo Fisher, 17502048), laminin (1 µg ml⁻¹, Thermo Fisher, OB1415-01) and bFGF (20 ng ml⁻¹, GoldBio, 1140-02). NPC identity was confirmed by immunostaining for NESTIN and SOX2. NPC-derived neurons were generated by transition of NPCs to DMEM/F12 (1:1) (Thermo Fisher, 11320033) supplemented with sodium pyruvate (1 mM, Thermo Fisher, 11360070), B27 supplement minus vitamin A (1×, Thermo Fisher, 12587010), N2 supplement (1×, Thermo Fisher, 17502048), laminin (1 µg ml⁻¹, Thermo Fisher, OB1415-01) and FBS (10%, GeminiBio, 100-106) on six-well plates coated with Matrigel (Corning 354234, diluted 1:100 in phosphate-buffered saline). Medium was replaced every 3 days for 30 days. Cells were passaged using Accutase (STEMCELL Technologies, 07922) and re-plated onto freshly coated plates at low density to promote differentiation. Neuronal identity was confirmed by RNA sequencing analysis of *DLX1* and *DLX2* expression.

## Animals

All care and treatment of experimental animals were carried out in strict accordance with Good Animal Practice as defined by the US Office of Laboratory Animal Welfare and approved by the UT Southwestern Medical Center Institutional Animal Care and Use Committee (protocols 2017-101840, 2019-102795 and 2022-102897). Animal welfare assessments were carried out daily during treatment periods. Animals were housed in a pathogen-free environment between 20–26 °C and at 30–70% humidity, with a 12-h light–dark cycle. Fox Chase SCID (Charles River 236, RRID: IMSR_CRL:236) mice pre-catheterized in the jugular vein with a one-channel 25-gauge vascular access button (Instech Laboratories, VABM1B/25) were obtained from Charles River Laboratories at 8–10 weeks of age. C57BL/6J (RRID: IMSR_JAX:000664) mice were obtained from the UT Southwestern Mouse Breeding Core or Jackson Laboratories. Mice were housed together (2–5 mice of the same sex per cage) and provided free access to chow diet (Teklad, 2916) and water. Catheters were flushed with 20 USP U ml⁻¹ heparin in normal saline (Instech Laboratories, USP-HS-020-0.5-NJ1P-50) every 3–5 days before infusion experiments.

## MGG152 orthotopic xenografts

Littermates of the same sex were randomized to receive or not receive orthotopic xenografts. Orthotopic glioma cell implantations were performed 1–2 weeks after receipt of mice. Mice were provided analgesia by subcutaneous injection of buprenorphine SR (0.1 mg kg⁻¹) and anaesthetized with isoflurane, then immobilized using a stereotactic frame. An incision was made to expose the skull surface and a hole was drilled into the skull 3 mm anterior and 2 mm lateral to the lambda, 2.5 mm below the surface of the brain. Then, $1 \times 10^5$ MGG152 cells suspended in 1–3 µl of culture medium were injected into the brain through the hole using a 5-µl syringe (Hamilton, 7634-01). The skin was closed with surgical clips. Mice were provided additional analgesia by subcutaneous injection of carprofen (5 mg kg⁻¹) every 24 h until 72 h after the procedure. Mice within groups were randomized to infusion groups using a computer-generated schedule and infusions were initiated in a randomized order. Mice were killed after infusion or when they exhibited neurologic symptoms, displayed a body condition score of 1 or less, or became moribund.

## Human participants

The study was conducted according to the principles of the Declaration of Helsinki. Patient tissue and blood were collected following ethical and technical guidelines on the use of human samples for biomedical research at the University of Pittsburgh Medical Center after informed patient consent under a protocol approved by the University of Pittsburgh Medical Center's Institutional Review Board. Samples were included consecutively as they became available. Sex was determined from clinical records. No sex-stratified analysis could be performed due to limited sample availability. All patient samples were de-identified

before processing. All patient samples and organoids were diagnosed and graded according to the 2021 WHO Classification of Tumours of the Central Nervous System[65]. Human tumour explants for stable isotope tracing were derived from patients with characteristics that can be found in Supplementary Table 4. Human platelets (Charles River, Platelet-1040) were obtained from Charles River Laboratories from a 28-year-old female donor.

## Vectors

lentiCRISPR v2-Blast (Addgene, 83480, a gift from M. Babu) was digested with *BsmBI-v2* (New England Biolabs, R0739) and sgRNAs targeting the *AAVS1* safe harbour locus (TAAGCAAACCTTAGAGGTTC)[66], *CAD* (CTCTGCGTGAGCCACAATG), *DHODH* (CATCTTATAAAGTCCGTCCA)[11] or *UMPS* (TGTATAAGGCACTCCACACA) were ligated into this backbone to generate a construct expressing Cas9 and the desired sgRNA. Reaction mixtures were transformed into XL10-Gold Ultracompetent Cells (Agilent, 200315) and confirmed by whole plasmid sequencing (Plasmidsaurus).

*CAD* WT (RefSeq NM_004341.5) complementary DNA was amplified by PCR with Q5 Hot Start High-Fidelity 2× Master Mix polymerase (New England Biolabs, M0494). A cDNA library prepared from NHA donor #1 cells with the RNeasy Mini kit (QIAGEN, 74004) and LunaScript RT Supermix (New England Biolabs, M3010) was used to provide template. The following primers were designed to append 5′ *attB1* and 3′ HA tag and *attB2* sites:

*attB1–CAD:* GGGGACAAGTTTGTACAAAAAAGCAGGCTTAATGGCGGCCCTAGTGTTGGAGGACGGGT

*CAD–HA–attB2:* GGGGACCACTTTGTACAAGAAAGCTGGGTTCTAAGCGTAATCTGGAACATCGTATGGGTAGAAACGGCCCAGCACGGTGG

PCR product was gel-purified with the QIAquick Gel Extraction kit (QIAGEN, 28706), PCR purified with the Monarch PCR & DNA Cleanup kit (New England Biolabs, T1030) and cloned into the Gateway vector pDONR223 by BP reaction (Thermo Fisher, 11789100). The resultant product was transformed into HB101 Competent Cells (Promega, L2015). N-terminal Flag and HA tags followed by the linker sequence AGCGGCCGCTCGTCTGCGTGCTTGCCCTGTCTCAGCTCCGACCCG were added by restriction cloning with *ApaI* (New England Biolabs, R0114) and *SalI* (New England Biolabs, R3138) with a gBlock synthesized by Twist Biosciences to generate pENTR223–Flag–HA–CAD–HA plasmid. The resultant product was transformed into XL10-Gold Ultracompetent Cells (Agilent, 200315).

S1900A and S1900D mutant cDNAs were generated by In-Fusion cloning using pENTR223–Flag–HA–CAD–HA. Inverse PCR with Q5 Hot Start High-Fidelity 2× Master Mix polymerase (New England Biolabs M0494) of pENTR223–Flag–HA–CAD–HA plasmid used the following primers:

*CADmut_Inv_F:* ACCAGCAGCTCCTTTGCAGC
*CADmut_Inv_R:* GGTCTGGCAGAATTGGCGAG

Gene fragments containing S1900A or S1900D mutations were synthesized by Twist Biosciences and amplified using the following insert primers:

*CADmut_Ins_F:* CTCGCCAATTCTGCCAGACC
*CADmut_Ins_R:* GCTGCAAAGGAGCTGCTGGT

Amplified inserts and inverse PCR products were purified with the Monarch PCR & DNA Cleanup kit (New England Biolabs, T1030) and used in an In-Fusion exchange reaction (Takara, 102518), then transformed into XL10-Gold Ultracompetent Cells (Agilent, 200315). Final pENTR223–Flag–HA–CAD–HA product for WT, S1900D and S1900A mutants were confirmed by whole plasmid sequencing (Plasmidsaurus).

pLenti–EF1α–DEST–IRES–Neo (deposited as Addgene, 253766) was generated by In-Fusion cloning from pLenti–EF1α–IRES–Neo EV (provided by G. Lu). Inverse PCR with Q5 Hot Start High-Fidelity 2× Master Mix polymerase (New England Biolabs, M0494) of pLenti–EF1α–IRES–Neo EV used the following primers:

*LEINgate_Inv_R:* GCGGATCCGAATTCGTCGACAATTC
*LEINgate_Inv_F:* GGCCGCGTTTAAACTTAATTAAGGC

The Gateway cassette was amplified from pSLIK–Hygro (Addgene, 25737, a gift from I. Fraser) using the following primers:

*LEINgate_Ins_R:* GGCCTTAATTAAGTTTAAACGCGGCCACCACTTTG TACAAGAAAGCTGAACG
*LEINgate_Ins_F:* GAATTGTCGACGAATTCGGATCCGCACAAGTTTGTA CAAAAAAGCTGAACGAG

Amplified inserts and inverse PCR products were purified with the Monarch PCR & DNA Cleanup kit (New England Biolabs, T1030) and used in an In-Fusion exchange reaction (Takara, 102518), then transformed into XL10-Gold Ultracompetent Cells (Agilent, 200315). Final pLenti–EF1α–DEST–IRES–Neo product was confirmed by whole plasmid sequencing (Plasmidsaurus).

pLV–EF1α–DEST–IRES–Neo (deposited as Addgene, 253768) was generated from pLV–EF1α–MCS–IRES–Neo (VB241113-1256hqp) plasmid backbone synthesized by VectorBuilder. Gateway destination cassette was amplified by PCR from pLenti–EF1α–DEST–IRES–Neo using the following primers:

*MluI–attR1–LEIN_F:* TAAGCAACGCGTTGTCGACGAATTCGGATCCG
*LEIN–attR2–XbaI_R:* TAAGCATCTAGATAATTAAGTTTAAACGCG GCCACCAC

PCR products were gel-purified. PCR products and each plasmid backbone were then digested with *MluI* and *XbaI*, ligated, and transformed into XL10-Gold Ultracompetent Cells (Agilent, 200315). Final products were confirmed by whole plasmid sequencing (Plasmidsaurus).

cDNAs were cloned into the lentiviral vector pLenti–EF1α-DEST–IRES–Neo, pLV–EF1α–DEST–IRES–Neo, or pLenti–PGK–Hygro-DEST[67] (a gift from E. CCampeau and P. Kaufman, Addgene, 19066) by Gateway cloning. pENTR223-EV, pENTR223–Flag–HA–CAD_WT–HA, pENTR223–Flag–HA–CAD_S1900A–HA and pENTR223–Flag–HA–CAD_S1900D–HA were separately mixed with pLenti–EF1α–DEST–IRES–Neo, pLV–EF1α–DEST–IRES–Neo or pLenti–PGK–Hygro–DEST and LR reactions (Thermo Fisher, 11791020) performed. Resultant pLenti–EF1α–EV–IRES–Neo (deposited as Addgene, 253783), pLenti–EF1α–Flag–HA–CAD_WT–HA–IRES–Neo (deposited as Addgene, 253784), pLenti–EF1α–Flag–HA–CAD_S1900A–HA–IRES-Neo (deposited as Addgene, 253785), pLenti–EF1α–Flag–HA–CAD_S1900D–HA–IRES–Neo (deposited as Addgene, 253786), pLV–EF1α–EV–IRES–Neo (deposited as Addgene 253787), pLV–EF1α–Flag–HA–CAD_WT–HA–IRES–Neo (deposited as Addgene, 253788), pLV–EF1α–Flag–HA–CAD_S1900A–HA–IRES–Neo (deposited as Addgene, 253789), pLV–EF1α–Flag–HA–CAD_S1900D–HA–IRES–Neo (deposited as Addgene, 253790), pLenti–PGK–EV–Hygro (deposited as Addgene, 253791) and pLenti–PGK–Flag–HA–CAD_WT–HA–Hygro (deposited as Addgene, 253792) lentiviral vectors for mammalian cell expression were transformed into HB101 Competent Cells (Promega, L2015) and confirmed by whole plasmid sequencing (Plasmidsaurus).

## Transfection and viral transduction

Lentiviral particles were produced by cotransfection of HEK293T cells with expression vectors and packaging plasmids psPAX2 (Addgene, 12260, a gift from D. Trono) and pMD2.G (Addgene, 12259, a gift from D. Trono) in a ratio of 4:3:1 using TransIT-LT1 transfection reagent (Mirus Bio, MIR2B-304). Virus-containing medium was collected 48 and 72 h after transfection, passed through a 0.45-µm filter (Corning, 431220), divided into 1-ml aliquots and frozen at −80 °C until use.

NHA donor #1, G9C CHO or HeLa cells were plated at a density of $0.15 \times 10^6$ cells per well on a six-well plate. The next day, Polybrene (8 µg ml$^{-1}$, MedChemExpress, HY-112735) was added to each well in addition to 1 ml viral supernatant. Plates were centrifuged at 4,000$g$ for 30 min at room temperature and incubated overnight. The following day, cells were expanded and re-plated in a 10-cm dish. Stable cell lines were selected in 1 mg ml$^{-1}$ G418 (GoldBio, G-418) or 2 µg ml$^{-1}$ blasticidin

(GoldBio, B-800) based on the selection cassette present in each vector. After initial selection, stable cell lines were maintained in 400 µg ml$^{-1}$ G418 (GoldBio, G-418) or 800 ng ml$^{-1}$ blasticidin (GoldBio, B-800) based on the selection cassette present in each vector.

## Preparation of HPLM and tracer HPLM

HPLM was prepared in individual pools[9,10]. The compositions and preparation protocols for the 19 constituent pools (both unlabelled and tracer formulations) are provided in the 'Pool' and 'Preparation' columns of Supplementary Table 1. For tracer HPLM used in profiling experiments, stock solutions were generated identically to the unlabelled version, substituting the labelled metabolite for its unlabelled counterpart at equimolar concentrations. The tracer HPLM library was assembled by mixing thawed frozen stocks of unlabelled HPLM pools, replacing one pool with its labelled version per tracing condition. Pools designated for fresh preparation were prepared at the time of library assembly. The medium was adjusted to pH 7.4 using NaOH or HCl, brought to the final volume, sterile filtered with a 0.22-µm PES filter (Millipore Sigma SCGP00525; Corning, 431153; Corning, 431097 or Corning, 431098) and supplemented as described below.

With the exception of conditions in which glutamate was traced, HPLM was prepared without glutamate. For experiments in GSCs, NSCs and HNF-derived NPCs, HPLM was supplemented with B27 supplement (1×, Thermo Fisher, 17504044), N2 supplement (0.25×, Thermo Fisher, 17502048), EGF (20 ng ml$^{-1}$, GoldBio, 1150-04), bFGF (20 ng ml$^{-1}$, Gold-Bio, 1140-02), heparin (2 µg ml$^{-1}$, STEMCELL Technologies, 07980), penicillin–streptomycin (100 U ml$^{-1}$ and 100 µg ml$^{-1}$, respectively, Thermo Fisher, 15-140-148), amphotericin B (250 ng ml$^{-1}$, GeminiBio, 400104) and Plasmocin (250 ng ml$^{-1}$, InvivoGen ant-mpp). For experiments in differentiated cells, HPLM was supplemented with dialysed FBS (10%, GeminiBio, 100-108) and penicillin–streptomycin (100 U ml$^{-1}$ and 100 µg ml$^{-1}$, respectively, Thermo Fisher, 15140148).

## Preparation of cultured cells for LC–MS

BT054 cell samples treated with dimethylsulfoxide (DMSO) or AGI-5198 (except those involving low-throughput $^{15}$N-BCAA tracing) were washed with ice-cold saline and snap frozen in liquid nitrogen[11]. Before LC–MS analysis, samples were suspended in ice-cold 80% methanol (Thermo Fisher, A456), vortexed for 20 min at 4 °C, and centrifuged for 10 min at 17,000–21,100$g$ at 4 °C. Samples were then dried using a SpeedVac (Thermo Fisher, SPD2030). Dried metabolites were resuspended in 1 µl ice-cold 80% acetonitrile (Thermo Fisher, A9554) per 10,000 cells, vortexed for 20 min at 4 °C and centrifuged for 10 min at 17,000–21,100$g$ at 4 °C. The resultant supernatant was then analysed by LC–MS.

Platelets were washed with ice-cold normal saline and snap frozen in liquid nitrogen. Before LC–MS analysis, samples were resuspended in 55 µl ice-cold 80% acetonitrile (Thermo Fisher, A9554), vortexed for 20 min at 4 °C and centrifuged for 10 min at 17,000–21,100$g$ at 4 °C. The supernatant was transferred to a fresh tube and again centrifuged for 10 min at 17,000–21,100$g$ at 4 °C. The resultant supernatant was then analysed by LC–MS.

All other cell samples were washed with ice-cold normal saline and snap frozen in liquid nitrogen. Before LC–MS analysis, samples were suspended in 1 µl ice-cold 80% acetonitrile (Thermo Fisher, A9554) per 1,000 cells, vortexed for 20 min at 4 °C, and centrifuged for 10 min at 17,000–21,100$g$ at 4 °C. The supernatant was transferred to a fresh tube and again centrifuged for 10 min at 17,000–21,100$g$ at 4 °C. The resultant supernatant was then analysed by LC–MS.

## Preparation of medium for LC–MS

Unconditioned HPLM stocks generated for the nitrogen metabolism profiling platform were collected and snap frozen in liquid nitrogen. Before LC–MS analysis, 50 µl of the medium was added to a 5-ml tube. Then, 1 ml 100% methanol (Thermo Fisher, A456), 1 ml of 100% chloroform (Millipore Sigma, 650471) and 1 ml of water were added, with brief

vortexing after each addition. Samples were centrifuged for 5 min at 410g at 4 °C. Then, 1 ml of the aqueous phase was transferred to a 1.5-ml tube. Samples were dried using a SpeedVac (Thermo Fisher, SPD2030). Dried metabolites were resuspended in 100 µl ice-cold 80% acetonitrile (Thermo Fisher, A9554), vortexed for 20 min at 4 °C and centrifuged for 10 min at 17,000–21,100g at 4 °C. The resultant supernatant was then analysed by LC–MS.

Conditioned medium samples were collected and snap frozen in liquid nitrogen. Before LC–MS analysis, the medium was diluted 1:100 in ice-cold 80% acetonitrile (Thermo Fisher, A9554) and vortexed for 20 min at 4 °C, then centrifuged for 10 min at 17,000–21,100g at 4 °C. The supernatant was transferred to a fresh tube and again centrifuged for 10 min at 17,000–21,100g at 4 °C. The resultant supernatant was then analysed by LC–MS.

## Preparation of blood for LC–MS
Blood samples were collected from mice and snap frozen in liquid nitrogen. Before LC–MS analysis, blood was diluted 1:100 in ice-cold 80% acetonitrile (Thermo Fisher, A9554) and vortexed for 20 min at 4 °C, then centrifuged for 10 min at 17,000–21,100g at 4 °C. The supernatant was transferred to a fresh tube and again centrifuged for 10 min at 17,000–21,100g at 4 °C. The resultant supernatant was then analysed by LC–MS.

## Preparation of tissue for LC–MS
Adult mouse tissues were collected and snap frozen in liquid nitrogen. Then, 50 µl ice-cold 80% methanol (Thermo Fisher, A456) per mg of tissue was added to samples, which were then homogenized using a TissueLyser II (QIAGEN, RRID: SCR_018623). Samples were vortexed for 20 min at 4 °C, then centrifuged for 10 min at 17,000–21,100g at 4 °C. The supernatant was transferred to a fresh tube and again centrifuge for 10 min at 17,000–21,100g at 4 °C. Then, 200 µl of supernatant was transferred to a fresh tube, then dried using a SpeedVac (Thermo Fisher, SPD2030). Dried metabolites were resuspended in 100 µl ice-cold 80% acetonitrile (Thermo Fisher, A9554), vortexed for 20 min at 4 °C, and centrifuged for 10 min at 17,000–21,100g at 4 °C. The resultant supernatant was then analysed by LC–MS.

Fetal mouse tissues were collected and snap frozen in liquid nitrogen. Ice-cold 80% acetonitrile (Thermo Fisher, A9554) was added to samples, which were then homogenized using a rubber Dounce homogenizer. Samples were flash frozen three times in liquid nitrogen and then centrifuged for 10 min at 17,000g at 4 °C. Supernatants were analysed for protein content using a BCA assay, normalized to 70 µg ml⁻¹, then analysed by LC–MS.

Human samples for steady-state metabolomics were collected from the operating room and snap frozen before preparation for LC–MS[24,68]. Samples were thawed and washed in 1 ml ice-cold normal saline. Then, 50 µl ice-cold 80% methanol (Thermo Fisher, A456) per mg of tissue was added to samples, which were then homogenized using a TissueLyser II (QIAGEN, RRID: SCR_018623). Then, 520 µl tert-butyl methyl ether (Millipore Sigma, 650560) and 380 µl water were added to 500 µl of homogenate in a glass vial and mixture was vortexed at room temperature for 1 h at 1,000 rpm in a thermal mixer (Benchmark Scientific, H5000-HC). Samples were centrifuged for 10 min at 1,000g at 4 °C. Then, 800 µl of the lower aqueous phase and 300 µl of the upper organic phase were combined in a new tube, avoiding interlayer debris. Samples were then dried using a SpeedVac (Thermo Fisher, SPD2030). Dried metabolites were resuspended in 100 µl ice-cold 80% acetonitrile (Thermo Fisher, A9554) and an internal standard mix of 3 µM $^{13}C_5$-glutamine (Cambridge Isotope Laboratories, CLM-1822), 3 µM $^{15}N$-valine (Cambridge Isotope Laboratories, NLM-316), 3 µM $^{15}N$-methionine (Cambridge Isotope Laboratories, NLM-752) and 3 µM chloramphenicol (Millipore Sigma, C0378) was added. Samples were sonicated in a room temperature water bath for 10 min to ensure metabolite dissolution. Samples were then centrifuged for 10 min

at 21,100g at 4 °C. The resultant supernatant was then analysed by LC–MS.

Human tissue explants were snap frozen following stable isotope tracing. 50 µl ice-cold 80% acetonitrile (Thermo Fisher, A9554) per mg of tissue was added to samples, which were then homogenized using a rubber Dounce homogenizer. Samples were vortexed for 20 min at 4 °C, then centrifuged for 10 min at 17,000–21,100g at 4 °C. The resultant supernatant was again centrifuged for 10 min at 17,000–21,100g at 4 °C. The resultant supernatant was then analysed by LC–MS.

## Metabolite quantification by LC–MS
For metabolite quantification, 10–20 µl of acetonitrile-resuspended metabolite extract was injected and analysed with a Q-Exactive HF-X (RRID: SCR_020425), Orbitrap Fusion Lumos 1 M (RRID: SCR_020562) or Orbitrap Exploris 480 (RRID: SCR_027000) hybrid quadrupole-orbitrap mass spectrometer (Thermo Fisher) coupled to a Vanquish Flex UHPLC system (Thermo Fisher)[12,13]. Chromatographic resolution of metabolites was achieved using a Millipore ZIC-pHILIC column using a linear gradient of 10 mM ammonium formate pH 9.8 and acetonitrile. Spectra were acquired with a resolving power of 60,000, 120,000 or 240,000 full width at half maximum (FWHM), a scan range set to 60–900, 70–1,050 or 80–1,200 m/z, and polarity switching or a targeted selected ion monitoring (tSIM) scan for low-abundance metabolites. Data-dependent MS/MS data were acquired on unlabelled pooled samples to confirm metabolite IDs when necessary. All solvents used for MS experiments were LC–MS Optima grade (Thermo Fisher).

Peaks were integrated using El-Maven v.0.12.0 software (Elucidata, RRID: SCR_022159) or TraceFinder v.5.1 SP2 software (Thermo Fisher, OPTON-31001, RRID: SCR_023045). Total ion counts were quantified using TraceFinder v.5.1 SP2 software (Thermo Fisher, OPTON-31001, RRID: SCR_023045). Peaks were normalized using probabilistic quotient normalization[69] and total ion counts using R (RRID: SCR_001905). For stable isotope-tracing studies, correction for natural abundance of metabolite labelling was performed using AccuCor[70] v.0.3.0 (RRID: SCR_023046) in R (RRID: SCR_001905).

## MALDI-MSI tissue preparation and microscopy
Tissue samples for MALDI-MSI analysis were collected from mice and flash frozen for three cycles of 3 s each, separated by 2 s at room temperature, in liquid nitrogen. Samples were cryo-sectioned in the coronal plane to 10 µm thickness and thaw-mounted onto indium tin oxide slides. Serial sections were collected for H&E staining and imaged using a ×10 objective (Zeiss Observer Z.1).

## MALDI matrix preparation
Glutamine and UMP were imaged using 1,5-diaminonaphthalene hydrochloride (4.3 mg ml⁻¹) matrix solution prepared in 4.5/5/0.5 HPLC grade water/ethanol/1 M HCl (v/v/v). The matrix was applied using a four-pass cycle with 0.09 ml min⁻¹ flow rate, spray nozzle velocity (1,200 mm min⁻¹), spray nozzle temperature (75 °C), nitrogen gas pressure (10 psi) and track spacing (2 mm).

## Glutamine and UMP MALDI-MSI
Glutamine and UMP were imaged from serial tissue sections. Mass spectrometry imaging data was acquired using a 15 Tesla SolariX XR FT-ICR MS (Bruker, RRID: SCR_027095). Instrument parameters were set to negative ion mode, and the mass range was from m/z 46.07 to 3,000. The laser raster size was 100 µm and each sampling point consisted of 200 laser shots at a laser power of 21% (arbitrary scale) with a laser repletion rate of 100 Hz. Continuous Accumulation of Selected Ions mode was used with setting Q1 to m/z 150 with an isolation window of 200. Instrument calibration was carried out using a tune mix solution (Agilent) with the electrospray ionization source. A $^{15}N$ glutamate internal standard was used for online calibration during acquisition. MSI data analysis was completed using SCiLS Lab 2024a Pro (Bruker,

RRID: SCR_014426) with normalization to $^{15}$N glutamate. Glutamine and UMP were putatively annotated based on an accurate mass with Δppm < 0.1 and MS/MS measurements.

## MALDI-MSI isotopic labelling analysis

Acquired MALDI-MSI datasets were preprocessed, visualized, and exported to imZML[71] format using SCiLS Lab 2023a Core (Bruker, RRID: SCR_014426). Per-pixel natural abundance correction and generation of fractional enrichment images were performed using a pipeline implemented in R v.4.1.1 (RRID: SCR_001905). Data were imported with rMSIproc[72] v.0.3.166 and theoretical mass spectra or each isotopologue were calculated using enviPat[73] v.2.767 (RRID: SCR_003034). Natural abundance correction was then performed by solving a system of linear equations incorporating theoretical isotopologue ratios alongside the experimental spectra at each pixel[74].

## Nitrogen metabolism profiling of AGI-5198- or DMSO-treated BT054 cells

BT054 cells were plated in 2 ml six-well ultra-low adherence plates ($0.5 \times 10^6$ cells per well) in NeuroCult NS-A Basal Medium (Human) prepared as described above. After 24 h, the medium was changed to 2 ml of 50% NeuroCult (Human) and 50% unlabelled HPLM prepared as described above. After 24 h, the medium was changed to 100% unlabelled HPLM. Either DMSO or 3 µM AGI-5198 (MedKoo Biosciences 406264) was added. After 24 h, the medium was again changed to fresh 100% unlabelled HPLM with either DMSO or 3 µM AGI-5198. After 24 h, this was changed to fresh 100% HPLM with one nitrogen-containing metabolite per sample exchanged for an equimolar amount of its $^{15}$N-labelled counterpart (Supplementary Table 1) and either DMSO or 3 µM AGI-5198. After 18 h, samples were collected and prepared for LC−MS analysis as described above.

## Nitrogen metabolism profiling of NHA and BT054 cells

NHA donor #1 cells were plated in six-well plates ($0.025 \times 10^6$ cells per well) in 2 ml DMEM prepared as described above. BT054 cells were plated in six-well ultra-low adherence plates ($0.2 \times 10^6$ cells per well) in 2 ml NeuroCult NS-A Basal Medium (Human) prepared as described above. After 24 h, 2 ml unlabelled HPLM prepared as described above was added to produce a mixture of 50% native medium and 50% HPLM. After 24 h, the medium was changed to 100% unlabelled HPLM (10 ml for NHA donor #1 cells and 4.5 ml for BT054 cells). After 24 h, this was changed to fresh 100% HPLM with one nitrogen-containing metabolite per sample exchanged for an equimolar amount of its $^{15}$N-labelled counterpart (Supplementary Table 1; 10 ml for NHA donor #1 cells and 4.5 ml for BT054 cells). After 18 h, samples were collected and prepared for LC−MS analysis as described above.

## Pyrimidine synthesis tracing in cells

Differentiated cells were plated in six-well plates ($0.025 \times 10^6$ cells per well for NHA donor #1, $0.15 \times 10^6$ cells per well for NHA donor #2 and G9C CHO, $0.05 \times 10^6$ cells for BJ, $0.125 \times 10^6$ cells per well for HNF, $0.02 \times 10^6$ cells for NPC-derived neurons) in 2 ml RPMI 1640 (G9C CHO cells) or DMEM (all other differentiated cells) prepared as described above. GSCs were plated in six-well ultra-low adherence plates ($0.2 \times 10^6$ cells per well) in 2 ml NeuroCult NS-A Basal Medium (Human) prepared as described above. NSCs were plated in six-well ultra-low adherence plates ($0.2 \times 10^6$ cells per well) in 2 ml Neurobasal Medium prepared as described above. HNF-derived NPCs were plated in six-well plates ($0.15 \times 10^6$ cells per well) in 2 ml DMEM/F12 (1:1) prepared as described above. After 24 h, 2 ml unlabelled HPLM prepared as described above was added to produce a mixture of 50% native medium and 50% HPLM. After 24 h, the medium was changed to 100% unlabelled HPLM (10 ml for differentiated cells, 4.5 ml for GSCs, NSCs, HNF-derived NPCs and neurons). After 24 h, the medium was changed to fresh 100% HPLM (10 ml for differentiated cells, 4.5 ml for GSCs, NSCs and HNF-derived NPCs)

modified as follows: glutamine exchanged for an equimolar amount of amide-$^{15}$N-glutamine (Cambridge Isotope Laboratories, NLM-557), glutamine exchanged for an equimolar amount of amide-$^{15}$N-glutamine (Cambridge Isotope Laboratories, NLM-557) and uridine removed, or uridine exchanged for an equimolar amount of $^{15}N_2$ uridine (Cambridge Isotope Laboratories, NLM-812). After incubation with tracer for indicated timepoints, samples were collected and prepared for LC−MS analysis as described above.

For stable isotope tracing experiments with altered growth factor sources, 24 h after plating in their native medium NHA donor #1 cells were cultured in conditions described for GSCs, NSCs and HNF-derived NPCs and BT054 cells were cultured in conditions described for differentiated cells. For stable isotope tracing experiments with altered adherence conditions, ENSA and NSC11 cells were plated on six-well plates coated with Matrigel (Corning, 354234) diluted 1:100 in phosphate-buffered saline. For stable isotope tracing experiments with uridine deprivation preconditioning, during the final 24 h before culture in amide-$^{15}$N-glutamine HPLM, NHA donor #1 cells were instead cultured in 10 ml 100% HPLM with uridine removed.

## HPLM metabolite depletion analysis

NHA donor #1 cells were plated in six-well plates (Extended Data Fig. 2a, $0.025 \times 10^6$ cells per well and Fig. 5a, $0.2 \times 10^6$ cells per well) in 2 ml DMEM prepared as described above. BT054 cells were plated in six-well ultra-low adherence plates ($0.2 \times 10^6$ cells per well) in 2 ml NeuroCult NS-A Basal Medium (Human) prepared as described above. After 24 h, 2 ml unlabelled HPLM prepared as described above was added to produce a mixture of 50% native medium and 50% HPLM. After 24 h, the medium was changed to 100% unlabelled HPLM (10 ml for NHA donor #1 in Extended Data Fig. 2a, 4.5 ml for NHA donor #1 in Fig. 5a and BT054). After 24 h, the medium was changed to fresh 100% unlabelled HPLM (10 ml for NHA donor #1 in Extended Data Fig. 2a, 4.5 ml for NHA donor #1 in Fig. 5a and BT054). After 18 h, the medium was collected and prepared for LC−MS analysis as described above.

## Doubling time quantification

NHA donor #1 cells were plated in six-well plates ($0.05 \times 10^6$ cells per well) in 2 ml DMEM prepared as described above. 24, 48 and 72 h later, cells were counted using a Vi-CELL XR (Beckman Coulter, RRID: SCR_019664) cell viability analyser. Doubling time was calculated using nonlinear regression to fit an exponential growth curve in GraphPad Prism (RRID: SCR_002798).

## RNA sequencing

RNA was prepared from cells using the RNeasy Plus Mini kit (QIAGEN, 74136) according to the manufacturer's instructions. RNA sequencing and analysis was performed by Plasmidsaurus using Illumina sequencing with a 3′ end counting approach. Results were normalized to counts per million to allow for comparison across samples.

## NPC self-renewal experiments

Self-renewal capacity was assessed using a limiting dilution assay. NPCs were plated into six-well ultra-low adherence plates (Corning, 3474) at 3, 6, 12, 25, 50, 100, 200 or 400 cells per well and cultured in DMEM/F12 (1:1) (Thermo Fisher, 11320033) supplemented with sodium pyruvate (1 mM, Thermo Fisher, 11360070), B27 supplement minus vitamin A (1×, Thermo Fisher, 12587010), N2 supplement (1×, Thermo Fisher, 17502048), laminin (1 µg ml$^{-1}$, Thermo Fisher, OB1415-01) and bFGF (20 ng ml$^{-1}$, GoldBio, 1140-02) supplemented with either 0 µM, 3 µM or 100 µM uridine and treated with either 10 nM orludodstat or DMSO. After 7 days of culture, the presence of neurospheres in each well was recorded using a Celigo 4 Channel Image Cytometer (Nexcelom, 200-BFFL-S). Extreme limiting dilution analysis was performed using an online tool (https://bioinf.wehi.edu.au/software/elda/)[75] according to the developer's instructions.

## Neuron differentiation experiments

Neuron differentiation under varying uridine concentrations was assessed by culturing NPCs in DMEM/F12 (1:1) (Thermo Fisher, 11320033) supplemented with sodium pyruvate (1 mM, Thermo Fisher, 11360070), B27 supplement minus vitamin A (1×, Thermo Fisher, 12587010), N2 supplement (1×, Thermo Fisher, 17502048), laminin (1 µg ml$^{-1}$, Thermo Fisher, OB1415-01) and dialysed FBS (10%, GeminiBio, 100-108) and either 0 µM or 3 µM uridine. Then, $2 \times 10^5$ cells per well in six-well plates coated with Matrigel (Corning, 354234, diluted 1:100 in phosphate-buffered saline). Medium was replaced every day until day 10, when cells were collected for RNA sequencing analysis.

## Cell death assay

Cells were stained with annexin V-FITC (BD Biosciences, 556547) and 4,6-diamidino-2-phenylindole (DAPI) (162.5 ng ml$^{-1}$ final concentration) according to manufacturer's instructions to identify early and late apoptotic cells, respectively. Cells were analysed using an LSR Fortessa (BD Biosciences) flow cytometer and data were processed using FCS Express software (De Novo). Dead cells included those that were annexin V$^+$/DAPI$^-$, annexin V$^-$/DAPI$^+$ or annexin V$^+$/DAPI$^+$.

## Adult mouse infusions

Infusions and tissue collections were performed by researchers who were not blinded to the treatment arms or genotypes of the mice. No successfully infused mice were excluded from analysis. Mice that did not undergo orthotopic xenograft surgery were infused 1–7 weeks after receipt. Mice that did undergo orthotopic xenograft surgery were infused 34 days after surgery. Littermates of the same sex and xenograft status were randomized to tracer groups at the time of infusion. Amide-$^{15}$N-glutamine (Cambridge Isotope Laboratories, NLM-557) was prepared at 30 mg ml$^{-1}$ in sterile normal saline and infused at a rate of 3 mg kg$^{-1}$ min$^{-1}$. $^{15}$N$_2$-uridine (Cambridge Isotope Laboratories, NLM-812) was prepared at 3 mg ml$^{-1}$ in sterile normal saline and infused at a rate of 0.3 mg kg$^{-1}$min$^{-1}$. Three females and one male were infused per group of tracer and xenograft status. For saline infusions, one female and one male mouse per group of xenograft status were infused at a rate of 1 µl g$^{-1}$ min$^{-1}$. The infusion apparatus (Instech Laboratories) included a swivel and tether to allow free movement around the cage. Mice were monitored constantly during infusion. During infusion, blood was collected by tail snip (~5 µl) in blood collection tubes with K$_2$EDTA (Becton Dickinson, 366643). Mice were infused for 18 h, after which mice were anaesthetized with isoflurane and killed by decapitation. Tissues were quickly dissected and divided into two halves: one for MALDI-MSI analysis, which was immediately frozen, and one for LC–MS analysis, which was dissected manually to isolate tumour region, washed in ice-cold normal saline, then frozen. Tissue and blood were stored at −80 °C until analysis.

## Fetal mouse infusions

Healthy 12-week-old naive pregnant female C57BL/6J mice were set up for mating between 5:00 and 7:00 with proven studs of the same genotype. The following morning, females displaying vaginal plugs were identified as pregnant and moved to a new cage until gestational day 11. Infusion took place between 9:00 and 11:00 with no previous fasting of pregnant dams. Mice were initially anaesthetized using intraperitoneal injection of ketamine and xylazine (120 mg kg$^{-1}$ and 16 mg kg$^{-1}$, respectively) and maintained under anaesthesia using subsequent intraperitoneal doses of ketamine (20 mg kg$^{-1}$) as needed. 25-gauge catheters were inserted into the tail vein and infusion begun immediately after an initial retro-orbital blood collection. Amide-$^{15}$N-glutamine (Cambridge Isotope Laboratories, NLM-557) was prepared as a total dose of 1.73 g kg$^{-1}$ in 1,500 µl sterile normal saline infused at a rate of 147 µl min$^{-1}$ for 1 min followed by 3 µl min$^{-1}$ for 5 h (ref. 76). Retro-orbital blood was collected throughout the infusion to monitor tracer enrichment in maternal blood. Mice were killed at the end of the infusion,

then the uterus was removed, and embryonic brain was dissected in cold sodium chloride irrigating solution (Baxter, 2F7122) then snap frozen in liquid nitrogen. Tissue was stored at −80 °C until analysis.

## Tracing in human explants

Tumour tissue was collected using Brainlab Stereotactic Frameless Navigation Suite (Brainlab, RRID: SCR_026474). Tumour or brain tissue was collected from the operating room and immediately suspended in ice-cold Hibernate A (BrainBits, HA). Within 30 min of explantation, tissue was transferred to RBC Lysis Buffer (Thermo Fisher, 00433357) for 10 min with rocking. Next, tissue was washed three times with Hibernate A containing GlutaMAX (2 mM, Thermo Fisher, 35050061), penicillin–streptomycin (100 U ml$^{-1}$ and 100 µg ml$^{-1}$, respectively, Thermo Fisher, 15140148), and amphotericin B (250 ng ml$^{-1}$, GeminiBio, 400104). Tissues were cut using dissection scissors or scalpels into 1–2 mm$^3$ pieces and plated in a 24-well ultra-low adherence plate (Corning, 3473) in 1 ml HPLM supplemented with B27 supplement minus vitamin A (1×, Thermo Fisher, 12587010), N2 supplement (1×, Thermo Fisher, 17502048), penicillin–streptomycin (100 U ml$^{-1}$ and 100 µg ml$^{-1}$, respectively, Thermo Fisher, 15140148), Plasmocin (250 ng ml$^{-1}$, Invivo-Gen ant-mpp), 2-mercaptoethanol (55 µM, Thermo Fisher, BP176100) and insulin (2.375–2.875 µg ml$^{-1}$, Millipore Sigma, I9278). Tissues were randomized to groups of 1–4 technical replicates per tracer from each biological sample, depending on available tissue. Plated tissues were placed in a 37 °C incubator at ambient oxygen and 5% CO$_2$ with shaking. Remaining tissue was used for histopathology as described below. After 30 min, the medium was exchanged for 1 ml HPLM supplemented as above with either glutamine exchanged for an equimolar amount of amide-$^{15}$N-glutamine (Cambridge Isotope Laboratories, NLM-557) or uridine exchanged for an equimolar amount of $^{15}$N$_2$ uridine (Cambridge Isotope Laboratories, NLM-812). Tissues were incubated for 18 h in a 37 °C incubator at ambient oxygen and 5% CO$_2$ with shaking. After 18 h, tissues were collected, washed with ice-cold normal saline and stored at −80 °C until analysis. Tissues were then prepared for LC–MS analysis as described above.

## Histopathology of human explants

Residual human tumour and brain tissues from parcellation for tracing experiments were fixed for 1 h in neutral buffered 10% formalin solution (Millipore Sigma, HT501128). Following fixation, tissues were washed and stored in 70% ethanol. Tissues were embedded in paraffin, sectioned, and stained with H&E by the University of Texas Southwestern Medical Center (UTSW) Histo Pathology Core. H&E sections were reviewed by a board-certified neuropathologist (T.E.R.).

## Pyrimidine synthesis analysis in continuous cell culture with medium change

NHA donor #1 cells were plated on 6 cm dishes (0.8 × 10$^6$ cells per dish) in 2 ml DMEM prepared as described above. After 18 h, and every subsequent 6 h for an additional 24 h, the medium was changed to 2 ml fresh DMEM. At the indicated time points after the medium changes, the medium and cells were collected and prepared for LC–MS analysis as described above. At the same time, a paired plate was counted with a Vi-CELL XR (Beckman Coulter, RRID: SCR_019664) cell viability analyser.

## Immunoblot analysis of protein expression

Cells were lysed in EBC lysis buffer containing either a protease inhibitor cocktail (Millipore Sigma, 11836153001) or a protease and phosphatase inhibitor cocktail (Thermo Fisher, 78440). Lysates were resolved by SDS–PAGE and transferred to nitrocellulose membranes (Bio-Rad, 1620112). Primary antibodies used included: anti-IDH1-R132H (Dianova, DIA-H09, 1:500, mouse monoclonal, RRID: AB_2335716), anti-vinculin (Sigma, V9131, 1:100,000, mouse monoclonal, RRID:AB_477629), anti-phospho-CAD (Ser1859) (Cell Signalling Technologies, 70307,

1:1,000, rabbit monoclonal, RRID: AB_2799782), anti-CAD (Cell Signalling Technologies, 11933, 1:1,000, rabbit polyclonal, RRID: AB_2797772), anti-DHODH (Proteintech, 14877, 1:1,000, rabbit polyclonal, RRID: AB_2091723), anti-UMPS (Millipore Sigma, HPA036179, 1:1,000, rabbit polyclonal, RRID: AB_10673615), anti-UMPS (Proteintech, 14830, 1:1,000, rabbit polyclonal, RRID: AB_2212392), anti-HA (Thermo Fisher, 26183, 1:1,000, mouse monoclonal, RRID: AB_10978021), anti-HA (BioLegend, 901513, 1:1,000, mouse monoclonal, RRID: AB_2565335), anti-Flag (Millipore Sigma, F1804, 1:1,000, mouse monoclonal, RRID: AB_262044), anti-phospho-S6 ribosomal protein (Ser240/244) (Cell Signalling Technologies, 5364, 1:1,000, rabbit monoclonal, RRID: AB_10694233), anti-S6 ribosomal protein (Cell Signalling Technologies, 2217, 1:1,000, rabbit monoclonal, RRID: AB_331355), anti-phospho-p70 S6 kinase (Thr389) (Cell Signalling Technologies, 9234, 1:1,000, rabbit monoclonal, RRID: AB_2269803) and anti-p70 S6 kinase (Cell Signalling Technologies, 2708, 1:1,000, rabbit monoclonal, RRID: AB_390722). HRP-conjugated secondary antibodies used included: anti-mouse IgG (Thermo Fisher, 31430, 1:2,000, goat polyclonal, RRID: AB_228307) and anti-rabbit IgG (Thermo Fisher, 31460, 1:2,000, goat polyclonal, RRID: AB_228341).

### Immunoblot densitometry

Immunoblot band intensities were quantified by densitometry using Fiji/ImageJ2 1.54p using images that were not saturated. A fixed-size rectangular region of interest was placed over each band for quantification and the corresponding band for loading control. Local background signal was measured from an adjacent area and subtracted from each band measurement. Background-corrected signals were then normalized to loading control.

### Pyrimidine synthesis tracing in platelets

Human donor platelets (Charles River, Platelet-1040) were received on dry ice and thawed at 37 °C with manual agitation. Then, 1 ml of platelet suspension was washed in 9 ml HPLM supplemented with dialysed FBS (10%, GeminiBio, 100-108) and penicillin–streptomycin (100 U ml⁻¹ and 100 µg ml⁻¹, respectively, Thermo Fisher, 15140148), then centrifuged for 10 min at 300$g$ at room temperature. Supernatant was aspirated, then platelets were resuspended in 5 ml unlabelled HPLM. Suspension was centrifuged for 15 min at 600$g$ at room temperature. Supernatant was aspirated, and pellet was resuspended in 100 ml unlabelled HPLM. Platelets were counted using a Vi-CELL XR (Beckman Coulter, RRID: SCR_019664) cell viability analyser and plated ($2 \times 10^6$ platelets per well) in six-well plates in 10 ml HPLM. After 24 h, each well was collected and centrifuged for 10 min at 600$g$ at room temperature, then resuspended in 10 ml HPLM supplemented as above and modified as follows: glutamine exchanged for an equimolar amount of amide-¹⁵N-glutamine (Cambridge Isotope Laboratories, NLM-557), glutamine exchanged for an equimolar amount of amide-¹⁵N-glutamine (Cambridge Isotope Laboratories, NLM-557) and uridine removed, or uridine exchanged for an equimolar amount of ¹⁵N₂ uridine (Cambridge Isotope Laboratories, NLM-812). After incubation with tracer for indicated time points, samples were collected and prepared for LC–MS analysis as described above.

### Immunoprecipitation–mass spectrometry

NHA donor #1 cells expressing pLenti–PGK–Flag–HA–CAD–HA–Hygro were plated in 15-cm dishes ($2.5 \times 10^6$ cells per dish) in 50 ml DMEM (Thermo Fisher, 11995065) supplemented with dialysed FBS (10%, GeminiBio, 100-108), penicillin/streptomycin (100 U ml⁻¹ and 100 µg ml⁻¹, respectively, Thermo Fisher, 15140148) and 3 µM uridine. After 18 h, the medium was changed for 50 ml fresh DMEM supplemented as above with either 0 µM or 3 µM uridine. After 3 h, cells were scraped at 4 °C in Dulbecco's phosphate-buffered saline without calcium and magnesium (Thermo Fisher, 14190250), supplemented either with 0 µM or 3 µM uridine, and lysed in IP lysis buffer (Thermo Fisher, 87787;

25 mM Tris-HCl, pH 7.4, 150 mM NaCl, 1 mM EDTA, 1% NP-40 and 5% glycerol) with Halt Protease and Phosphatase Inhibitor Cocktail (1×, Thermo Fisher, 78440) with rotation for 45 min to 2 h. The lysate was centrifuged at 21,100$g$ at 4 °C for 15 min, and the supernatant was transferred to a fresh tube. The protein concentration was determined by Bradford assay. Then, 100 µl packed anti-Flag M2 affinity gel (Sigma, A2220) per 1 mg lysate was thoroughly resuspended, then centrifuged at 8,000$g$ at 4 °C for 30 s. The supernatant was aspirated, then affinity gel was washed twice with 500 µl of Tris-buffered saline (TBS; 50 mM Tris-HCl, pH 7.5 and 150 mM NaCl), each time centrifuging at 8,000$g$ at 4 °C for 30 s and aspirating supernatant. Protein was added to the washed affinity gel and incubated with rotation at 4 °C for 2 h. Samples were centrifuged at 8,000$g$ at 4 °C for 30 s and the supernatant was stored for analysis as flowthrough. The resin was washed three times with 500 µl of TBS, centrifuging each time at 8,000$g$ at 4 °C for 30 s and aspirating supernatant. Protein was eluted by competition with 100 µl per 1 mg input lysate of 200 ng µl⁻¹ Flag peptide (Sigma F3290) in TBS for 1 h at 4 °C, with rotation. Samples were then centrifuged at 8,000$g$ at 4 °C for 30 s, and supernatant was stored for analysis.

Samples were digested overnight with trypsin (Thermo Fisher, 90057) following reduction and alkylation with dithiothreitol and iodoacetamide (Millipore Sigma, I1149). Following solid-phase extraction cleanup with an Oasis HLB µElution plate (Waters), the resulting peptides were reconstituted in 10 µl of 2% (v/v) acetonitrile and 0.1% trifluoroacetic acid in water. Then 5 µl of this mixture was injected onto an Orbitrap Fusion Lumos mass spectrometer (Thermo Fisher) coupled to an Ultimate 3000 RSLC-Nano liquid chromatography system (Thermo Fisher). Samples were injected onto a 75-µm internal diameter 75-cm long EasySpray column (Thermo Fisher) and eluted with a gradient from 0–28% buffer B over 90 min. Buffer A contained 2% (v/v) acetonitrile and 0.1% formic acid in water, and buffer B contained 80% (v/v) acetonitrile, 10% (v/v) trifluoroethanol and 0.1% formic acid in water. The mass spectrometer operated in positive-ion mode with a source voltage of 2.5 kV and an ion transfer tube temperature of 300 °C. MS scans were acquired at 120,000 resolution in the Orbitrap and up to ten MS/MS spectra were obtained in the Orbitrap for each full spectrum acquired using higher-energy collisional dissociation for ions with charges 2–7. Dynamic exclusion was set for 25 s after an ion was selected for fragmentation.

Raw data files were analysed using Proteome Discoverer v.3.0 (Thermo Fisher), with peptide identification performed using Sequest HT searching against the human reviewed protein database from UniProt (downloaded 4 January 2025; 20,354 entries). Fragment and precursor tolerances of 10 ppm and 0.6 Da were specified, and three missed cleavages were allowed. Carbamidomethylation of Cys was set as a fixed modification, with oxidation of Met and phosphorylation of Ser, Thr and Tyr set as variable modifications. The false discovery rate (FDR) cutoff was 1% for all peptides. Peptide peak intensities were used as peptide abundance values.

### Structure of human CAD

The AlphaFold predicted model[36,77] of full-length human CAD protein (AF-P27708-F1-v4) was downloaded from the AlphaFold Protein Structure Database (https://alphafold.ebi.ac.uk, RRID: SCR_023662) and visualized using PyMol (3.1.1, Schrödinger, RRID: SCR_000305).

### S1900 conservation analysis

The sequence of all proteins with >50% identity to the entry for human CAD (P27708) were downloaded from UniProt[40]. Of the 571 retrieved sequences, the longest isoform sequence available per species was retained to represent orthologue sequences in 289 unique species. The 84 sequences that were shorter than 1,000 amino acids were removed from the analysis to exclude CAD fragments and orthologues that do not contain trifunctional CAD domains. The remaining full 205 sequences, 125 sequences containing class Mammalia in the taxonomy

field, or 80 sequences not containing class Mammalia in the taxonomy field were aligned with Clustal Omega[78] v.1.2.4 (RRID: SCR_001591) implemented in SnapGene v.8.1.1 (GSL Biotech, RRID: SCR_015052). Sequence logos were generated from alignment files with WebLogo[79,80] v.3.9.0 (https://weblogo.threeplusone.com) using probability units, a ±6 amino acid window from human S1900, and the hydrophobicity colour scheme.

### Analysis of nitrogen metabolism profiling platform data

Integrated peaks from LC–MS analysis of nitrogen metabolism profiling platform data were analysed using R (RRID: SCR_001905). Metabolites that were not quantified in more than one sample, had a mean total pool size below an experiment-defined threshold (for AGI-5198 versus DMSO experiments: $2 \times 10^6$, for NHA versus BT054 experiments: $1 \times 10^5$), had calculated total labelling across all conditions of less than 1%, had quantified labelling of >3% in any unlabelled sample, and/or had calculated total labelling across all conditions of greater than 500% were filtered and removed from subsequent analysis. Differential labelling scores were calculated using the formula: $|\ln(\text{fractionalenrichment}_{condition1}/\text{fractionalenrichment}_{condition2})| \times (\text{fractionalenrichment}_{condition1} - \text{fractionalenrichment}_{condition2})$. For visualization, metabolites with less than 5% total labelling were filtered and removed from Sankey diagrams. Sankey diagrams were generated using the R packages ggalluvial[81] (RRID: SCR_021253) and ggplot[82] (RRID: SCR_014601). Metabolites plotted as representative of individual pathways are available in Supplementary Table 1. Sankey diagrams were generated using the R packages ggalluvial[81] (RRID: SCR_021253) and ggplot[82] (RRID: SCR_014601). Metabolites plotted as representative of individual pathways are available in Supplementary Table 1.

### CPTAC proteomic data analysis

For proteomic analysis of human brain and GBM tissues, raw phosphoproteome and proteome mass spectra converted to .mzML open standard format from the Clinical Proteomic Tumour Analysis Consortium (CPTAC) GBM discovery cohort were downloaded from the Proteomic Data Commons (https://pdc.cancer.gov, study IDs PDC000204 and PDC000205, RRID: SCR_018273). FragPipe (RRID: SCR_022864) was used to analyse mass spectra. Peptides were identified using MSFragger[83–85] using a 10 ppm parent ion tolerance, allowing for isotopic error in precursor ion selection, and searched against the UniProt human reference proteome UP000005640 retrieved on 3 December 2024, with reviewed sequences only, with 50% of the database representing decoys, and common contaminants added. Peptide search identified peptide cleavages for Lys-C and strict trypsin digestion, with a maximum of two missed cleavages. For the proteome dataset, carbamidomethylation on cysteine residues, TMT modification on the peptide N terminus and lysine residues, and oxidation on methionine residues were considered. For the phosphoproteome dataset, the previous modifications as well as phosphorylation on serine, threonine and tyrosine were considered. Peptide-to-spectrum matches were scored and validated using MSBooster[86] and Percolator[87]. Proteins were inferred using ProteinProphet[88] and FDR filtering and reporting was performed using Philosopher[89]. A maximum FDR at the peptide level of 1% was used. Protein groups covering 13,007 genes were identified in the proteome dataset and protein groups covering 9,124 genes were identified in the phosphoproteome dataset. Isobaric quantification was performed using TMT Integrator[90], with comparison of each sample to the pooled reference sample labelled with TMT 126 reagent. Phosphopeptide abundances were normalized to the total abundance of each protein.

For LC–MS/MS tandem spectrum generation, files were analysed using Proteome Discoverer (v.3.0, Thermo Fisher, RRID: SCR_014477), with peptide identification performed using Sequest HT searching against the human reviewed protein database from UniProt

(downloaded 4 January 2025; 20,354 entries). Fragment and precursor tolerances of 10 ppm and 0.6 Da were specified, and three missed cleavages were allowed. Carbamidomethylation of Cys was set as a fixed modification, with oxidation of Met and phosphorylation of Ser, Thr and Tyr set as variable modifications. The FDR cutoff was 1% for all peptides. Peptide peak intensities were used as peptide abundance values.

### Mouse proteome data analysis

For proteomic analysis of mouse tissues, C57BL/6N mouse tissue-specific proteome and phosphoproteome data were obtained from Giansanti et al.[91] $\log_{10}$-transformed abundance of tissue-specific total Cad and Cad phosphopeptide levels were transformed to raw peptide intensities. Phosphopeptide levels were normalized to total Cad abundance within each tissue, then $\log_{10}$ transformed. Total Cad, Umps, Dhodh, Uck1, Uck2 and Uprt tissue-specific abundance were plotted as $\log_{10}$-transformed values.

### Other statistical analyses

Information related to data presentation and statistical analysis for individual experiments can be found in the corresponding figure legends. Statistical analyses were carried out using GraphPad Prism (RRID: SCR_002798). No statistical methods were used to pre-determine sample sizes, but our sample sizes are similar to those reported in previous publications[2,4,11,13,76]. For all experiments, $n$ represents biological replicates rather than technical replicates and measurements were taken from distinct samples. Student's unpaired, two-tailed $t$-tests were used for pairwise comparisons assuming normality and without adjustment for multiple comparisons. For each $t$-test, an $F$ test was performed to compare variances between groups. If the variances differed significantly ($F$ test $P < 0.05$), Welch's correction was applied. Data distribution was assumed to be normal, but this was not formally tested. One-way one-way analysis of variance (ANOVA) tests were performed for significance comparison of three or more groups. Linear regression analysis was used to test statistical significance of correlations. To model metabolite labelling over time, curves were fit to labelling data by nonlinear regression analysis using a sigmoidal dose–response (variable slope) model. Statistical significance is indicated as follows: NS, not significant, $*P < 0.05$, $**P < 0.01$, $***P < 0.001$. Error bars represent the s.e.m.

### Reporting summary

Further information on research design is available in the Nature Portfolio Reporting Summary linked to this article.

## Data availability

All data and additional information required to reanalyse the data reported in this paper are available upon request. Further information and requests for resources and reagents should be directed to and will be fulfilled by the corresponding author S.K.M. (samuel.mcbrayer@ utsouthwestern.edu). Plasmids generated in this study are available at Addgene. Metabolomics data are publicly available at the National Metabolomics Data Repository, the Metabolomics Workbench[92], under project accession number PR002945 (ref. 93). RNA sequencing data are publicly available at the Gene Expression Omnibus under accession number GSE320388. Proteomics data are publicly available at the Mass Spectrometry Interactive Virtual Environment under accession number MSV000100431 (ref. 94). Source data are provided with this paper.

## Code availability

Original code is publicly available at Zenodo via https://doi.org/10.5281/zenodo.18736712 (ref. 95).

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

## Acknowledgements

The authors thank McBrayer, Abdullah and DeBerardinis laboratory members, J. Engel, C. Menezes, W. Chen and G. Hoxhaj for helpful discussions; D. Cahill (MGH), H. Kornblum (UCLA), I. Mellinghoff (MSKCC), R. Pieper (UCSF), R. Possemato (NYU), J. Rich (UNC) and S. Weiss (University of Calgary) for sharing cell lines; the UTSW Histo Pathology Core for assistance with H&E staining; the UTSW Animal Resource Center for assistance with mouse studies; and the UTSW Proteomics Core Facility for assistance collecting and interpreting proteomics data.

This study was supported by National Institutes of Health grants R01CA258586 and R01CA289260 to S.K.M. and K.G.A., R01NS142141, R01GM158820, P50CA165962, and U19CA264504 to S.K.M., R35CA220449 to R.J.D., F32HL174084 to T.S.T., K99CA277576 to Y.X. and F30CA271634 to M.R.S. This work was also supported by awards from Oligo Nation to S.K.M. and K.G.A. and by Cancer Prevention and Research Institute of Texas grants RR190034, RP230344 and RP240489 to S.K.M. T.P.M., L.G.Z. and the Children's Research Institute Metabolomics facility are supported by Cancer Prevention and Research Institute of Texas Core Facilities Support Award RP240494. T.S.T. was supported by an American Heart Association Postdoctoral fellowship 24POST1200352. Y.X. was supported by Human Frontier Science Program postdoctoral fellowship LT0018/2022-L. The National Metabolomics Data Repository is supported by National Institutes of Health grants U2C-DK119886 and OT2-OD030544. The funders had no role in study design, data collection and analysis, decision to publish or preparation of the manuscript.

The figures contain adapted vector artwork created by Helicase 11, KeHan, DBCLS and Servier on Bioicons.com, Clker and VLB31 on Pixabay, brgfx on Freepik, OpenClipart and NIAID BioArt.

The authors extend heartfelt gratitude to the patients who generously donated samples, without whom this study would not have been possible.

## Author contributions

Conceptualization: M.R.S., K.G.A. and S.K.M.; data curation: M.R.S., B.L., B.C.S., Y.X., G.B., S.S.O., N.M., S.A.S., M.S.R., W.H.H. and S.A.; formal analysis: M.R.S., B.L., W.G., G.B., A.L. and S.K.M.; funding acquisition: M.R.S., K.G.A. and S.K.M.; investigation: M.R.S., B.L., B.C.S., W.G., Y.X., G.B., T.S., S.S.O., N.M., L.G.Z., V.T.P., S.A.S., M.S.R., M.M.L., C.K.E., W.H.H., S.A., M.S.M.-S., R.W., J.S.P., X.J. and T.S.T.; methodology: M.R.S., B.L., B.C.S., W.G., Y.X., S.S.O., N.M., V.T.P., M.M.L., D.D.S., P.O.Z., A.S. and K.G.A.; project administration: M.R.S., S.S.O., N.M., K.G.A. and S.K.M.; resources: M.R.S., B.L., B.C.S., W.G., Y.X., S.S.O., N.M., V.T.P., M.M.L., D.D.S., P.O.Z., A.S. and K.G.A.; software: M.R.S. and W.G.; supervision: A.S., T.P.M., N.Y.A., R.J.D., K.G.A. and S.K.M.; validation: A.L. and T.E.R.; visualization: M.R.S., B.L., B.C.S., G.B., S.A.S., M.S.R. and S.K.M.; writing – original draft: M.R.S. and S.K.M.; writing – review & editing: M.R.S., B.L., D.D.S., T.E.R., R.J.D., K.G.A. and S.K.M.

## Competing interests

S.K.M. receives research funding from Servier Pharmaceuticals. S.K.M. and K.G.A. have intellectual property interests related to brain tumour metabolism and are co-founders of Gliomet. S.K.M. is a co-founder of Gliomic. R.J.D. is a founder and advisor at Atavistik Bioscience and an advisor for Vida Ventures, Faeth Therapeutics and Illumina. N.Y.R.A. is a consultant for Bruker, serves on the Scientific Advisory Board for National Brain Tumour Society and receives research support from Thermo, EMD Serono and iTeos Therapeutics. T.E.R. has received consulting fees from Servier Pharmaceuticals.

## Additional information

**Extended data** is available for this paper at https://doi.org/10.1038/s42255-026-01520-0.

**Correspondence and requests for materials** should be addressed to Kalil G. Abdullah or Samuel K. McBrayer.

[1]Children's Medical Center Research Institute, University of Texas Southwestern Medical Center, Dallas, TX, USA. [2]Medical Scientist Training Program, University of Texas Southwestern Medical Center, Dallas, TX, USA. [3]Department of Neurosurgery, Brigham and Women's Hospital, Harvard Medical School, Boston, MA, USA. [4]Department of Radiology, Brigham and Women's Hospital, Harvard Medical School, Boston, MA, USA. [5]Department of Neurosurgery, University of Pittsburgh School of Medicine, Pittsburgh, PA, USA. [6]Hillman Comprehensive Cancer Center, University of Pittsburgh Medical Center, Pittsburgh, PA, USA. [7]Department of Radiation Oncology, Dana-Farber/Brigham and Women's Cancer Center, Harvard Medical School, Boston, MA, USA. [8]Department of Medical Oncology, Mass General Brigham, Harvard Medical School, Boston, MA, USA. [9]Department of Biochemistry, University of Texas Southwestern Medical Center, Dallas, TX, USA. [10]Department of Pathology, Molecular and Cell-Based Medicine, Icahn School of Medicine at Mount Sinai, New York, NY, USA. [11]Cecil H. and Ida Green Center for Reproductive Biology Sciences, University of Texas Southwestern Medical Center, Dallas, TX, USA. [12]Department of Obstetrics and Gynecology, University of Texas Southwestern Medical Center, Dallas, TX, USA. [13]Department of Cancer Biology, Dana-Farber Cancer Institute, Harvard Medical School, Boston, MA, USA. [14]Howard Hughes Medical Institute, University of Texas Southwestern Medical Center, Dallas, TX, USA. [15]Department of Pediatrics, University of Texas Southwestern Medical Center, Dallas, TX, USA. [16]The Eugene McDermott Center of Human Growth and Development, University of Texas Southwestern Medical Center, Dallas, TX, USA. [17]Harold C. Simmons Comprehensive Cancer Center, University of Texas Southwestern Medical Center, Dallas, TX, USA. [18]Peter O'Donnell Jr. Brain Institute, University of Texas Southwestern Medical Center, Dallas, TX, USA. ✉e-mail: abdullahkg@upmc.edu; samuel.mcbrayer@utsouthwestern.edu

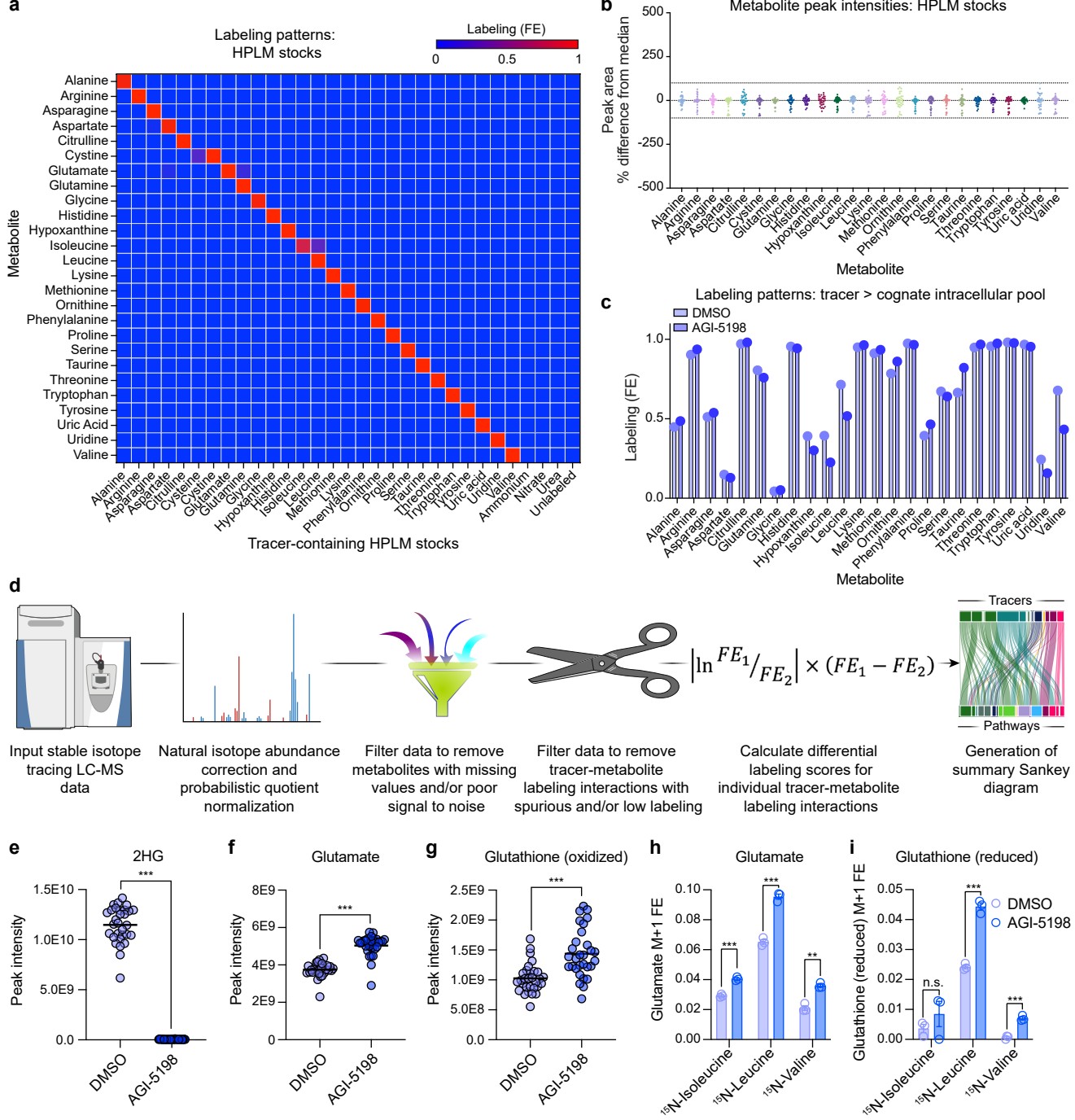

**Extended Data Fig. 1 | Development of a nitrogen metabolism profiling platform and differential labelling score, related to Fig. 1. a,** Heatmap depicting [15]N fractional enrichment (FE) (1-M + 0) in multiple metabolite pools (*y* axis) in each [15]N-labelled HPLM stock (x-axis) (*n* = 1 per tracer). **b,** Total metabolite levels in HPLM stocks (*n* = 30). **c,** FE of cognate intracellular metabolite pool for each tracer in BT054s treated with 3 μM AGI-5198 or DMSO for 72 hours, followed by tracing for 18 hours (*n* = 1 per tracer per treatment). **d,** Schema depicting nitrogen metabolism profiling platform analytical pipeline and derivation of differential labelling score. **e-g,** Peak intensity of (**e**) 2-hydroxyglutarate (2HG)

(*p* < 0.0001), (**f**) glutamate (*p* < 0.0001), or (**g**) oxidized glutathione (*p* < 0.0001) from nitrogen metabolism profiling of BT054s treated with 3 μM AGI-5198 or DMSO for 72 hours, followed by tracing for 18 hours (*n* = 30 per treatment). **h-i,** [15]N FE from indicated tracers in (**h**) glutamate (*p* for isoleucine = 0.0003, *p* for leucine = 0.0002, *p* for valine = 0.0014) or (**i**) reduced glutathione (*p* for leucine = 0.0001, *p* for valine = 0.0005) pools in BT054s treated with 3 μM AGI-5198 or DMSO for 72 hours, followed by tracing for 18 hours (*n* = 3). n.s. = not significant, \*\**p* < 0.01, \*\*\**p* < 0.001. Two-tailed *p* values were determined by unpaired *t*-test. For all panels, data are means ± s.e.m. Credit: icons, Pixabay.com.

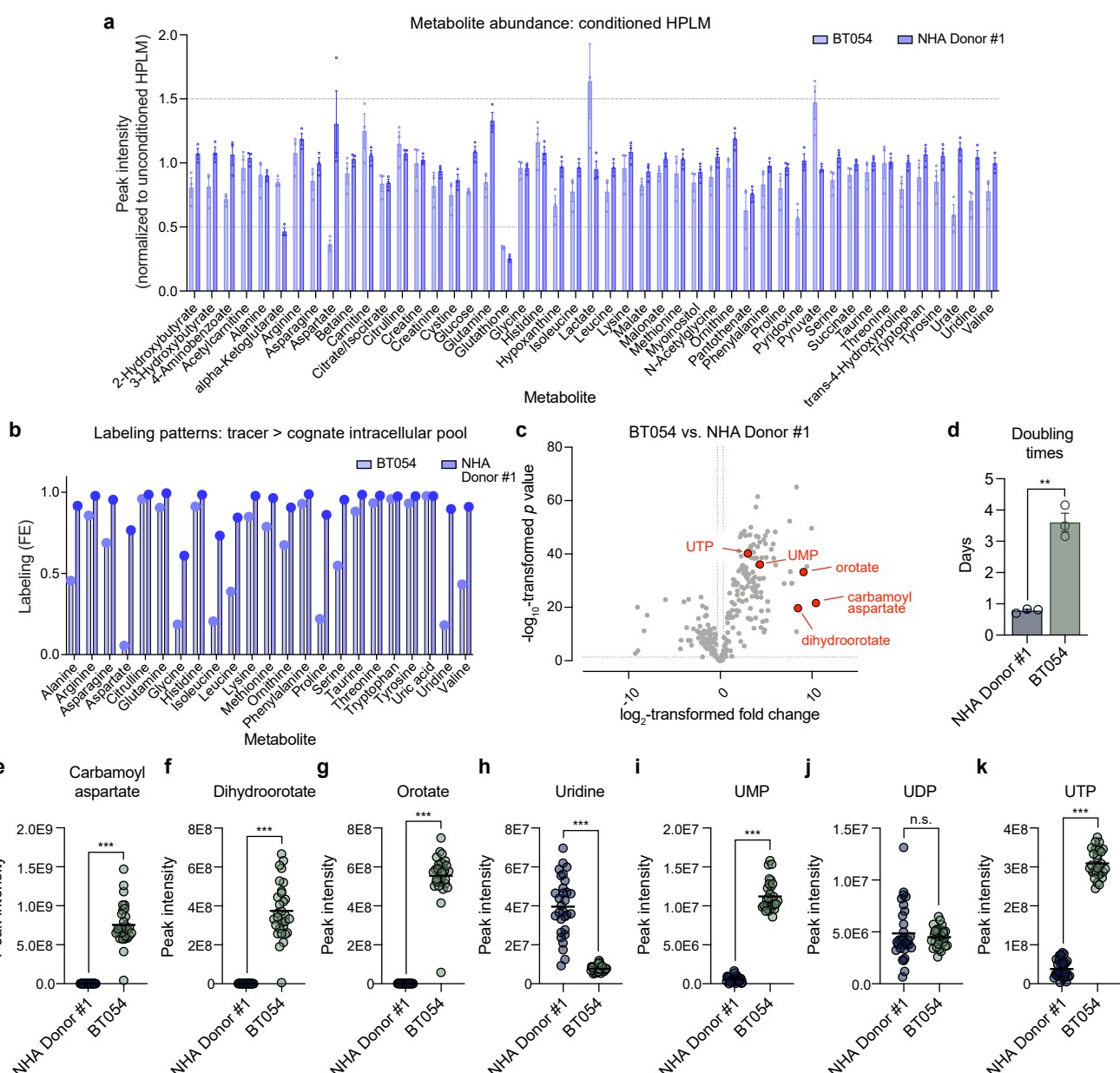

**Extended Data Fig. 2 | Nitrogen metabolism profiling of NHA and BT054, related to Fig. 2. a**, Metabolite levels in conditioned HPLM samples after BT054 or NHA donor #1 cell culture for 18 hours. Data are normalized to unconditioned HPLM ($n = 3$ per cell line). **b**, FE of cognate intracellular metabolite pool for each tracer in BT054 or NHA donor #1 cells at 18 hours ($n = 1$ per tracer per cell line). **c**, Volcano plot of metabolites in BT054 or NHA donor #1 cells ($n = 30$). **d**, Doubling times of NHA donor #1 and BT054 cells ($n = 3$ per cell line,

$p = 0.0096$). Doubling times for BT054 are from Shi et al., 2022[11]. **e-k**, Peak intensity of (**e**) carbamoyl aspartate ($p < 0.0001$), (**f**) dihydroorotate ($p < 0.0001$), (**g**) orotate ($p < 0.0001$), (**h**) uridine ($p < 0.0001$), (**i**) UMP ($p < 0.0001$), (**j**) UDP, (**k**) and UTP ($p < 0.0001$) from nitrogen metabolism profiling of BT054 or NHA donor #1 cells ($n = 30$ per cell line). n.s. = not significant, ***$p < 0.001$. Two-tailed $p$ values were determined by unpaired $t$-test. For all panels, data are means ± s.e.m.

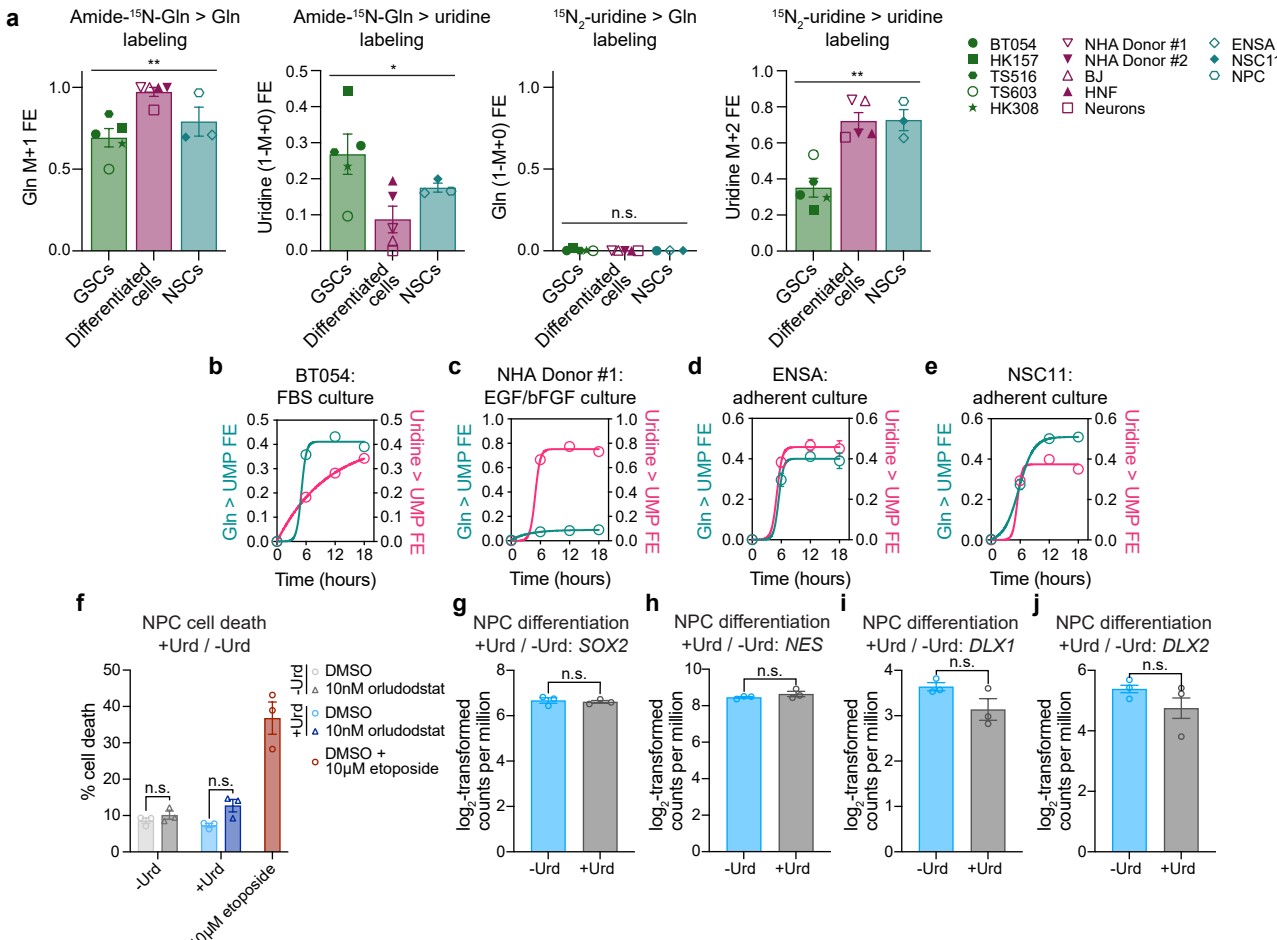

**Extended Data Fig. 3 | Evaluation of pyrimidine synthesis in differentiated and primitive cell cultures, related to Fig. 2. a**, Labelling from indicated tracer to intracellular metabolite pool in GSCs, differentiated cells, and NSCs/NPCs at 18 hours ($n = 3$ for all lines except HK308, for which $n = 2$, $p$ for Amide-$^{15}$N-glutamine (Gln) > Gln = 0.0068, $p$ for Amide-$^{15}$N-Gln > uridine = 0.0296, $p$ for $^{15}$N$_2$-uridine > uridine = 0.0033). **b**, Amide-$^{15}$N-Gln and $^{15}$N$_2$-uridine tracing to UMP in BT054 cells cultured in HPLM with dialysed FBS ($n = 3$). **c**, Amide-$^{15}$N-glutamine and $^{15}$N$_2$-uridine tracing to UMP in NHA donor #1 cells cultured in HPLM without dialysed FBS and with epidermal growth factor (EGF) and basic fibroblast growth factor (bFGF) ($n = 3$). **d-e**, Amide-$^{15}$N-glutamine and $^{15}$N$_2$-uridine tracing to UMP

in (**d**) ENSA and (**e**) NSC11 NSC lines grown in adherent culture conditions. All amide-$^{15}$N-glutamine to UMP labelling data are normalized to glutamine M + 1 FE. **f**, Cell death assay of NPCs in 3 μM or 0 μM uridine treated with 10 nM orludodstat or 10 μM etoposide ($n = 3$). **g-j**, Relative expression of (**g**) SOX2, (**h**) NES, (**i**) DLX1, and (**j**) DLX2 in NPCs undergoing differentiation to neurons at day 10. Data are log$_2$-transformed counts per million mapped reads from RNA sequencing. n.s. = not significant, **$p < 0.01$. In (**a**) $p$ values were determined by one-way ANOVA; in (**f-j**) two-tailed $p$ values were determined by unpaired $t$-test. For all panels, data are means ± s.e.m.

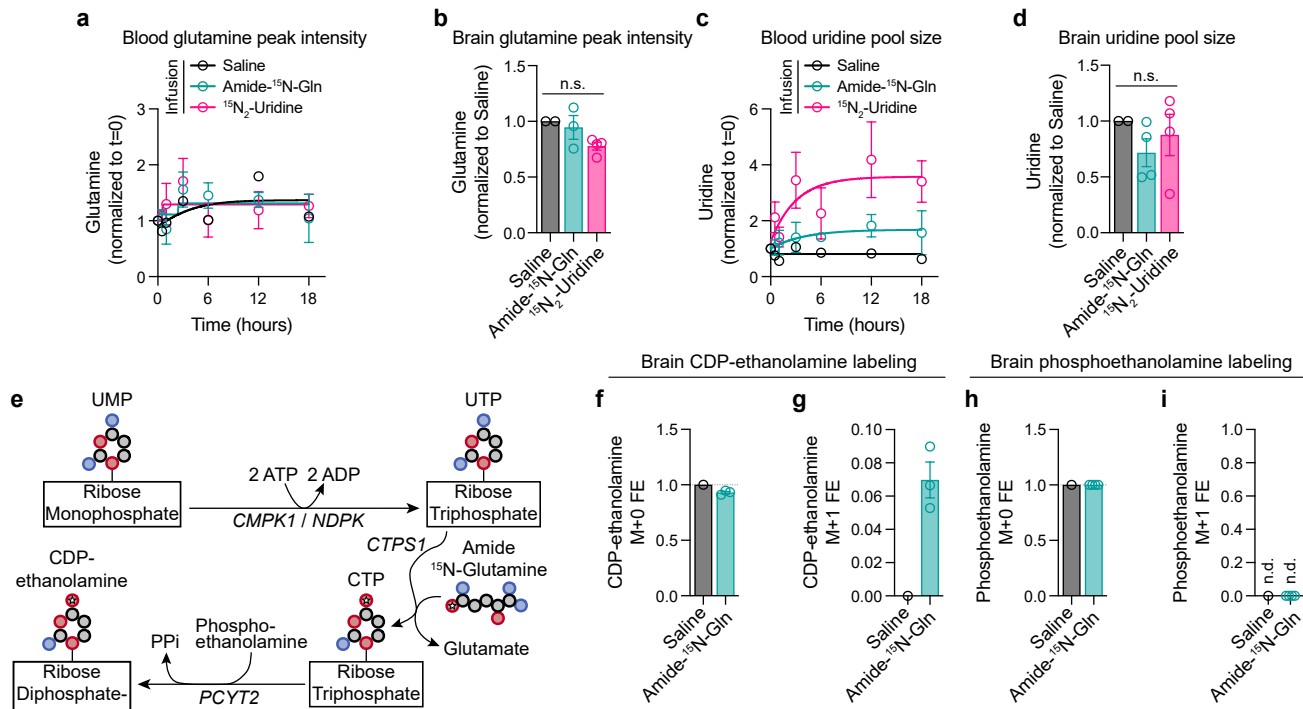

**Extended Data Fig. 4 | Validation of in vivo labelling patterns, related to Fig. 3.**
**a**, Glutamine (Gln) levels in mouse blood ($n = 2$ for saline infusion, $n = 4$ for amide-$^{15}$N-glutamine and $^{15}$N$_2$-uridine infusions). **b**, Glutamine levels in mouse brain after 18 hour infusions ($n = 2$ for saline infusion, $n = 3$ for amide-$^{15}$N-glutamine infusion, $n = 4$ for $^{15}$N$_2$-uridine infusion). **c**, Uridine levels in mouse blood ($n = 2$ for saline infusion, $n = 4$ for amide-$^{15}$N-glutamine and $^{15}$N$_2$-uridine infusions). **d**, Uridine levels in mouse brain after 18 hour infusions ($n = 2$ for saline infusion, $n = 4$ for amide-$^{15}$N-glutamine and $^{15}$N$_2$-uridine infusions). **e**, Schema of cytidine triphosphate synthetase (*CTPS1*)-dependent nitrogen transfer in amide-$^{15}$N-glutamine stable isotope tracing assays. ATP = adenosine triphosphate.

ADP = adenosine diphosphate. *CMPK1* = cytidine/uridine monophosphate kinase 1. *NDPK* = nucleoside-diphosphate kinase. CTP = cytidine triphosphate. *PCYT2* = phosphate cytidylyltransferase 2. CDP-ethanolamine = cytidine diphosphate ethanolamine. **f**, M + 0 and **g**, M + 1 FE in CDP-ethanolamine in mouse brain at 18 hours ($n = 1$ for saline infusion, $n = 3$ for amide-$^{15}$N-glutamine infusion). **h**, M + 0 and **i**, M + 1 FE in phosphoethanolamine in mouse brain at 18 hours ($n = 1$ for saline infusion, $n = 4$ for amide-$^{15}$N-glutamine infusion). n.d. = not detected. n.s. = not significant. *p* values were determined by one-way ANOVA. For all panels, data are means ± s.e.m.

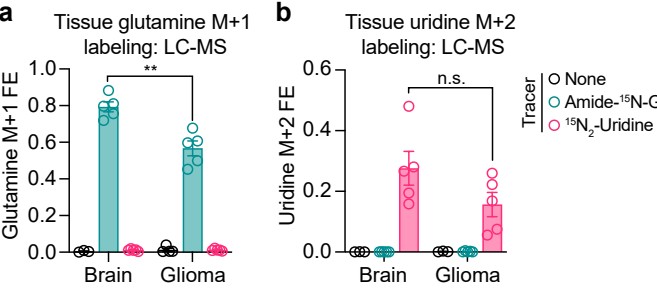

**Extended Data Fig. 5 | Validation of explant labelling patterns, related to Fig. 4. a**, Glutamine (Gln) M + 1 FE in explants incubated in unlabelled ($n = 3$ for brain cohort, $n = 4$ for glioma cohort), amide-$^{15}$N-Gln ($n = 5$ per cohort), or $^{15}$N$_2$-uridine ($n = 5$ per cohort) HPLM for 18 hours ($p = 0.0016$). **b**, Uridine

M + 2 FE in explants incubated in unlabelled ($n = 3$ for brain cohort, $n = 4$ for glioma cohort), amide-$^{15}$N-Gln ($n = 5$ per cohort), or $^{15}$N$_2$-uridine ($n = 5$ per cohort) HPLM for 18 hours. n.s. = not significant, **$p < 0.01$. Two-tailed $p$ values were determined by unpaired $t$-test. For all panels, data are means ± s.e.m.

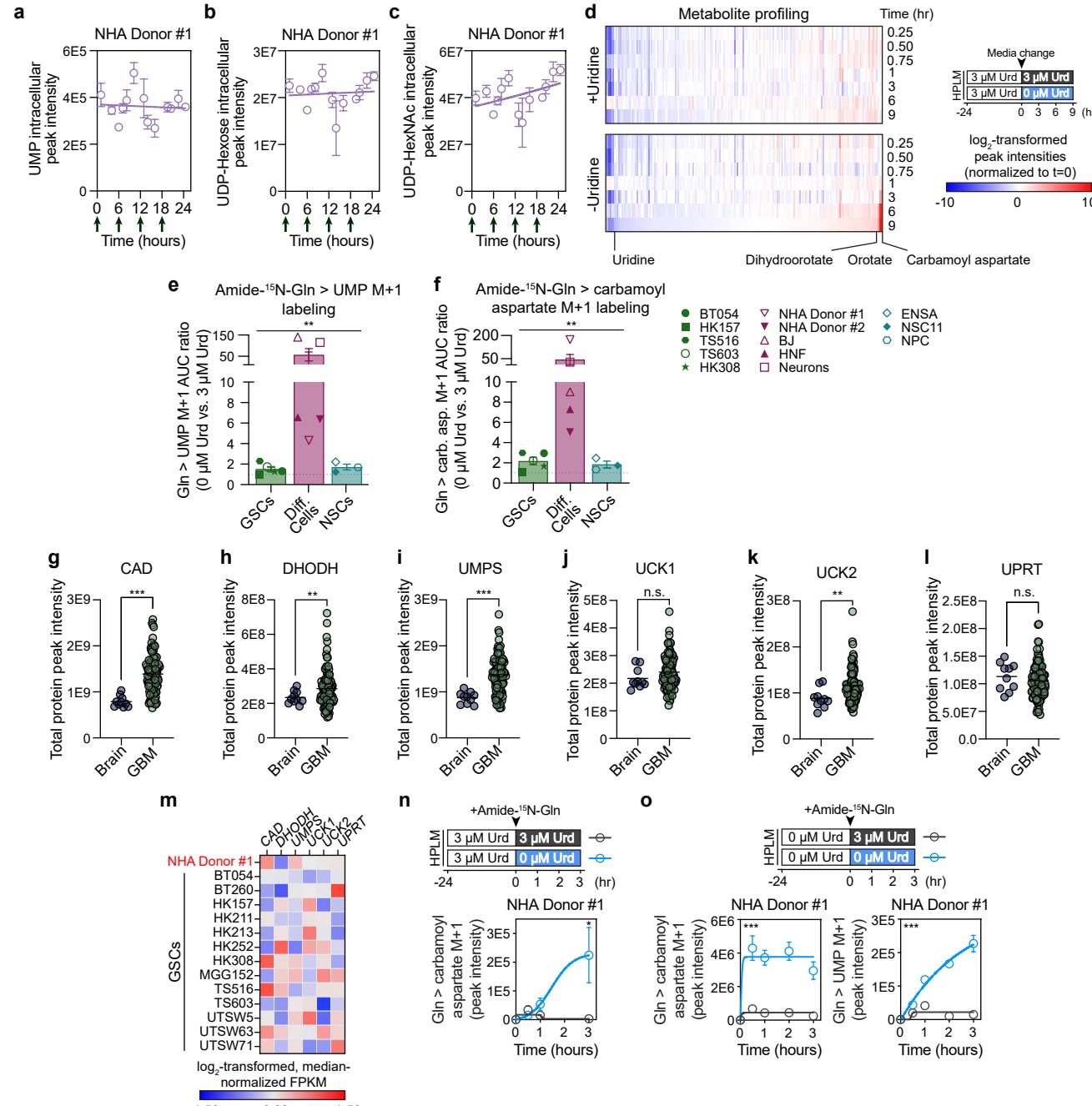

**Extended Data Fig. 6 | Uridine deprivation acutely activates de novo pyrimidine synthesis in differentiated cells, related to Fig. 5. a-c,** Quantification of (**a**) UMP, (**b**) UDP-hexose, and (**c**) uridine diphosphate-N-acetylhexosamine (UDP-HexNAc) levels in NHA donor #1 cells cultured in DMEM with 3 μM uridine. Media was changed at timepoints indicated by arrows. Curves are fit for full 24-hour experiment (n = 3). **d,** Heatmap of metabolite levels in NHA donor #1 cells cultured in 0 μM or 3 μM uridine for 0.25 to 9 hours after 24 hours of culture in 3 μM uridine (n = 6 per group). Data are log₂-transformed peak intensities normalized to 0-hour timepoint for each metabolite. **e-f,** Ratio of area under the curve of (**e**) UMP M + 1 (p = 0.0039) or (**f**) carbamoyl aspartate M + 1 (p = 0.0033) after 18-hour amide-¹⁵N-glutamine (Gln) tracing in 0 μM versus 3 μM uridine. **g-l,** Protein expression of (**g**) CAD (p < 0.0001), (**h**) DHODH (p = 0.0029), (**i**) UMPS (p < 0.0001), (**j**) UCK1, (**k**) UCK2 (p = 0.0023), and (**l**) UPRT in human brain and GBM samples (n = 10 brain, n = 99 GBM). Data are results of reanalysis

of raw data published in Wang et al.[35]. **m,** Heatmap depicting relative abundance of indicated transcripts in a panel of GSCs and NHA donor #1 cells. Data are log₂-transformed, median-normalized fragments per kilobase of transcript per million mapped reads (FPKM) from RNA sequencing and are reanalysis of data published in Wu et al[17]. and Savani et al.[25]. **n,** Carbamoyl aspartate M + 1 peak intensity after amide-¹⁵N-glutamine tracing in NHA donor #1 cells cultured in 0 μM (blue) or 3 μM (grey) uridine (n = 6 per group). **o,** Amide-¹⁵N-glutamine stable isotope tracing in NHA donor #1 cells cultured in 0 μM or 3 μM uridine after 24 hours of culture with 0 μM uridine (n = 3 per group). n.s. = not significant, *p < 0.05, **p < 0.01, ***p < 0.001. In (**e-f**) p values were determined by one-way ANOVA; in (**g-l, n-o**) two-tailed p values were determined by unpaired t-test. t-tests compare (**n**) 3-hour timepoint (p = 0.0456) or (**o**) area under the curve values (p for carbamoyl aspartate M + 1 < 0.0001, p for UMP M + 1 < 0.0001) for 0 μM uridine and 3 μM uridine conditions. For all panels, data are means ± s.e.m.

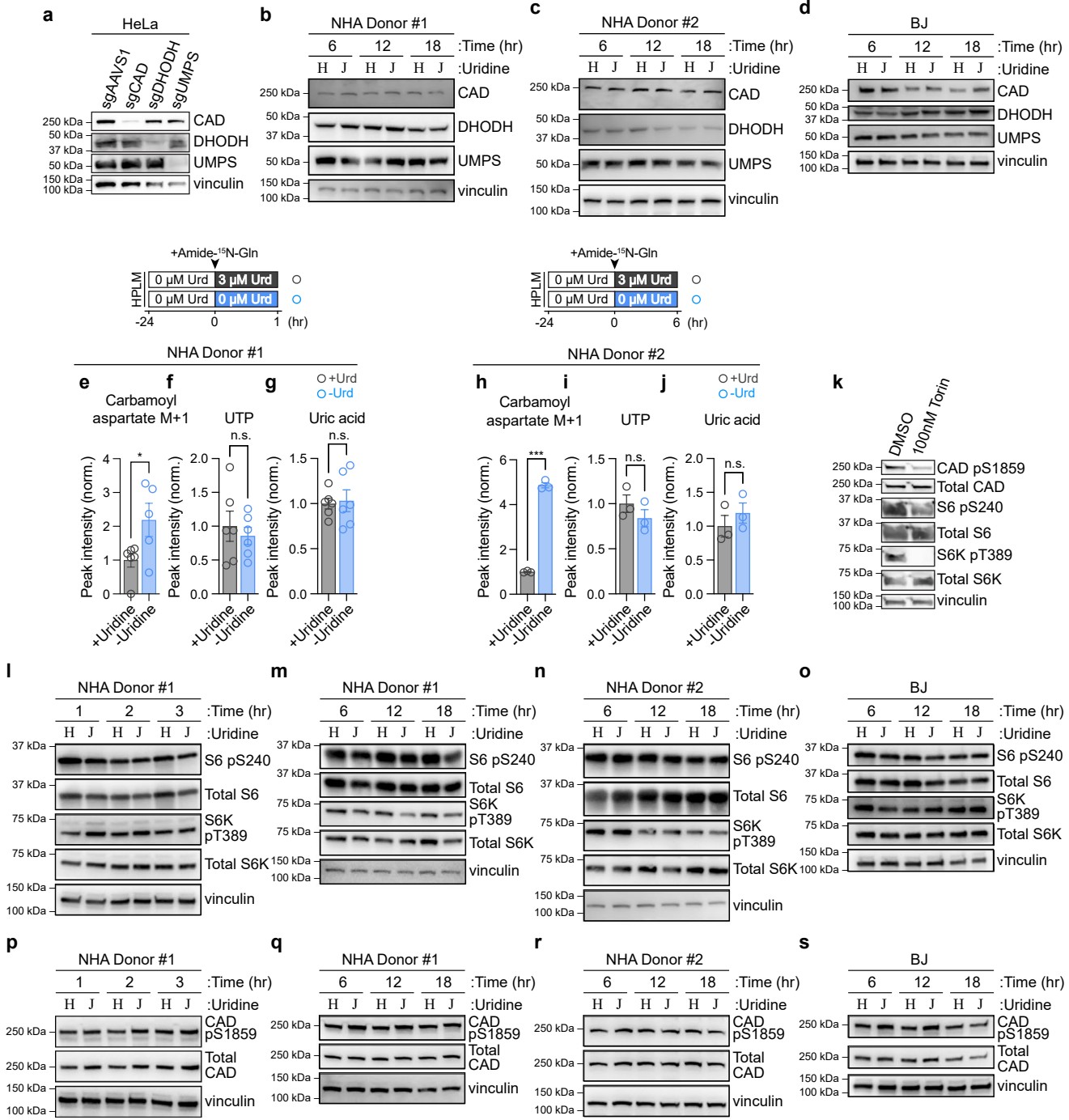

**Extended Data Fig. 7 | Uridine deprivation does not impact known de novo pyrimidine synthesis regulatory mechanisms, related to Fig. 5. a**, Immunoblot for CAD, DHODH, and UMPS in HeLa cells expressing Cas9 and *AAVS1* safe harbour locus control, *CAD*, *DHODH*, or *UMPS* sgRNAs ($n = 1$). **b-d**, Representative immunoblots for CAD, DHODH, and UMPS in (**b**) NHA donor #1, (**c**) NHA donor #2, or (**d**) BJ cells cultured in 0 μM or 3 μM uridine for indicated times after 24 hours of culture in 3 μM uridine ($n = 3$). **e-g**, Normalized peak intensities of (**e**) carbamoyl aspartate M + 1 ($p = 0.0403$), (**f**) UTP, or (**g**) uric acid in NHA donor #1 cells cultured in 0 μM (blue) or 3 μM (grey) uridine for 1 hour after 24 hours of culture in 3 μM uridine ($n = 6$). **h-j**, Normalized peak intensities of (**h**) carbamoyl aspartate M + 1 ($p < 0.0001$), (**i**) UTP, or (**j**) uric acid in NHA donor #2 cells cultured in 0 μM (blue) or 3 μM (grey) uridine for 6 hours after 24 hours of culture in 3 μM uridine ($n = 3$). **k**, Immunoblots for S1859 phosphorylation site on CAD,

total CAD, the S240/S244 phosphorylation site of S6 ribosomal protein, total S6 ribosomal protein, the T389 phosphorylation site on S6 kinase (S6K), and total S6K in NHA donor #1 cells treated for 1 hour with DMSO or 100 μM torin ($n = 1$). **l-o**, Representative immunoblots for S240/S244 phosphorylation site of S6 ribosomal protein, total S6 ribosomal protein, the T389 phosphorylation site on S6K, and total S6K in (**l-m**) NHA donor #1, (**n**) NHA donor #2, or (**o**) BJ cells cultured in 0 μM or 3 μM uridine for indicated times after 24 hours of culture in 3 μM uridine ($n = 3$). **p-s**, Representative immunoblots for S1859 phosphorylation site on CAD and total CAD in (**p-q**) NHA donor #1, (**r**) NHA donor #2, or (**s**) BJ cells cultured in 0 μM or 3 μM uridine for indicated times after 24 hours of culture in 3 μM uridine ($n = 3$). n.s. = not significant, \*$p < 0.05$, \*\*\*$p < 0.001$. Two-tailed $p$ values were determined by unpaired $t$-test. For all panels, data are means ± s.e.m.

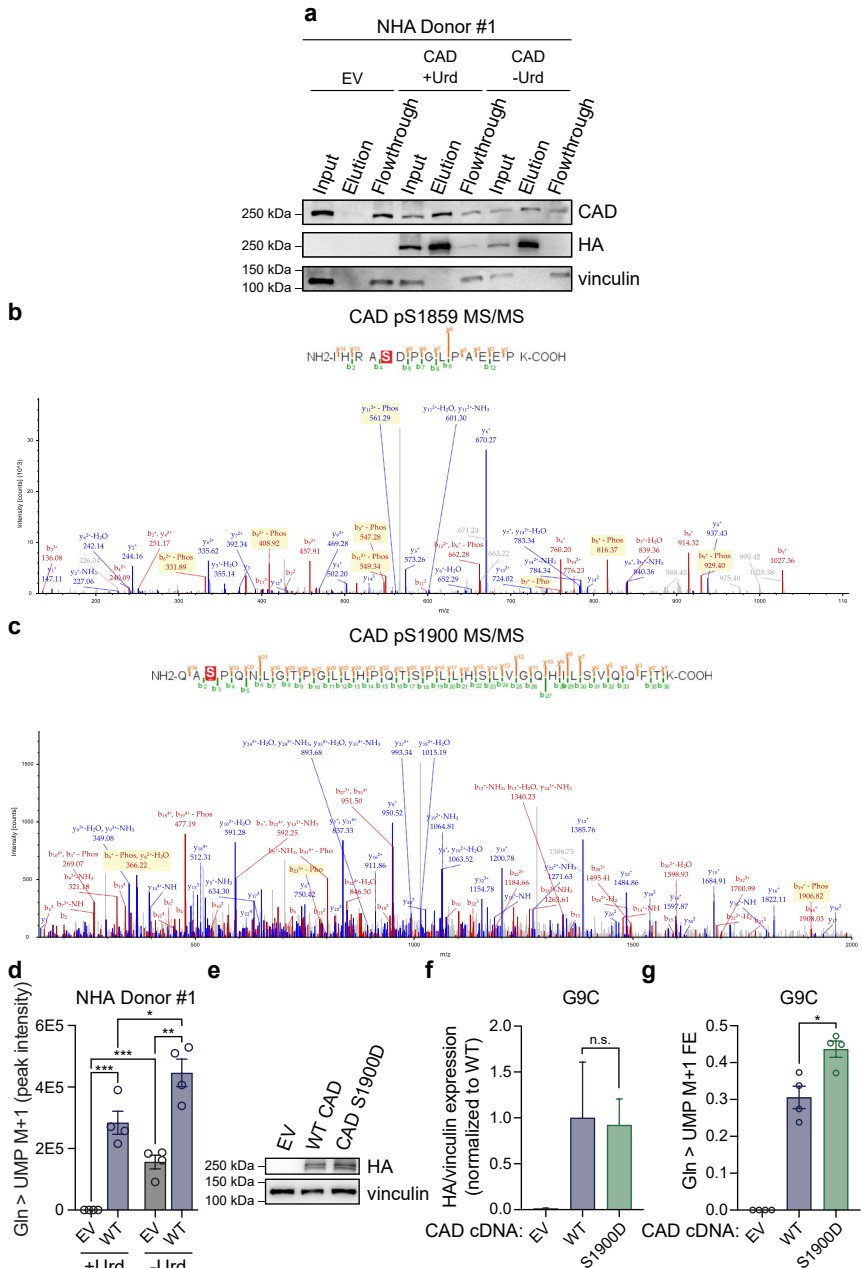

**Extended Data Fig. 8 | CAD S1900 phosphorylation is sufficient to induce de novo pyrimidine synthesis, related to Fig. 6. a**, Immunoblot of FLAG immunoprecipitation of NHA donor #1 cells expressing FLAG-tagged CAD or empty vector (EV) cultured in DMEM with 0 µM or 3 µM uridine (Urd) for 3 hours after 18 hours of culture in DMEM with 3 µM uridine (Urd, n = 3 per group). **b-c**, LC–MS/MS tandem mass spectrum for the peptides (**b**) IHRApSDPGLPAEEPK, corresponding to phosphorylation at S1859 or (**c**) QApSPQNLGTPGLLHPQTSPLLHSLVGQHILSVQQFTK, corresponding to phosphorylation at S1900, of full-length CAD protein from IP–MS experiment with NHA donor #1 cells. **d**, Amide-$^{15}$N-glutamine stable isotope tracing for 18 hours in HPLM with 0 µM or 3 µM uridine in NHA donor #1 cells expressing either EV or WT CAD. Data depict peak intensity of the M + 1 isotopologue of UMP

(n = 4 per group, p for EV +Urd-WT +Urd = 0.0003, p for EV +Urd-EV -Urd = 0.0004, p for WT +Urd-WT -Urd = 0.0318, p for EV -Urd-WT -Urd = 0.0011). **e**, Immunoblot for HA tag on exogenous CAD enzymes in G9C CHO cells expressing empty vector (EV), wild-type (WT) CAD, or CAD S1900D (n = 4 per line). **f**, Densitometry quantifying relative HA expression in G9C CHO cells expressing either EV, WT CAD, or CAD S1900D (n = 4 per line). Data are normalized to WT CAD. **g**, Amide-$^{15}$N-glutamine (Gln) stable isotope tracing for 18 hours in HPLM with 0 µM or 3 µM uridine in G9C CHO cells expressing either EV, WT CAD, CAD S1900D. Data depict fractional enrichment (FE) in the M + 1 isotopologue of UMP (n = 4 per group, p = 0.0133). n.s. = not significant, *p < 0.05, **p < 0.01, ***p < 0.001. Two-tailed p values were determined by unpaired t-test. For all panels, data are means ± s.e.m.

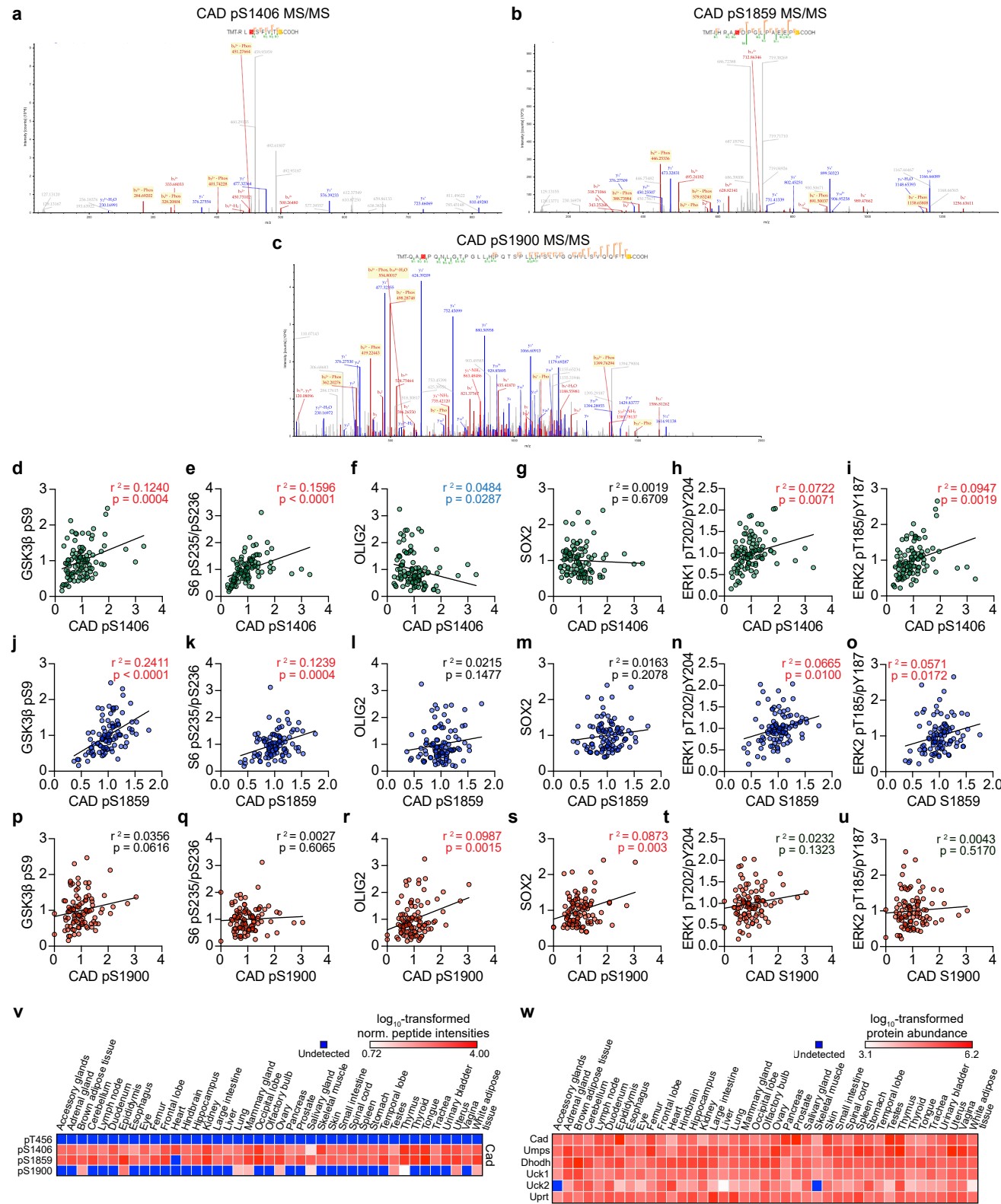

**Extended Data Fig. 9 | See next page for caption.**

**Extended Data Fig. 9 | CAD S1900 phosphorylation is enriched in primitive cell populations, related to Fig. 7. a-c**, LC–MS/MS tandem mass spectrum for the peptides (**a**) RLpSSFVTK, corresponding to phosphorylation at S1406, (**b**) IHRApSDPGLPAEEPK, corresponding to phosphorylation at S1859, or (**c**) QApSPQNLGTPGLLHPQTSPLLHSLVGQHILSVQQFTK, corresponding to phosphorylation at S1900, of full-length CAD protein from human brain and GBM samples. **d-u**, Correlation between (**d-i**) S1406, (**j-o**) S1859, and (**p-u**) S1900 phosphorylation of CAD and (**d,j,p**) GSK3β phosphorylation at S9, (**e, k, q**) S6 ribosomal protein phosphorylation at S235/S236, (**f, l, r**) OLIG2 protein levels, (**g, m, s**) SOX2 protein levels, (**h, n, t**) ERK1 phosphorylation at T202/Y204, or (**i, o, u**) ERK2 phosphorylation at T185/Y187. Phosphorylation data are normalized to total protein and are mean normalized. Correlation coefficients and $p$ values in red indicate statistically significant positive correlation, blue indicate statistically significant negative correlation, and black indicate lack of significant correlation ($n = 99$). $r$ = Pearson's correlation coefficient. $p$ values were determined by simple linear regression analysis. Panels (**a-u**) are results of reanalysis of raw data published in Wang et al.[35]. **v**, Heatmaps depicting relative phosphorylation of residues analogous to T456, S1406, S1859, and S1900 sites on human CAD protein in mouse tissues. Data are $\log_{10}$-transformed peptide intensities normalized to total CAD protein. **w**, Heatmaps depicting protein expression of Cad, Umps, Dhodh, Uck1, Uck2, and Uprt in mouse tissues. Data are $\log_{10}$-transformed protein abundance in each tissue. Panels (**v-w**) are results of reanalysis of data published in Giansanti et al.[91].

# Reporting Summary

## Statistics

For all statistical analyses, confirm that the following items are present in the figure legend, table legend, main text, or Methods section.

| n/a | Confirmed | |
|---|---|---|
| ☐ | ☒ | The exact sample size (*n*) for each experimental group/condition, given as a discrete number and unit of measurement |
| ☐ | ☒ | A statement on whether measurements were taken from distinct samples or whether the same sample was measured repeatedly |
| ☐ | ☒ | The statistical test(s) used AND whether they are one- or two-sided<br>*Only common tests should be described solely by name; describe more complex techniques in the Methods section.* |
| ☒ | ☐ | A description of all covariates tested |
| ☐ | ☒ | A description of any assumptions or corrections, such as tests of normality and adjustment for multiple comparisons |
| ☐ | ☒ | A full description of the statistical parameters including central tendency (e.g. means) or other basic estimates (e.g. regression coefficient) AND variation (e.g. standard deviation) or associated estimates of uncertainty (e.g. confidence intervals) |
| ☐ | ☒ | For null hypothesis testing, the test statistic (e.g. *F*, *t*, *r*) with confidence intervals, effect sizes, degrees of freedom and *P* value noted<br>*Give P values as exact values whenever suitable.* |
| ☒ | ☐ | For Bayesian analysis, information on the choice of priors and Markov chain Monte Carlo settings |
| ☒ | ☐ | For hierarchical and complex designs, identification of the appropriate level for tests and full reporting of outcomes |
| ☒ | ☐ | Estimates of effect sizes (e.g. Cohen's *d*, Pearson's *r*), indicating how they were calculated |

*Our web collection on statistics for biologists contains articles on many of the points above.*

## Software and code

Policy information about availability of computer code

| Data collection | TraceFinder (5.1 SP2), SCiLS Lab (2023a Core), Brainlab Automatic Image Registration |
|---|---|
| Data analysis | El-MAVEN (0.12.0), TraceFinder (5.1 SP2), R (4.1.1 or 4.4.2), RStudio (2024.09), AccuCor (0.3.0), SCiLS Lab (2023a Core), rMSIproc (0.3.166), enviPat (2.767), GraphPad Prism (10.4.1), PyMol (3.1.1), Clustal Omega (1.2.4), SnapGene (8.1.1), WebLogo 3 (3.9.0), ggalluvial (0.12.5), ggplot2 (3.4.2), FragPipe, Proteome Discoverer (3.0), Fiji/ImageJ2 (1.54p), ELDA (Last Modified October 24, 2014) |

For manuscripts utilizing custom algorithms or software that are central to the research but not yet described in published literature, software must be made available to editors and reviewers. We strongly encourage code deposition in a community repository (e.g. GitHub). See the Nature Portfolio guidelines for submitting code & software for further information.

## Data

Policy information about availability of data

All manuscripts must include a data availability statement. This statement should provide the following information, where applicable:
- Accession codes, unique identifiers, or web links for publicly available datasets
- A description of any restrictions on data availability
- For clinical datasets or third party data, please ensure that the statement adheres to our policy

All data and additional information required to reanalyze the data reported in this paper are available upon request. Further information and requests for resources and reagents should be directed to and will be fulfilled by the corresponding author Samuel K. McBrayer (samuel.mcbrayer@utsouthwestern.edu).

# Research involving human participants, their data, or biological material

Policy information about studies with **human participants or human data**. See also policy information about **sex, gender (identity/presentation), and sexual orientation** and **race, ethnicity and racism**.

| | |
|---|---|
| Reporting on sex and gender | Sex of human participants in explant stable isotope tracing studies are reported in Supplementary Table 4. |
| Reporting on race, ethnicity, or other socially relevant groupings | Race, ethnicity, and other socially relevant groupings are not reported for explant stable isotope tracing studies. |
| Population characteristics | Age, brain regions, and molecular features of human samples in explant stable isotope tracing studies are reported in Supplementary Table 4. |
| Recruitment | Participants in explant stable isotope tracing experiments were recruited from patients undergoing neurosurgical procedures at the University of Pittsburgh Medical Center. |
| Ethics oversight | Patient tissue and blood were collected following ethical and technical guidelines on the use of human samples for biomedical research at the University of Pittsburgh Medical Center after informed patient consent under a protocol approved by the University of Pittsburgh Medical Center's Institutional Review Board. |

Note that full information on the approval of the study protocol must also be provided in the manuscript.

# Field-specific reporting

Please select the one below that is the best fit for your research. If you are not sure, read the appropriate sections before making your selection.

☒ Life sciences ☐ Behavioural & social sciences ☐ Ecological, evolutionary & environmental sciences

For a reference copy of the document with all sections, see nature.com/documents/nr-reporting-summary-flat.pdf

# Life sciences study design

All studies must disclose on these points even when the disclosure is negative.

| | |
|---|---|
| Sample size | Sample size was determined based on availability of samples and a minimum of three biological replicates to ensure sufficient statistical power. |
| Data exclusions | No data were excluded. |
| Replication | Multiple biologic replicates were performed for each experiment (as noted in each figure legend). |
| Randomization | All human explant tissues, cell cultures, and mice were assigned to experimental groups randomly. |
| Blinding | Investigators were not blinded during sample collection or analysis. Sample processing was equivalent for all samples. |

# Reporting for specific materials, systems and methods

We require information from authors about some types of materials, experimental systems and methods used in many studies. Here, indicate whether each material, system or method listed is relevant to your study. If you are not sure if a list item applies to your research, read the appropriate section before selecting a response.

## Materials & experimental systems

| n/a | Involved in the study |
|---|---|
| ☐ | ☒ Antibodies |
| ☐ | ☒ Eukaryotic cell lines |
| ☒ | ☐ Palaeontology and archaeology |
| ☐ | ☒ Animals and other organisms |
| ☒ | ☐ Clinical data |
| ☒ | ☐ Dual use research of concern |
| ☒ | ☐ Plants |

## Methods

| n/a | Involved in the study |
|---|---|
| ☒ | ☐ ChIP-seq |
| ☒ | ☐ Flow cytometry |
| ☒ | ☐ MRI-based neuroimaging |

# Antibodies

| | |
|---|---|
| Antibodies used | Primary antibodies used included: anti-IDH1 R132H (Dianova DIA-H09, 1:500, Mouse monoclonal, RRID: AB_2335716), anti-vinculin (Sigma V9131, 1:100,000, mouse monoclonal, RRID:AB_477629), anti-Phospho-CAD (Ser1859) (Cell Signaling Technologies 70307, 1:1,000, rabbit monoclonal, RRID: AB_2799782), anti-CAD (Cell Signaling Technologies 11933, 1:1,000, rabbit polyclonal, RRID: AB_2797772), anti-DHODH (Proteintech 14877, 1:1,000, rabbit polyclonal, RRID: AB_2091723), anti-UMPS (Millipore Sigma HPA036179, 1:1,000, rabbit polyclonal, RRID: AB_10673615), anti-UMPS (Proteintech 14830, 1:1,000, rabbit polyclonal, RRID: AB_2212392), anti-HA (Thermo Fisher 26183, 1:1,000, mouse monoclonal, RRID: AB_10978021), anti-HA (BioLegend 901513, 1:1,000, mouse monoclonal, RRID: AB_2565335), anti-FLAG (Millipore Sigma F1804, 1:1,000, mouse monoclonal, RRID: AB_262044), anti-Phospho-S6 ribosomal protein (Ser240/244) (Cell Signaling Technologies 5364, 1:1,000, rabbit monoclonal, RRID: AB_10694233), anti-S6 ribosomal protein (Cell Signaling Technologies 2217, 1:1,000, rabbit monoclonal, RRID: AB_331355), anti-Phospho-p70 S6 kinase (Thr389) (Cell Signaling Technologies 9234, 1:1,000, rabbit monoclonal, RRID: AB_2269803), and anti-p70 S6 kinase (Cell Signaling Technologies 2708, 1:1,000, rabbit monoclonal, RRID: AB_390722). HRP-conjugated secondary antibodies used included: anti-Mouse IgG (Thermo Fisher 31430, 1:2,000, goat polyclonal, RRID: AB_228307) and anti-Rabbit IgG (Thermo Fisher 31460, 1:2,000, goat polyclonal, RRID: AB_228341). |
| Validation | anti-IDH1 R132H antibody was validated in Figure 1c<br>anti-CAD, anti-DHODH, and anti-UMPS antibodies were validated in Extended Data Figure 7a<br>anti-Phospho-CAD (Ser1859), anti-Phospho-S6 ribosomal protein (Ser240/244), anti-S6 ribosomal protein, anti-Phospho-p70 S6 kinase (Thr389), and anti-p70 S6 kinase were validated in Extended Data Figure 7k<br>anti-vinculin, anti-FLAG, and anti-HA antibodies were validated by manufacturer by Western blot analysis |

# Eukaryotic cell lines

Policy information about cell lines and Sex and Gender in Research

| | |
|---|---|
| Cell line source(s) | NHA Donor #1 cells (human astrocytes immortalized with HPV E6 and E7 and hTERT, sex unknown) were obtained from R. Pieper at the University of California San Francisco. NHA Donor #2 cells were generated from commercially obtained primary human astrocytes (Lonza CC-3187). HEK293T (female, ATCC CRL-3216, RRID: CVCL_0063), BJ (male, ATCC CRL-2522, RRID: CVCL_3653), HNF primary fibroblast (Lonza NHDF-Neo), and HeLa (female, Millipore Sigma 93021013, RRID: CVCL_0030) cells were obtained commercially. G9C CHO cells were a gift of R. Possemato at New York University. BT054 (female, RRID: CVCL_N707) cells were obtained from S. Weiss at the University of Calgary. HK157 (female) and HK308 (male) cells were obtained from H. Kornblum at the University of California Los Angeles. TS516 (sex unknown, RRID: CVCL_A5HY) and TS603 (sex unknown, RRID: CVCL_A5HW) cells were obtained from I. Mellinghoff at Memorial Sloan-Kettering Cancer Center. ENSA (sex unknown) and NSC11 (sex unknown) cells were obtained from J. Rich at University of North Carolina-Chapel Hill. MGG152 (male) cells were obtained from D. Cahill at Massachusetts General Hospital. Murine GSC lines PC948-1 and PIC144-1 were derived from autochthonous astrocytomas that formed in a modified version of a genetically engineered mouse model. |
| Authentication | Cell line authentication was not performed because reference short term tandem repeat profiles have not been established for these lines. |
| Mycoplasma contamination | All cell lines were routinely evaluated for mycoplasma contamination with the e-Myco Mycoplasma PCR Detection Kit (Bulldog Bio 2523348), e-Myco PLUS Mycoplasma PCR Detection Kit (Bulldog Bio 25233), or MycoAlert Mycoplasma Detection Kit (Lonza LT07-318) and confirmed to be negative. |
| Commonly misidentified lines<br>(See ICLAC register) | None. |

# Animals and other research organisms

Policy information about studies involving animals; ARRIVE guidelines recommended for reporting animal research, and Sex and Gender in Research

| | |
|---|---|
| Laboratory animals | Animal welfare assessments were carried out daily during treatment periods. Animals were housed in a pathogen-free environment between 20-26°C and at 30-70% humidity, with a 12 hour:12 hour light:dark cycle. Fox Chase SCID (Charles River 236, RRID: IMSR_CRL:236) mice pre-catheterized in the jugular vein with a one-channel 25-gauge vascular access button (Instech Laboratories VABM1B/25) were obtained from Charles River Laboratories at 8-10 weeks of age. C57BL/6J (RRID: IMSR_JAX:000664) mice were |

obtained from the UT Southwestern Mouse Breeding Core or Jackson Laboratories. Mice were housed together (2-5 mice of the same sex per cage) and provided free access to chow diet (Teklad 2916) and water.

Wild animals

No wild animals were used.

Reporting on sex

Mice of both male and female sexes were used and proportions of mouse sexes used in infusion experiments are reported in the Methods.

Field-collected samples

No field-collected samples were used.

Ethics oversight

All care and treatment of experimental animals were carried out in strict accordance with Good Animal Practice as defined by the US Office of Laboratory Animal Welfare and approved by the UT Southwestern Medical Center (protocols 2017-101840, 2019-102795, and 2022-102897) Institutional Animal Care and Use Committee.

Note that full information on the approval of the study protocol must also be provided in the manuscript.

# Plants

Seed stocks

*Report on the source of all seed stocks or other plant material used. If applicable, state the seed stock centre and catalogue number. If plant specimens were collected from the field, describe the collection location, date and sampling procedures.*

Novel plant genotypes

*Describe the methods by which all novel plant genotypes were produced. This includes those generated by transgenic approaches, gene editing, chemical/radiation-based mutagenesis and hybridization. For transgenic lines, describe the transformation method, the number of independent lines analyzed and the generation upon which experiments were performed. For gene-edited lines, describe the editor used, the endogenous sequence targeted for editing, the targeting guide RNA sequence (if applicable) and how the editor was applied.*

Authentication

*Describe any authentication procedures for each seed stock used or novel genotype generated. Describe any experiments used to assess the effect of a mutation and, where applicable, how potential secondary effects (e.g. second site T-DNA insertions, mosiacism, off-target gene editing) were examined.*

