## [Peer Review File · Nature Metabolism]

Nitrogen metabolism profiling reveals cell state-specific pyrimidine synthesis pathway choice

Corresponding Author: Dr Samuel McBrayer

Version 0:

Reviewer comments:

Reviewer #1

(Remarks to the Author)

This manuscript comprehensively explores nitrogen metabolism in human cells. They begin with a multiplexed tracer technique allowing for simultaneous analysis of 30 nitrogen sources in cell culture. There is nice validation of this comprehensive multiplexed analysis technique based on this group's prior work showing that 2HG can inhibit BCAA-driven glutamate (and glutathione) synthesis. The comprehensive nature of this multiplexed ^{15}N tracing technique allows validation with several different tracers simultaneously. Applying this technique to other cell types and conditions allows them to find a striking difference in how pyrimidines are synthesized in cancerous/primitive/stem-like cells (which synthesize pyrimidines both de novo and from salvage) and differentiated cells which synthesize pyrimidines almost entirely from salvage. The authors then validate their findings in human samples. In flash-frozen tissue, the de novo intermediate Carbamoyl aspartate is elevated in tumor, while the salvage substrate uridine is unchanged. Then in primary explants, GLN-driven UMP synthesis is elevated in glioma explants, while uridine-driven UMP synthesis is identical in glioma and brain. They then sought to understand the mechanism behind the preference of pyrimidine salvage in differentiated cells and de novo synthesis in more primitive cell types. They noted that uridine depletion caused an accumulation of intermediates in de novo synthesis and increased de novo synthesis, but only in differentiated cells. Thus, they conclude that uridine availability can suppress de novo pyrimidine synthesis in differentiated cells. After exhaustively ruling out other possible mechanisms of regulation, the authors identify phosphorylation events on CAD that are upregulated in stem-like cells, in particular a novel site S1900, whose phosphorylation appears to be independent of MTOR. They then perform elegant mutagenesis of this residue to show that its phosphorylation activates de novo pyrimidine synthesis. Overall, this is an exciting manuscript that develops a new robust technique to comprehensively trace nitrogen metabolism, validates it using known biology, uses it to find an interesting new observation with cancer relevance (essentially absent de novo pyrimidine synthesis in differentiated cells), and identify a plausible mechanism and a new regulatory phosphorylation event on the rate limiting enzyme in the pathway. Controls are bountiful and numerous model systems and approaches are used. Conclusions are well drawn. I have only a few modest suggestions to improve upon this work, primarily regarding some further characterization of the S1900 phosphorylation site.

Major Comments

The experiments in 7a-d convincingly show that the S1900 residue affects the activity of de novo pyrimidine synthesis in differentiated cells, but the causality of this mechanism for the rest of their findings is somewhat incompletely explored but I think could be answered and strengthen the impact and importance of the manuscript:

Does mutating S1900 to A in neoplastic cells impair their ability to synthesize pyrimidines de novo?

Do the S1900A mutants in NHAs/CHO cells fail to upregulate de novo pyrimidine synthesis in response to uridine (and secondarily, in the uridine depletion experiments in differentiated cells where de novo pyrimidine synthesis increases-does S1900 phosphorylation increase there too?)

Minor comment: Can more details be provided regarding non-malignant brain parenchyma removal during VP shunt placement? I was not aware that was a standard part of shunt placement.

It is likely beyond the scope of this manuscript, but I would love to know if the surrounding amino acids of S1900 suggest a possible kinase responsible for this newly-described phosphorylation.

(Remarks on code availability)

Reviewer #2

(Remarks to the Author)

In their manuscript, Savani et al comprehensively measure nitrogen labeling patterns upon providing 30 different metabolites in two cell based models. Based on these data, they identify differences in de novo pyrimidine synthesis versus pyrimidine salvage activity, which they then assay in an expanded panel of cell-based models, mouse models, and human models. The authors conclude that differentiated tissues primarily engage in nucleotide salvage, whereas primitive tissues and transformed cells primarily engage in de novo pyrimidine synthesis. The authors go on to demonstrate that differentiated cells can activate de novo pyrimidine synthesis pathways when uridine is withdrawn through an unidentified mechanism. The authors identify a phosphorylation event in CAD (S1900) that is not present in differentiated tissues and may be involved in this phenomenon. A mutant CAD enzyme that harbors a S1900D phosphomimetic mutation exhibits elevated CAD activity in uridine containing media.

Overall, the manuscript is a technical tour de force which reports a massive amount of data and will be an important resource for the field.

Major Points

1. There is slightly less enthusiasm for the novelty of the specific downstream pathway that the authors chose to focus on given that there is substantial literature describing the relative de novo versus salvage pathway activity of various tissues and transformed cells. However, one might consider this an important example that demonstrates the robustness of the platform and analyses. The authors might spend some more time describing what is known about salvage versus de novo pyrimidine synthesis pathway activity from the literature in the discussion.
2. The authors report that a CAD phosphorylation site correlates with models expected to have higher de novo synthesis pathway activity. However, the importance of this mutation is not well-addressed in this manuscript. As it is unclear if phosphorylation of CAD S1900 is necessary for activity, the investigators should report the activity of a S1900A single mutant. Beyond that, too much additional work in this area is probably out of scope; however, some additional discussion is warranted. Indeed, it is unclear from the manuscript what the authors consider might be the function of the S1900A mutant. Is it important for uridine/pyrimidine sensing? Do they think that it is a state-specific mark that increases baseline CAD activity? Or do they think it is altered in response to specific stimuli. It is understandable that the authors might not have answers to all of these questions, but it is difficult to understand from the discussion what possibilities the authors might be considering.
3. The authors conclude that allosteric mechanisms are not at play when uridine is depleted due to the unchanged levels of allosteric regulators at a single time point 24hrs after uridine depletion. However, as allosteric regulation is expected to occur very rapidly and should act to restore downstream metabolite levels, it seems difficult to reject this hypothesis. Is there another method that the investigators could use to address this question? Alternatively, the authors might further discuss this point.

Minor Points / Suggestions

4. While impressive, I am a bit confused by the labeling scheme description and the use of the word “simultaneous” to describe the experimental setup. From my understanding of the manuscript the investigators replaced individual metabolites in HPLM with ¹⁵N labeled derivatives followed by LC/MS and analysis of mass shifts for each individual condition. This setup seems to be parallelized rather than simultaneous—the latter makes me think that multiple metabolites were labeled in a single experiment, in which case one would need to better describe how the data were deconvoluted. Could the authors clarify? If the setup is indeed parallel it would probably be necessary to reframe the beginning of the abstract and introduction to better describe the significance of the study, for example that they (quite impressively) conducted a massively parallel effort to trace a comprehensive panel ¹⁵N metabolites in a single experimental system.

(Remarks on code availability)

Reviewer #3

(Remarks to the Author)

This manuscript by Savani et al uses stable-isotope tracing methods for nitrogen-containing metabolites and identifies differential preferences for de novo versus salvage pyrimidine synthesis pathways that distinguish stem/progenitor (and malignant) cells from differentiated cells. Progenitor/malignant cells are identified to have phosphorylation at S1900 of the CAD enzyme which is proposed to make them impervious to uridine-mediated repression of de novo pyrimidine synthesis, in contrast to differentiated cells which robustly suppress de novo pyrimidine synthesis in response to uridine.

Thirty different formulations of human plasma-like media (HPLM) are developed in which the major nitrogen containing components are replaced with the ¹⁵N-containing counterpart. Tracing is performed with each of the 30 HPLM variants with detection of label incorporation into ~100 nitrogen containing metabolites. Data integrated into Sankey plots for global visualization. Data can be used to compare different cell types or cells subjected to defined interventions. Effect of mutant IDH derived 2HG on inhibiting BCAT-dependent glutamate synthesis is used to validate the system. Next labeling system is

applied to compare glioma stem cells (GSC) to immortalized normal human astrocytes (NHA). GSC show usage of de novo (via glutamine) and salvage (via uridine) pathways for pyrimidine synthesis, whereas NHA show near exclusive use of salvage pathway. This is finding confirmed in many different cell lines (including isogenic fibroblast and iPSC/NPC pairs), in vivo mice using conventional tracing and MALDI special tracing, in vivo adult vs fetal mice, and in human brain/tumor specimens cultured ex vivo. Differentiated cells are shown to be able to activate de novo pyrimidine synthesis in the absence of uridine, and uridine exposure rapidly suppresses de novo pyrimidine synthesis only in differentiated cells. Phosphorylation at S1900 on CAD enzyme is identified to be enriched in primitive and/or tumor cells. Finally, some experiments are done to assess functional role of CAD S1900 phosphorylation in dictating switch between de novo or salvage pathways.

Overall, this is a very well-written manuscript. The experiments are innovative and rigorous. The results mostly support the conclusions. The newly described approaches for assessing differential metabolism of nitrogen-containing nutrients will be of interest to the readers of Nature Metabolism. The phenomenon of differential usage of de novo versus salvage pathways for pyrimidine synthesis is robustly supported. There are a few issues that should be addressed to strengthen the conclusions of the manuscript.

Major:

1) Causal role of CAD S1900 phosphorylation is not well supported. Figure 7 seems to be the weakest part of the manuscript. In this figure, the activation of the de novo pathway in the absence of uridine is much less robust than shown earlier throughout the paper. It is not clear why NHA EV cells do not seem to label carbamoyl aspartate at all in the absence of uridine where in Fig 5p the labelling was rapid and robust. Likewise, the difference between WT CAD cells in the presence and absence of uridine is negligible. Extended Data Fig 7 shows that expression of the WT is less than the CAD mutants, so its overall lesser effects may just be due to different expression levels. For this reason, the WT CAD cells should not be used as the sole reference point. The figure should show the +uridine and -uridine for each of the constructs and conditions, for both labeling and the metabolomics volcano plots. This will allow the reader to clearly see if manipulating S1900 abolishes (with phospho mimetic aspartate) or enhances (with Alanine mutants) responsiveness to uridine.

- Another issue related to the volcano plots in Fig 7. An argument is made here stating that only the S1900D was able to boost pyrimidine levels. This needs to be reconciled with the previous statement in Line 314 that "Pyrimidine synthesis pathway switching occurred without cyclical effects on UMP, UTP, or other metabolites downstream of de novo and salvage pathway convergence." Again, this effect of CAD S1900D compared to CAD WT may simply be due to higher expression levels, so the volcano plot would be more informative if it showed +uridine and -uridine conditions for each construct.

2) Lack any physiologic relevance for preference of pyrimidine pathway usage. The manuscript does not offer any experimental evidence (or even speculation) on the potential physiologic function for the "switch" in pyrimidine pathway preference upon differentiation. There are simple in vitro systems of progenitor cell differentiation, such as C2C12 cells (PMID: 35264789) or 10T1/2 cells (PMID: 31182575), that could be used to better demonstrate that a cell induced to differentiate undergoes this pyrimidine pathway switch, and to then determine whether manipulating pathway usage impacts differentiation. It would be important to show whether uridine withdrawal impedes differentiation, or whether disrupting the de novo pathway (CAD KO, CAD S1900 mutation, or even glutamine restriction) promotes differentiation. While the isogenic fibroblast/iPSC/NPC model provided some better evidence that the switch appears to dependent on differentiation state, this system is not really amenable to mechanistic studies (unless it is possible to differentiate the NPCs to neurons).

Minor:

1) The discussion about normalization is confusing and could use more explicit clarification. The manuscript states in Line 1761 "All amide-15N-glutamine to UMP labeling data represent UMP M+1 FE normalized to glutamine M+1 FE." In Line 293 "Like cell culture experiments (Extended Data Fig. 2l), glutamine tracer accumulation was higher in non-malignant brain than glioma explants, whereas uridine accumulation was similar (Extended Data Fig. 4a-b). These data suggest our assay may underestimate differences in de novo pathway activity between glioma and nonmalignant brain due to larger unlabeled glutamine pools in the former." Where does the evidence for larger unlabeled pools of glutamine in glioma come from? Is this shown directly in the manuscript or it is from the literature? If the primitive/malignant cells have a lower value for glutamine M+1 FE (due to larger pool size of unlabeled) wouldn't this cause a greater difference after normalization (i.e. make the de novo pathway seem more active in the primitive cells)? To make this clearer, it would be helpful to include examples of unnormalized versus normalized data in the extended data.

2) Please comment on Ext Data 2b showing major differences in uridine and aspartate (and other AA) uptake by BT54, whether this would or would not be expected have any impact on the findings.

3) To expand the significance of the CAD S1900 findings, would it be possible to use the kinase atlas to identify predicted kinases for this motif (PMID: 36631611).

(Remarks on code availability)

I do not have the expertise to review code

Version 1:

Reviewer comments:

Reviewer #1

(Remarks to the Author)

All of my comments have been addressed. The further functional validation of S1900 with extensive biochemical studies has greatly improved the manuscript.

(Remarks on code availability)

Reviewer #2

(Remarks to the Author)

This authors have addressed all of my points. I find this an outstanding contribution.

(Remarks on code availability)

Reviewer #3

(Remarks to the Author)

The authors have done an admirable job of addressing the reviewer critiques. There is substantial new data to bolster the conclusions and strengthen the quality of the manuscript.

(Remarks on code availability)

NA

We sincerely thank the reviewers and editors for their insightful feedback. Their valuable comments have helped us considerably strengthen the manuscript. Please see our responses to comments below:

Response to Editors' Comments

All three reviewers have made suggestions how to strengthen the evidence for the role of CAD phosphorylation in the regulation of the proposed pathway switch, which we ask you to incorporate.

We include additional functional data demonstrating the important regulatory role of CAD S1900 phosphorylation:

- 1) We show that the CAD S1900D phosphomimetic mutant is sufficient to activate de novo pyrimidine synthesis in differentiated cells, including both NHA (**Fig. 6d-j**) and CAD-deficient G9C lines (**Extended Data Fig. 8e-g**), grown under uridine-replete conditions.
- 2) We show that endogenous CAD S1900 phosphorylation is increased in differentiated NHA cells during activation of de novo pyrimidine synthesis caused by acute uridine starvation (**Fig. 6a-c**).
- 3) We show that CAD S1900 phosphorylation is necessary for differentiated NHA cells to activate de novo pyrimidine synthesis in response to acute uridine starvation (**Fig. 6d-n**).

We also like the suggestion from reviewer #3 to study the physiological relevance of the reported pathway switch in differentiating cells, and encourage to consider these experiments, as we think they would increase the novelty and impact of your study.

We now include new data that describe how pyrimidine synthesis pathway choice affects the behavior of both primitive and differentiating cells. Building on the reviewer's suggestion, we induced differentiation of NPCs to neurons in the presence and absence of uridine (**Fig. 2q-r**). Neurons expressed markers of the interneuron lineage, including DLX1 and DLX2, and suppressed stemness markers, including OLIG2, SOX2, and NES (**Fig. 2r**). We assessed gene expression programs in differentiating neurons grown in the presence and absence of uridine to gain insight into the function of pyrimidine salvage in this cell state. We also evaluated self-renewal capacity of NPCs treated with or without an inhibitor (orludodstat) of the de novo pyrimidine synthesis pathway enzyme DHODH and cultured under a range of uridine concentrations. We found:

- 1) Neurons differentiated from NPCs completely repress de novo pyrimidine synthesis and rely on uridine salvage to synthesize UMP (**Fig. 2u**). De novo pyrimidine synthesis in these cells is rapidly stimulated by uridine deprivation (**Fig. 5o**), similar to other differentiated cell types. This sharply contrasts with results in NPCs, in which de novo pyrimidine synthesis predominates under basal conditions (**Fig. 2t**) and is less affected by uridine withdrawal (**Fig. 5o**).
- 2) DHODH inhibition reduced NPC self-renewal in the absence of uridine and in the presence of physiological (3uM) and supraphysiological (100uM) levels of uridine (**Fig. 2v-y**). The effect of DHODH inhibition on self-renewal was reduced but not abolished by uridine supplementation (**Fig. 2y**). Effects of DHODH inhibition on NPC self-renewal were not merely secondary effects of cell death, as we did not observe substantial cytotoxicity of DHODH inhibitor treatment (**Extended Data Fig. 3f**). Therefore, we conclude that de novo pyrimidine synthesis is required to sustain NPC multipotency and that engagement of the pyrimidine salvage pathway cannot fully compensate for this requirement.

- 3) Deprivation of uridine, the preferred substrate for neuronal pyrimidine synthesis (**Fig. 2u**), did not affect the efficiency with which neurons could be differentiated from NPCs. Indeed, the expression of stemness (**Extended Data Fig. 3g-h**) and neuronal (**Extended Data Fig. 3i-j**) markers was not affected by depriving differentiating neurons of uridine. However, whole transcriptome analysis of these cultures revealed that uridine salvage promoted expression of gene programs that are vital to neuronal function. These programs involved genes related to regulation of cell polarity, neuronal axonogenesis, and neuronal migration (**Fig. 2aa-bb**). These data suggest that the switch in pyrimidine synthesis pathway dependence observed during neuronal differentiation may play an important role in establishing neuronal function.

As reviewer #2 pointed out, it will be important that the language in the study reflects the nature of the experimental approach ("simultaneous" vs "parallel" measurements").

We have replaced all "simultaneous" descriptors with "parallel". Moreover, we have modified the abstract and introduction sections to clarify that our nitrogen metabolism profiling platform does not involve combining ¹⁵N-labeled metabolite tracers (see lines 57 and 89). Rather, tracing experiments were conducted individually in a massively parallel fashion.

Response to Reviewers' Comments

Reviewer 1:

The experiments in 7a-d convincingly show that the S1900 residue affects the activity of de novo pyrimidine synthesis in differentiated cells, but the causality of this mechanism for the rest of their findings is somewhat incompletely explored but I think could be answered and strengthen the impact and importance of the manuscript:

Does mutating S1900 to A in neoplastic cells impair their ability to synthesize pyrimidines de novo?

This is an important point and one that we have been working diligently to address. Glioma stem-like cells (GSCs) are primitive, neoplastic cells and a definitive test of necessity would require allelic replacement of endogenous CAD. In patient-derived GSCs, sustained CAD knockout is strongly selected against. This is not unexpected given that CAD inactivation can be poorly tolerated, as evidenced by CAD being a strongly selective dependency in cancer cells (DepMap), CAD knockout being embryonic lethal in mice (Mouse Genome Informatics), and partial loss of CAD activity causing human disease (PMID 25678555). To address necessity of CAD S1900 phosphorylation in GSCs, we have tried to first exogenously express CRISPR-resistant CAD WT or CAD S1900A cDNAs, followed by knocking out the endogenous CAD gene and performing amide-¹⁵N-glutamine tracing to assess de novo pyrimidine synthesis. Anticipating that the CAD S1900A-expressing line may display loss of fitness, we tried to generate these lines in the presence of supraphysiological uridine (100 μM) to alleviate nucleotide depletion stress. However, this experiment has been hampered by loss of cell growth and viability. We considered an alternative approach involving CRISPR HDR editing to introduce the S1900A mutation into GSCs, but this approach typically requires single cell cloning to isolate engineered clones. Under low density conditions, GSC lines display doubling times of 4+ days. This slow growth may be exacerbated by stress caused by the CAD S1900A mutation. Therefore, this approach is poorly suited to CAD manipulation in GSCs.

We have, however, addressed the broader theme of the functional relevance of CAD S1900 phosphorylation through several key studies:

- 1) We show that the CAD S1900D phosphomimetic mutant is sufficient to activate de novo pyrimidine synthesis in differentiated cells, including both NHA (**Fig. 6d-j**) and CAD-deficient G9C lines (**Extended Data Fig. 8e-g**), grown under uridine-replete conditions.
- 2) We show that endogenous CAD S1900 phosphorylation is increased in differentiated NHA cells during activation of de novo pyrimidine synthesis caused by acute uridine starvation (**Fig. 6a-c**).
- 3) We show that CAD S1900 phosphorylation is necessary for differentiated NHA cells to activate de novo pyrimidine synthesis in response to acute uridine starvation (**Fig. 6d-n**).

Collectively, these data establish an important role for CAD S1900 phosphorylation in controlling de novo pyrimidine synthesis pathway activity.

Do the S1900A mutants in NHAs/CHO cells fail to upregulate de novo pyrimidine synthesis in response to uridine (and secondarily, in the uridine depletion experiments in differentiated cells where de novo pyrimidine synthesis increases-does S1900 phosphorylation increase there too?)

We thank the reviewer for this valuable suggestion. We now include the following new data:

- 1) We show that endogenous CAD S1900 phosphorylation is increased in differentiated NHA cells during activation of de novo pyrimidine synthesis caused by acute uridine starvation (**Fig. 6a-c**).
- 2) We show that CAD S1900 phosphorylation is necessary for differentiated NHA cells to activate de novo pyrimidine synthesis in response to acute uridine starvation (**Fig. 6f-n**). Notably, data in **Fig. 6k and 6m** show that NHA cells expressing CAD S1900A are defective in their ability to i) generate pyrimidines from the de novo synthesis pathway and ii) maintain global pyrimidine pools following uridine starvation compared to NHA cells expressing WT CAD. This defect is specific for uridine starvation because pyrimidine pool sizes do not differ between NHA cells expressing WT CAD or CAD S1900A cultured under uridine-replete conditions (**Fig. 6g and 6i**).

Minor comment: Can more details be provided regarding non-malignant brain parenchyma removal during VP shunt placement? I was not aware that was a standard part of shunt placement.

We are happy to provide more details on this approach. During ventriculoperitoneal shunt placement, after a burr hole is made in the calvarium, the dura is opened, and the pia gently incised. Then, bipolar electrocautery is used to obtain hemostasis in this region before a catheter is passed through that cortical rim and into the ventricular corridor. During this process, normal brain is coagulated and undergoes damage from the catheter and displacement. Under an IRB-approved protocol, a small (1-2 mm) piece of cortical tissue is taken prior to coagulation and passing the catheter through the parenchymal region and used for investigation. This small piece of tissue would otherwise be coagulated, discarded, or damaged by catheter placement at the entry site.

It is likely beyond the scope of this manuscript, but I would love to know if the surrounding amino acids of S1900 suggest a possible kinase responsible for this newly-described phosphorylation.

We share the reviewer's interest in this point and have begun to search for the relevant kinase(s). We see a strong signature for AMPK family kinases emerge from a bioinformatic analysis of the primary amino acid sequence surrounding CAD S1900, including high scores for NUAKs, SIKs, and MARKs. We now include results of this bioinformatic analysis in

Supplementary Table 6. We agree that functional evaluation of the kinase(s) responsible for CAD S1900 phosphorylation extends beyond the scope of this paper.

Reviewer 2:

Major Points

1. There is slightly less enthusiasm for the novelty of the specific downstream pathway that the authors chose to focus on given that there is substantial literature describing the relative de novo versus salvage pathway activity of various tissues and transformed cells. However, one might consider this an important example that demonstrates the robustness of the platform and analyses. The authors might spend some more time describing what is known about salvage versus de novo pyrimidine synthesis pathway activity from the literature in the discussion.

We thank the reviewer for this helpful feedback. We acknowledge that there are papers describing differences in pyrimidine synthesis pathway use in various biological contexts. However, we believe a novel contribution of our study is showing that these differences in pyrimidine synthesis pathway choice are i) cell autonomous, ii) manifest under conditions of equivalent nutrient availability, and iii) are principally linked to cell state, not oncogenic mutations or malignant transformation per se. Moreover, this line of investigation has led us to identify CAD S1900 phosphorylation as a key driver of de novo pyrimidine synthesis pathway activity, which was unexpected. We agree that a more extensive discussion of pyrimidine synthesis pathway choice in the context of past studies is warranted, and we now include that in the revised manuscript (see lines 563-576, 593-605, and 621-628).

2. The authors report that a CAD phosphorylation site correlates with models expected to have higher de novo synthesis pathway activity. However, the importance of this mutation is not well-addressed in this manuscript. As it is unclear if phosphorylation of CAD S1900 is necessary for activity, the investigators should report the activity of a S1900A single mutant. Beyond that, too much additional work in this area is probably out of scope; however, some additional discussion is warranted. Indeed, it is unclear from the manuscript what the authors consider might be the function of the S1900A mutant. Is it important for uridine/pyrimidine sensing? Do they think that it is a state-specific mark that increases baseline CAD activity? Or do they think it is altered in response to specific stimuli. It is understandable that the authors might not have answers to all of these questions, but it is difficult to understand from the discussion what possibilities the authors might be considering.

We thank the reviewer for pointing out this lack of clarity in our original manuscript. We include new data and more precisely worded conclusions to describe the regulatory role of CAD S1900 phosphorylation in our revised paper. We now include the following two points in the Discussion section:

First, we conclude that differentiated cells stimulate CAD S1900 phosphorylation during uridine starvation and that this signaling response is necessary for de novo pyrimidine synthesis pathway activation (see lines 586-591). This conclusion is supported by the following new data:

- 1) We show that endogenous CAD S1900 phosphorylation is increased in differentiated NHA cells during activation of de novo pyrimidine synthesis caused by acute uridine starvation (**Fig. 6a-c**).
- 2) We show that CAD S1900 phosphorylation is necessary for differentiated NHA cells to activate de novo pyrimidine synthesis in response to acute uridine starvation (**Fig. 6f-n**). Notably, data in **Fig. 6k and 6m** show that NHA cells expressing CAD S1900A are

defective in their ability to i) generate pyrimidines from the de novo synthesis pathway and ii) maintain global pyrimidine pools following uridine starvation compared to NHA cells expressing WT CAD. This defect is specific for uridine starvation because pyrimidine pool sizes do not differ between NHA cells expressing WT CAD or CAD S1900A cultured under uridine-replete conditions (**Fig. 6g and 6i**).

Second, we propose that CAD S1900 phosphorylation contributes to constitutive de novo pyrimidine synthesis pathway activation in primitive cells (see lines 607-615). This model is supported by data showing that i) expression of the phosphomimetic CAD S1900D mutant is sufficient to activate de novo pyrimidine synthesis in both differentiated NHA (**Fig. 6f-j**) and G9C (**Extended Data Fig. 8e-g**) cells and ii) that CAD S1900 phosphorylation positively correlates with primitive cell markers in human tissues (**Fig. 7a-b, 7g-i and Extended Data Fig. S9r-s**).

These findings underpin our working model linking cell state, pyrimidine synthesis pathway choice, uridine availability, and CAD S1900 phosphorylation, which is now depicted in **Fig. 7n**.

3. The authors conclude that allosteric mechanisms are not at play when uridine is depleted due to the unchanged levels of allosteric regulators at a single time point 24hrs after uridine depletion. However, as allosteric regulation is expected to occur very rapidly and should act to restore downstream metabolite levels, it seems difficult to reject this hypothesis. Is there another method that the investigators could use to address this question? Alternatively, the authors might further discuss this point.

We agree that evaluating early time points following uridine deprivation is required to assess potential relevance of allosteric regulatory mechanisms governing CAD and UMPS activity by UTP and uric acid, respectively. We evaluated levels of UTP and uric acid at 1hr (**Extended Data Fig. 7e-g**) and 6hr (**Extended Data Fig. 7h-j**) of uridine deprivation in immortalized astrocytes. Importantly, at these time points we observe increases in carbamoyl aspartate labeling by amide-¹⁵N-glutamine (**Extended Data Fig. 7e and 7h**), indicative of de novo pyrimidine synthesis activation. However, we do not observe significant depletion of UTP or uric acid at these time points. These data support our conclusion that de novo pyrimidine synthesis activation in response to uridine starvation manifests, at least at early time points, independently of established allosteric regulatory mechanisms. To clarify the time points evaluated in **Extended Data Fig. 7e-j**, we have included schema describing the timelines of these experiments above the data.

Minor Points / Suggestions

4. While impressive, I am a bit confused by the labeling scheme description and the use of the word “simultaneous” to describe the experimental setup. From my understanding of the manuscript the investigators replaced individual metabolites in HPLM with ¹⁵N labeled derivatives followed by LC/MS and analysis of mass shifts for each individual condition. This setup seems to be parallelized rather than simultaneous—the latter makes me think that multiple metabolites were labeled in a single experiment, in which case one would need to better describe how the data were deconvoluted. Could the authors clarify? If the setup is indeed parallel it would probably be necessary to reframe the beginning of the abstract and introduction to better describe the significance of the study, for example that they (quite impressively) conducted a massively parallel effort to trace a comprehensive panel ¹⁵N metabolites in a single experimental system.

We regret the confusion caused by our use of the term “simultaneous” to describe our platform. The reviewer’s interpretation is indeed correct: we conducted a parallel effort to trace a panel of

¹⁵N-labeled metabolites. Each ¹⁵N-labeled metabolite was traced individually; there were no experimental conditions in which distinct ¹⁵N-labeled metabolites were combined and traced simultaneously in the same tissue culture well or dish. We have replaced all “simultaneous” descriptors with “parallel”. Moreover, we have modified the abstract and introduction sections to clarify this important issue the reviewer highlights (see lines 57 and 89). We thank the reviewer for raising this helpful suggestion.

Reviewer 3:

Major:

1) Causal role of CAD S1900 phosphorylation is not well supported. Figure 7 seems to be the weakest part of the manuscript. In this figure, the activation of the de novo pathway in the absence of uridine is much less robust than shown earlier throughout the paper. It is not clear why NHA EV cells do not seem to label carbamoyl aspartate at all in the absence of uridine where in Fig 5p the labelling was rapid and robust. Likewise, the difference between WT CAD cells in the presence and absence of uridine is negligible. Extended Data Fig 7 shows that expression of the WT is less than the CAD mutants, so its overall lesser effects may just be due to different expression levels. For this reason, the WT CAD cells should not be used as the sole reference point. The figure should show the +uridine and -uridine for each of the constructs and conditions, for both labeling and the metabolomics volcano plots. This will allow the reader to clearly see if manipulating S1900 abolishes (with phospho mimetic aspartate) or enhances (with Alanine mutants) responsiveness to uridine.

We agree with the reviewer’s point that the data provided in Figure 7 in the initial submission could be improved and thank them for this helpful feedback. We have extensively revised this figure and now include updated data in Figure 6. We have swapped Figures 6 and 7 so that we can more effectively introduce CAD S1900 phosphorylation as a modification that is stimulated by acute uridine deprivation in differentiated cells, a finding that is new to the revised manuscript (**Fig. 6a-c**). We have also simplified comparisons across culture conditions and between CAD enzymes. We compare differentiated NHA stable cell lines expressing an empty vector negative control (EV), WT CAD, the phosphomimetic CAD S1900D mutant, or the phosphodeficient CAD S1900A mutant grown under uridine-replete conditions or subjected to uridine deprivation for 18 hours (**Fig. 6d-n and Extended Data Fig. 8d**). We show that CAD S1900 phosphorylation is necessary for differentiated NHA cells to activate de novo pyrimidine synthesis in response to acute uridine starvation (**Fig. 6f-n**). Notably, data in **Fig. 6k and 6m** show that NHA cells expressing CAD S1900A are defective in their ability to i) generate pyrimidines from the de novo synthesis pathway and ii) maintain global pyrimidine pools following uridine starvation compared to NHA cells expressing WT CAD. This defect is specific for uridine starvation because pyrimidine pool sizes do not differ between NHA cells expressing WT CAD or CAD S1900A cultured under uridine-replete conditions (**Fig. 6g and 6i**). We have addressed specific comments by the reviewer as follows:

- 1) We show UMP M+1 peak intensities normalized to NHA-WT CAD cells in both uridine-replete and uridine-depleted conditions following amide-¹⁵N-glutamine tracing in **Fig. 6f**. These data are normalized within each condition to facilitate comparisons between CAD mutant and CAD WT enzymes. Activation of de novo pyrimidine synthesis in NHA-EV cells is demonstrated by a narrowing in the difference of UMP M+1 levels between NHA-EV and NHA-WT CAD lines in the absence of uridine versus the presence of uridine. To complement the normalized data and show that the effect of uridine deprivation reported in earlier figures remains intact in this figure, we show un-normalized data for NHA-EV

and NHA-WT CAD cells in **Extended Data Fig. 8d**. UMP M+1 pool sizes are affected by both CAD expression levels and uridine availability.

- 2) The blot we used to compare ectopically expressed WT and mutant CAD enzymes in the initial submission was affected by variable protein loading. We now show representative blots for the FLAG tag on exogenous CAD enzymes, total CAD levels, and a loading control in **Fig. 6d**. We also performed 4 independent blots with densitometry and show that expression levels of FLAG-tagged exogenous CAD enzymes do not differ between NHA-WT CAD, NHA-CAD S1900A, and NHA-CAD S1900D stable lines (**Fig. 6e**). Therefore, metabolic differences between these lines shown in **Fig. 6f-n** cannot be explained by differences in enzyme expression.
- 3) In **Fig. 6f**, we show UMP M+1 peak intensities relative to NHA-WT CAD cells. To avoid using these cells as the sole reference point, we show volcano plots (**Fig. 6g-n**) describing differences in both amide-¹⁵N-Gln-labeled metabolites as well as total metabolite pool sizes relative to NHA-CAD S1900A cells. We chose NHA-CAD S1900A cells as the reference point for volcano plots because we can be assured that this exogenous CAD enzyme remains fully unphosphorylated in both the presence and absence of uridine. All data are shown in both uridine-replete and uridine-depleted conditions. Importantly, in the presence of uridine, pyrimidine pools do not differ between NHA-WT CAD cells and NHA-CAD S1900A cells (**Fig. 6f, 6g, 6i**), consistent with low basal CAD S1900 phosphorylation under uridine-replete conditions in NHA cells (**Fig. 6c**). In contrast, NHA-WT CAD cells display increases in both labeled and total pyrimidine pool sizes relative to NHA-CAD S1900A cells under uridine-depleted conditions (**Fig. 6f, 6k, 6m**). These data establish that CAD S1900 phosphorylation is required for de novo pyrimidine synthesis pathway activation in differentiated NHA cells subjected to uridine starvation.
- 4) We also directly compare NHA-CAD S1900D and NHA CAD S1900A cells via volcano plots. In both the presence (**Fig. 6f, 6h, 6j**) and absence (**Fig. 6f, 6l, 6n**) of uridine, NHA-CAD S1900D cells display increases in both labeled and total pyrimidine pool sizes relative to NHA-CAD S1900A cells. These data establish that CAD S1900 phosphorylation is sufficient to activate de novo pyrimidine synthesis in differentiated cells independent of uridine availability. This conclusion is corroborated by similar findings in G9C CHO cells (**Extended Data Fig. 8e-g**). Comparing NHA-WT CAD and NHA-CAD S1900D cells (**Fig. 6f**), we find that uridine starvation diminishes, but does not abolish, the difference in UMP M+1 abundance between these lines. This finding is consistent with the observation that some, but not all, of the CAD enzyme pool undergoes phosphorylation at S1900 in NHA cells in response to uridine deprivation (**Fig. 6c**).

• Another issue related to the volcano plots in Fig 7. An argument is made here stating that only the S1900D was able to boost pyrimidine levels. This needs to be reconciled with the previous statement in Line 314 that “Pyrimidine synthesis pathway switching occurred without cyclical effects on UMP, UTP, or other metabolites downstream of de novo and salvage pathway convergence.” Again, this effect of CAD S1900D compared to CAD WT may simply be due to higher expression levels, so the volcano plot would be more informative if it showed +uridine and -uridine conditions for each construct.

We thank the reviewer for this valuable point. As mentioned above, we now include immunoblot/densitometry quantification of CAD enzyme expression in NHA (**Fig. 6d-e**) and G9C (**Extended Data Fig. 8e-f**) cells and volcano plots for both labeled and total metabolite pool sizes in NHA stable lines under both uridine-replete and uridine-depleted conditions (**Fig. 6g-n**).

In our prior Figure 7 and current **Fig. 6i-j**, we indeed show that CAD S1900D, but not CAD WT, increased global pyrimidine levels in NHA cells cultured under uridine-replete conditions. We grew these cells at a density at which uridine would not be substantially depleted over the course of the experiment. In contrast, our experiment referenced on line 314 of the initial submission (referencing current **Fig. 5b-c and Extended Data Fig. 6a-c**) was performed at a higher cell density that induces substantial uridine depletion from the culture medium over time. Therefore, these two experiments are difficult to compare directly. To address this important issue raised by the reviewer, we have modified the Results section to clearly indicate that cells were grown near confluence to induce substantial medium uridine depletion so that we could observe the resulting cellular response (see lines 336-339).

2) Lack any physiologic relevance for preference of pyrimidine pathway usage. The manuscript does not offer any experimental evidence (or even speculation) on the potential physiologic function for the “switch” in pyrimidine pathway preference upon differentiation. There are simple in vitro systems of progenitor cell differentiation, such as C2C12 cells (PMID: 35264789) or 10T1/2 cells (PMID: 31182575), that could be used to better demonstrate that a cell induced to differentiate undergoes this pyrimidine pathway switch, and to then determine whether manipulating pathway usage impacts differentiation. It would be important to show whether uridine withdrawal impedes differentiation, or whether disrupting the de novo pathway (CAD KO, CAD S1900 mutation, or even glutamine restriction) promotes differentiation. While the isogenic fibroblast/iPSC/NPC model provided some better evidence that the switch appears to dependent on differentiation state, this system is not really amenable to mechanistic studies (unless it is possible to differentiate the NPCs to neurons).

We completely agree with the concern raised by the reviewer. We now include new data that describe how pyrimidine synthesis pathway choice affects the behavior of both primitive and differentiating cells. Building on the reviewer's suggestion, we induced differentiation of NPCs to neurons in the presence and absence of uridine (**Fig. 2q-r**). Neurons expressed markers of the interneuron lineage, including DLX1 and DLX2, and suppressed stemness markers, including OLIG2, SOX2, and NES (**Fig. 2r**). We assessed gene expression programs in differentiating neurons grown in the presence and absence of uridine to gain insight into the function of pyrimidine salvage in this cell state. We also evaluated self-renewal capacity of NPCs treated with or without an inhibitor (orludodstat) of the de novo pyrimidine synthesis pathway enzyme DHODH and cultured under a range of uridine concentrations. We found:

- 1) Neurons differentiated from NPCs completely repress de novo pyrimidine synthesis and rely on uridine salvage to synthesize UMP (**Fig. 2u**). De novo pyrimidine synthesis in these cells is rapidly stimulated by uridine deprivation (**Fig. 5o**), similar to other differentiated cell types. This sharply contrasts with results in NPCs, in which de novo pyrimidine synthesis predominates under basal conditions (**Fig. 2t**) and is less affected by uridine withdrawal (**Fig. 5o**).
- 2) DHODH inhibition reduced NPC self-renewal in the absence of uridine and in the presence of physiological (3uM) and supraphysiological (100uM) levels of uridine (**Fig. 2v-y**). The effect of DHODH inhibition on self-renewal was reduced but not abolished by uridine supplementation (**Fig. 2y**). Effects of DHODH inhibition on NPC self-renewal were not merely secondary effects of cell death, as we did not observe substantial cytotoxicity of DHODH inhibitor treatment (**Extended Data Fig. 3f**). Therefore, we conclude that de novo pyrimidine synthesis is required to sustain NPC multipotency and that engagement of the pyrimidine salvage pathway cannot fully compensate for this requirement.

- 3) Deprivation of uridine, the preferred substrate for neuronal pyrimidine synthesis (**Fig. 2u**), did not affect the efficiency with which neurons could be differentiated from NPCs. Indeed, the expression of stemness (**Extended Data Fig. 3g-h**) and neuronal (**Extended Data Fig. 3i-j**) markers was not affected by depriving differentiating neurons of uridine. However, whole transcriptome analysis of these cultures revealed that uridine salvage promoted expression of gene programs that are vital to neuronal function. These programs involved genes related to regulation of cell polarity, neuronal axonogenesis, and neuronal migration (**Fig. 2aa-bb**). These data suggest that the switch in pyrimidine synthesis pathway dependence observed during neuronal differentiation may play an important role in establishing neuronal function.

Minor:

1) The discussion about normalization is confusing and could use more explicit clarification. The manuscript states in Line 1761 “All amide-¹⁵N-glutamine to UMP labeling data represent UMP M+1 FE normalized to glutamine M+1 FE.” In Line 293 “Like cell culture experiments (Extended Data Fig. 2l), glutamine tracer accumulation was higher in non-malignant brain than glioma explants, whereas uridine accumulation was similar (Extended Data Fig. 4a-b). These data suggest our assay may underestimate differences in de novo pathway activity between glioma and nonmalignant brain due to larger unlabeled glutamine pools in the former.” Where does the evidence for larger unlabeled pools of glutamine in glioma come from? Is this shown directly in the manuscript or it is from the literature? If the primitive/malignant cells have a lower value for glutamine M+1 FE (due to larger pool size of unlabeled) wouldn't this cause a greater difference after normalization (i.e. make the de novo pathway seem more active in the primitive cells)? To make this clearer, it would be helpful to include examples of unnormalized versus normalized data in the extended data.

We thank the reviewer for this point. The key issue is that GSCs and NSCs display substantial glutamine synthetase activity that contributes to a “cold” pool of intracellular glutamine. This is evident in the 20-30% unlabeled glutamine pool in these cells when cultured in amide-¹⁵N-glutamine tracer-containing medium (**Extended Data Figure 3a, leftmost panel**). We also observed this difference between glioma tissues and non-malignant brain specimens (**Extended Data Figure 5a**). This “cold” pool of glutamine will contribute to UMP synthesis but will not be captured in the UMP M+1 isotopologue pool that we rely on to assess de novo pyrimidine synthesis.

To accurately account for this “cold” pool in cell culture experiments, we normalized data for all cells in **Fig. 2**. In response to the reviewer's comment, we also now report un-normalized data for these experiments in **Supplementary Figure 1**. We are confident that the data in **Fig. 2** reflect a more faithful representation of the differences in pyrimidine synthesis pathway usage among these cells but opted to include un-normalized data to be transparent about the effects of normalization.

Because there may be differences in glutamine tracer penetration into glioma explants and non-malignant brain explants from patients that we cannot account for, we have not normalized glutamine tracing data depicted in **Fig. 4**. This decision served as the rationale to include the statement pertaining to potential underestimation of de novo pyrimidine synthesis pathway activity in glioma explants that the reviewer highlighted.

2) Please comment on Ext Data 2b showing major differences in uridine and aspartate (and

other AA) uptake by BT54, whether this would or would not be expected have any impact on the findings.

Uridine is produced via de novo pyrimidine synthesis pathway activity in BT054 cells. Therefore, uridine-dependent labeling of the intracellular uridine pool is lower in these cells versus NHA cells that do not display substantial de novo pathway flux under basal conditions. This is the case for all GSC lines tested (**Extended Data Fig. 3a, rightmost panel**). Astrocytes are known to consume glutamate and aspartate from the extracellular space. This at least partly attributable to selective expression of the Excitatory Amino Acid Transporter 1 (EAAT1 or SLC1A3). We have included a sentence in the Results section that helps clarify these differences (see lines 202-205).

3) To expand the significance of the CAD S1900 findings, would it be possible to use the kinase atlas to identify predicted kinases for this motif (PMID: 36631611).

We agree that this is a very valuable analysis. We have completed this analysis and observe a strong signature for AMPK family kinases, including high scores for NUAKs, SIKs, and MARKs. We now include results of this bioinformatic analysis in **Supplementary Table 6**. We look forward to identifying and reporting the kinase(s) responsible for CAD S1900 phosphorylation in a future study.